
# Fusion of irreducible modules in the periodic Temperley–Lieb algebra

Yacine Ikhlef[1*] and Alexi Morin-Duchesne[2†]

**1** Sorbonne Université, CNRS, Laboratoire de Physique Théorique et Hautes Énergies, LPTHE, F-75005 Paris, France
**2** Department of Mathematics, Royal Military Academy, 1000 Brussels, Belgium

★ ikhlef@lpthe.jussieu.fr , † alexi.morin.duchesne@gmail.com

## Abstract

We propose a new family $Y_{k,\ell,x,y,[z,w]}$ of modules over the enlarged periodic Temperley–Lieb algebra $\mathcal{E}\mathsf{PTL}_N(\beta)$. These modules are built from link states with two marked points, similarly to the modules $X_{k,\ell,x,y,z}$ that we constructed in a previous paper. They however differ in the way that defects connect pairwise. We analyse the decomposition of $Y_{k,\ell,x,y,[z,w]}$ over the irreducible standard modules $W_{k,x}$ for generic values of the parameters $z$ and $w$, and use it to deduce the fusion rules for the fusion $W \times W$ of standard modules. These turn out to be more symmetric than those obtained previously using the modules $X_{k,\ell,x,y,z}$. From the work of Graham and Lehrer, it is known that, for $\beta = -q - q^{-1}$ where $q$ is not a root of unity, there exists a set of non-generic values of the twist $y$ for which the standard module $W_{\ell,y}$ is indecomposable yet reducible with two composition factors: a radical submodule $R_{\ell,y}$ and a quotient module $Q_{\ell,y}$. Here, we construct the fusion products $W \times R$, $W \times Q$ and $Q \times Q$, and analyse their decomposition over indecomposable modules. For the fusions involving the quotient modules $Q$, we find very simple results reminiscent of $\mathfrak{sl}(2)$ fusion rules. This construction with modules $Y_{k,\ell,x,y,[z,w]}$ is a good lattice regularization of the operator product expansion in the underlying logarithmic bulk conformal field theory. Indeed, it fits with the correspondence between standard modules and connectivity operators, and is useful for the calculation of their correlation functions. Remarkably, we show that the fusion rules $W \times Q$ and $Q \times Q$ are consistent with the known fusion rules of degenerate primary fields.

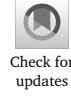
## Contents



# 1   Introduction

The study of random curves in two-dimensional critical models of statistical mechanics has been an active area of research in the last few decades. It is concerned with understanding the properties of curves associated to various types of geometric observables in statistical models, such as the domain walls of critical spin systems [1], the high-temperature closed polygons of the O($n$) vector model [2] and the contour curves of percolation clusters [3]. This field of research is currently referred to as *Random Geometries*. It has close ties to both conformal field theory (CFT) [4, 5] and to the representation theory of lattice diagram algebras, whose archetype is the Temperley–Lieb algebra [6] and its periodic counterpart [7–11]. In the CFT approach, the objects of interest are the *connectivity operators*, namely the non-local operators which encode the connectivity properties of the model's random curves. Depending on the geometry studied, these operators may either be inserted on the boundary of the domain or in its bulk. Boundary connectivity operators have in fact been well understood since the early stages of boundary CFT [12]. Their counterparts in boundary logarithmic CFTs are also well documented. We direct the reader to [13] for a more comprehensive literature review.

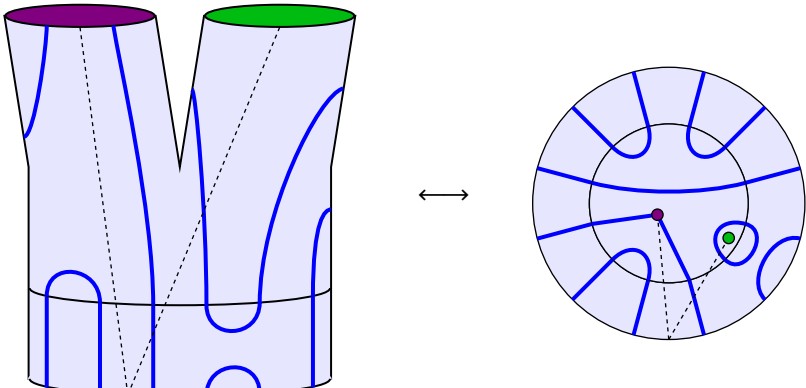

Figure 1: *Left panel:* The action of $e_2 \in \mathcal{E}\mathsf{PTL}_{N=12}(\beta)$ on a link state drawn on a pair of pants with two legs. *Right panel:* The same diagram projected in the plane and thus drawn on a disc with two punctures. Clearly, the diagrams drawn on the disc make the visualisation easier, thus justifying our choice to use this presentation throughout.

The central unsolved problem is then the determination of a consistent operator algebra for the *bulk* connectivity operators. Up to now, there have been three attempts at constructing this operator algebra using lattice regularisations: a first by Gainutdinov, Jacobsen and Saleur [14,15], a second by Belletête and Saint-Aubin [16], and a third proposed in our previous work [13]. All three focus on the representation theory of the *enlarged periodic Temperley–Lieb algebra* $\mathcal{E}\mathsf{PTL}_N(\beta)$, often also referred to as the *affine Temperley–Lieb algebra*. In [13], starting from the standard modules $\mathsf{W}_{k,x}(N)$ over this algebra, spanned by link states with one marked point and $2k$ defects, we constructed a new family of $\mathcal{E}\mathsf{PTL}_N(\beta)$-modules denoted[1] $\mathsf{X}_{k,\ell,x,y,z}(N)$. These modules are spanned by link states drawn on surfaces shaped like a pair of pants with *two* legs. By projecting these diagrams onto the plane, we draw these link states inside a disc with two marked points (or punctures), which we refer to as the points a and b and respectively draw in green and purple. Figure 1 presents an example of the action of an element of $\mathcal{E}\mathsf{PTL}_{N=12}(\beta)$ on a link state, in the two possible presentations. For generic values of the parameters, we determined in [13] the decomposition of $\mathsf{X}_{k,\ell,x,y,z}$ over the irreducible standard modules. We in fact designed the modules $\mathsf{X}_{k,\ell,x,y,z}$ so that they fit naturally within the correspondence between connectivity operators $\mathcal{O}_{k,x}$ and standard modules $\mathsf{W}_{k,x}$. In this sense, we argued that $\mathsf{X}_{k,\ell,x,y,z}$ is related to the fusion of the standard modules $\mathsf{W}_{k,x}$ and $\mathsf{W}_{\ell,y}$. The standard modules arising in the decomposition of $\mathsf{X}_{k,\ell,x,y,z}$ then correspond to the intermediate states arising in correlation functions in the Operator Product Expansion (OPE) $\mathcal{O}_{k,x} \cdot \mathcal{O}_{\ell,y}$.

The present work continues the investigation of the fusion of representations of $\mathcal{E}\mathsf{PTL}_N(\beta)$. We parameterise the loop weight as $\beta = -q - q^{-1}$ with $q \in \mathbb{C}^\times$, and focus on the values of $q$ that are *not* roots of unity. Below, we describe the new constructions and the new results presented in this paper.

First, we define new modules $\mathsf{Y}_{k,\ell,x,y,[z,w]}$ over $\mathcal{E}\mathsf{PTL}_N(\beta)$, also spanned by link states with two marked points a and b. These new modules have both similarities and differences compared to the modules $\mathsf{X}_{k,\ell,x,y,z}$. First, for both families of modules, $k$ and $\ell$ are integers or half-integers that count the maximal numbers of defects, $2k$ and $2\ell$, that can be attached to the points a and b, respectively. However some rules for the action of the algebra are

---

[1]For simplicity of notation, we often drop the argument $N$ and denote a module $\mathsf{M}(N)$ over $\mathcal{E}\mathsf{PTL}_N(\beta)$ as $\mathsf{M}$.

different for the modules $Y_{k,\ell,x,y,[z,w]}$. In particular, these modules allow for pairs of defects originating from the same marked point to connect together if they encircle the other marked point. Second, for both the modules $X_{k,\ell,x,y,z}$ and $Y_{k,\ell,x,y,[z,w]}$, the parameters $x, y \in \mathbb{C}^{\times}$ parameterise the winding of curves — both defects and closed loops — around the points a and b, respectively. Finally, the parameter $z$ in $X_{k,\ell,x,y,z}$ determines phase factors associated to the winding around b of a defect attached to a, and vice versa. It also parameterises the weight of closed loops encircling both marked points. In contrast, our definition of the module $Y_{k,\ell,x,y,[z,w]}$ includes two separate parameters $z$ and $w$ to parameterise these special windings of defects and the weights of the closed loops around both a and b. As we shall see, this definition is somewhat peculiar, as the presence of these two parameters in the module depends non-trivially on the values of $k$ and $\ell$. In particular, for $N$ odd, the new modules turn out to depend only on $z$, and in this case we denote them simply by $Y_{k,\ell,x,y,z}$. In general, the notation with $[z,w]$ serves as a reminder of the peculiarity of the definition.

For the new modules, we define a gluing operator $g_{[z,w]}$ that takes a link state of $W_{k,x}(N_a)$ and another from $W_{\ell,y}(N_b)$, and outputs a larger link state in $Y_{k,\ell,x,y,[z,w]}(N)$, with two marked points and $N = N_a + N_b$ nodes. The rules defining the module $Y_{k,\ell,x,y,[z,w]}(N)$ then make it a cyclic module with respect to a subset of states obtained from the gluing. We then define the *fusion module* for two standard modules as[2]

$$\left(W_{k,x}(N_a) \times W_{\ell,y}(N_b)\right)_{[z,w]} = Y_{k,\ell,x,y,[z,w]}(N) = g_{[z,w]}\left(W_{k,x}(N_a), W_{\ell,y}(N_b)\right). \tag{1}$$

Second, for generic values of the parameters $[z,w]$,[3] we find the decomposition of $Y_{k,\ell,x,y,[z,w]}$ as a direct sum of irreducible standard modules:

$$N \text{ even:} \qquad Y_{k,\ell,x,y,[z,w]}(N) \simeq \begin{cases} W_{0,w}(N) \oplus \bigoplus_{m=1}^{N/2} \bigoplus_{n=0}^{2m-1} W_{m,e^{i\pi n/m}}(N), & z^{2(k-\ell)} = 1, \\ \bigoplus_{m=1}^{N/2} \bigoplus_{n=0}^{2m-1} W_{m,z^{(k-\ell)/m}e^{i\pi n/m}}(N), & \text{otherwise,} \end{cases} \tag{2a}$$

$$N \text{ odd:} \qquad Y_{k,\ell,x,y,z}(N) \simeq \bigoplus_{m=1/2}^{N/2} \bigoplus_{n=0}^{2m-1} W_{m,z^{(k-\ell)/m}e^{i\pi n/m}}(N). \tag{2b}$$

This is different from the decomposition of the modules $X_{k,\ell,x,y,z}(N)$, in particular because in that case the standard modules $W_{m,t}(N)$ appearing in the decomposition have the constraint $m \geqslant |k - \ell|$.

Let $M_a(N_a)$ and $M_b(N_b)$ be two indecomposable modules over $\mathcal{E}PTL_{N_a}(\beta)$ and $\mathcal{E}PTL_{N_b}(\beta)$ respectively, for which we know how to construct the fusion module $(M_a \times M_b)_{[z,w]}$. Fixing the parities of $N_a$ and $N_b$, we define the fusion rule $M_a \times M_b$ by studying their fusion modules on increasing values of $N_a$ and $N_b$. For $M(N)$ an indecomposable $\mathcal{E}PTL_N(\beta)$-module, we say that we have the fusion rule $M_a \times M_b \to M$ if and only if there exists $[z,w]$ and an integer $N_0$ such that, for all nonnegative integers $N_a$ and $N_b$ satisfying $N_a + N_b \geqslant N_0$, the module $M(N_a + N_b)$ appears as a direct summand in the decomposition of $(M_a(N_a) \times M_b(N_b))_{[z,w]}$. We extend this notation by writing[4] $M_a \times M_b \to \{M_1, M_2, M_3, \ldots\}$, if each $M_j$ is indecomposable and satisfies $M_a \times M_b \to M_j$.

---

[2]In this equation, $g_{[z,w]}$ refers to the gluing between modules as defined in (70), which includes the gluing of link states of the two modules *and* the repeated action of the algebra $\mathcal{E}PTL_N(\beta)$ on these glued states.

[3]We refer to Section 3.3 for the definition of generic and non-generic values of $[z,w]$.

[4]Note that this notation does not mean that $\{M_1, M_2, M_3, \ldots\}$ is the full set of indecomposable modules such that $M_a \times M_b \to M_j$ — it can be any subset.

With these definitions, the decomposition (2) of $Y_{k,\ell,x,y,[z,w]}$ translates into the fusion rule for any pair of irreducible standard modules

$$W_{k,x} \times W_{\ell,y} \rightarrow \begin{cases} \{W_{m,t} \mid m \in \mathbb{Z}_{\geqslant 0},\ m + k + \ell \in \mathbb{Z},\ W_{m,t} \text{ is irreducible}\}, & k \neq \ell, \\ \{W_{m,t} \mid m \in \mathbb{Z}_{\geqslant 0},\ t^{2m} = 1,\ W_{m,t} \text{ is irreducible}\}, & k = \ell. \end{cases} \quad (3)$$

The fact that only irreducible modules $W_{m,t}$ arise in this fusion rule follows from the genericity of $[z,w]$ in (2).

In this paper, we will *not* elucidate the structure of $Y_{k,\ell,x,y,[z,w]}$ for non-generic $[z,w]$ and arbitrary values of $k$ and $\ell$. As a result, we do not currently know the full set of modules M arising in the fusion rule $W_{k,x} \times W_{\ell,y}$ — the above rules only indicate a subset. In the case $k = \ell = 0$ however, we will obtain the structure of $Y_{0,0,x,y,[z,w]}$ for non-generic $[z,w]$, and thus in this case we shall write down the full fusion rule $W_{0,x} \times W_{0,y}$, see Section 5.1. In this case, we find that the fusion rule involves one indecomposable yet reducible module with three composition factors. Based on this, we anticipate that the additional set of modules arising in $W_{k,x} \times W_{\ell,y}$ from the non-generic values of $[z,w]$ will in general involve additional indecomposable yet reducible modules.

These fusion rules for standard modules turn out to be more symmetric than the analogous rules that follow from the decomposition of modules $X_{k,\ell,x,y,z}$ obtained in [13]. We will also argue in this paper that the fusion constructed from the modules $X_{k,\ell,x,y,z}$ is incompatible with associativity. In contrast, the fusion built from the modules $Y_{k,\ell,x,y,[z,w]}$ does not exhibit this incompatibility. We thus claim that, compared to our previous construction using the modules $X_{k,\ell,x,y,z}$, our new prescription for constructing fusion from the lattice using the modules $Y_{k,\ell,x,y,[z,w]}$ is an even better candidate for the general purpose of constructing a consistent algebra of bulk connectivity operators.

Third, we initiate the study of the operator algebra problem for modules other than standard modules. For this, we focus on the irreducible submodules and quotient modules that arise as composition factors in standard modules $W_{\ell,y}(N)$ for *non-generic* values of $y$. These were studied previously by Graham and Lehrer [9]. For $q$ not a root of unity, they found that, for special values of $y$, the standard module $W_{\ell,y}(N)$ is indecomposable yet reducible with two composition factors: a submodule $R_{\ell,y}(N)$ and a complement quotient module $Q_{\ell,y}(N)$. [For the other values of $y$, $W_{\ell,y}(N)$ is an irreducible module]. These special non-generic values of $y$ are of the form

$$y = \varepsilon q^{\sigma m}, \qquad \text{where} \qquad 0 \leqslant \ell < m \leqslant \tfrac{N}{2}, \qquad m, \ell \in \mathbb{Z} + \tfrac{N}{2}, \qquad \sigma, \varepsilon \in \{+1, -1\}, \quad (4)$$

and the module structure is described by the Loewy diagram

$$W_{\ell,\varepsilon q^{\sigma m}}(N) \simeq \begin{bmatrix} Q_{\ell,\varepsilon q^{\sigma m}}(N) \\ \searrow \\ R_{\ell,\varepsilon q^{\sigma m}}(N) \end{bmatrix}. \quad (5)$$

Moreover, the radical submodule is isomorphic to another standard module: $R_{\ell,\varepsilon q^{\sigma m}}(N) \simeq W_{m,\varepsilon q^{\sigma \ell}}(N)$. In this paper, we propose a definition for the fusion modules $(W_{k,x} \times R_{\ell,\varepsilon q^{\sigma m}})_{[z,w]}$ and $(W_{k,x} \times Q_{\ell,\varepsilon q^{\sigma m}})_{[z,w]}$ in terms of the gluing operator $g_{[z,w]}$. Because $R_{\ell,\varepsilon q^{\sigma m}}$ is a submodule of $W_{\ell,\varepsilon q^{\sigma m}}$, it turns out that the definition $(W_{k,x} \times R_{\ell,\varepsilon q^{\sigma m}})_{[z,w]}$ follows directly from the construction of $(W_{k,x} \times W_{\ell,\varepsilon q^{\sigma m}})_{[z,w]}$, namely it is given by the following submodule of $Y_{k,\ell,x,\varepsilon q^{\sigma m},[z,w]}$:

$$(W_{k,x} \times R_{\ell,\varepsilon q^{\sigma m}})_{[z,w]} = g_{[z,w]}(W_{k,x}, R_{\ell,\varepsilon q^{\sigma m}}). \quad (6)$$

The fusion module $(W_{k,x} \times Q_{\ell,\varepsilon q^{\sigma m}})_{[z,w]}$ can be defined in two equivalent ways. On one hand, it can be defined directly in terms of a new representation with certain diagrammatic rules for the action of $\mathcal{E}PTL_N(\beta)$ on a specific subset of link states of $Y_{k,\ell,x,\varepsilon q^{\sigma m},[z,w]}$. On the other hand, it is intimately related to the fusion product $(W_{k,x} \times R_{\ell,\varepsilon q^{\sigma m}})_{[z,w]}$ involving the associated radical. Indeed, let us recall that the submodule $R_{\ell,\varepsilon q^{\sigma m}}$ is the kernel of the Gram bilinear form in $W_{\ell,\varepsilon q^{\sigma m}}$. In the scaling limit, its states become *null vectors* in the corresponding Virasoro modules. Then in the quotient module, these null vectors are quotiented out. The same ideas apply at the level of the lattice, namely the fusion of a quotient and a standard module can be defined as

$$(W_{k,x} \times Q_{\ell,\varepsilon q^{\sigma m}})_{[z,w]} = (W_{k,x} \times W_{\ell,\varepsilon q^{\sigma m}})_{[z,w]} \Big/ (W_{k,x} \times R_{\ell,\varepsilon q^{\sigma m}})_{[z,w]}. \tag{7}$$

This amounts to "suppressing" from $Y_{k,\ell,x,y,[z,w]}$ any vector resulting from the gluing of a diagram of $W_{k,x}$ with a null vector of $W_{\ell,\varepsilon q^{\sigma m}}$. This definition is directly analogous to the definition of fusion in CFT in the case where one of the fused modules has null vectors quotiented out. In CFT, the null-vector conditions then lead to severe restrictions on the fusion rules of primary operators and to differential equations satisfied by the conformal correlation functions. In the present work, we introduce and describe both definitions of the fusion modules $W \times Q$, argue why they are equivalent, and use them to obtain the corresponding lattice fusion rules. The study of lattice correlation functions is however beyond the scope of the present work.

With these definitions, we fix $\ell = 0$ and obtain the decomposition of the fusion modules $(W_{k,x} \times R_{0,\varepsilon q^{\sigma m}})_{[z,w]}$ and $(W_{k,x} \times Q_{0,\varepsilon q^{\sigma m}})_{[z,w]}$. Crucially, the modules $(W_{k,x} \times Q_{0,\varepsilon q^{\sigma m}})_{[z,w]}$ turn out to be non-zero only for certain special values of $[z,w]$. Moreover, they lead to fusion rules that are consistent with those expected for bulk connectivity operators in CFT. The results are easily summarised by writing the corresponding fusion rules for $W_{k,x} \times Q_{0,\varepsilon q^{\sigma m}}$. The simplest case is $m = 1$ with $\varepsilon = -1$, as in this case the quotient module $Q_{0,-q}$ is the vacuum module $V$. For generic values of $x$, we find

$$(W_{k,x} \times V)_{[z,w]} \simeq \begin{cases} W_{k,x}, & k > 0, z = x, \\ W_{0,x}, & k = 0, w = x^{\pm 1}, \\ 0, & \text{otherwise}. \end{cases} \tag{8}$$

Thus the fusion rule reads $W_{k,x} \times V \to \{W_{k,x}\}$, which is indeed consistent with the intuition that the connectivity operator associated to the vacuum module should act as the identity operator. More generally, for $m, k > 0$ and $\sigma, \varepsilon \in \{+1, -1\}$, we obtain the fusion rule

$$W_{k,x} \times Q_{0,\varepsilon q^{\sigma m}} \to \Big\{ W_{k+i-j,-\varepsilon x q^{i+j}} \Big| i, j \in \{-\tfrac{m-1}{2}, -\tfrac{m-3}{2}, \ldots, \tfrac{m-1}{2}\} \Big\}, \tag{9}$$

where on the right side, we use the convenient convention $W_{-k,x} = W_{k,x^{-1}}$ for standard modules with negative defect numbers, allowing this result to be expressed in a compact presentation. We also study separately the case of $k = 0$, for which the resulting fusion rule is slightly different.

Lastly, we also study the fusion $Q_{0,\varepsilon_a q^k} \times Q_{0,\varepsilon_b q^\ell}$ of two quotient modules with no defects. These are defined using similar ideas, namely either diagrammatically or with a definition similar to (7). In this case, the study of the decomposition of the fusion modules $Q \times Q$ requires that we understand the structure of $Y_{0,0,x,y,[z,w]}$ for non-generic values of $[z,w]$. This module is in fact precisely equivalent to the module $X_{0,0,x,y,w}$, whose decomposition was investigated in [13]. In certain circumstances, it involves an indecomposable module that is reducible with three composition factors. Our analysis of the fusion modules $Q \times Q$ leads to the simple

fusion rules[5]

$$Q_{0,\varepsilon_a q^k} \times Q_{0,\varepsilon_b q^\ell} \to \left\{ Q_{0,\varepsilon_{ab} q^m} \,\middle|\, m \in \{|k-\ell|+1, |k-\ell|+3, \dots, k+\ell-1\} \right\}, \qquad \varepsilon_{ab} = -\varepsilon_a \varepsilon_b. \quad (10)$$

Amazingly, these fusion rules have the same $\mathfrak{sl}(2)$ structure as those of degenerate operators $\Phi_{k,1} \times \Phi_{\ell,1}$ in conformal field theory.

The outline of the paper is as follows. In Section 2, we recall the definition of the algebra $\mathcal{E}\mathsf{PTL}_N(\beta)$ and of its standard modules $\mathsf{W}_{k,z}$. We also describe bases for the radical modules $\mathsf{R}_{k,z}$ and the quotient modules $\mathsf{Q}_{k,z}$ in the case where $z$ is non-generic. Section 3 discusses the fusion modules for the product $\mathsf{W} \times \mathsf{W}$ of two standard modules. We recall the definition of the modules $\mathsf{X}_{k,\ell,x,y,z}$ from our previous work and give the definition of the new modules $\mathsf{Y}_{k,\ell,x,y,[z,w]}$. We also present many features of these new modules, in particular their decomposition for the generic values of $[z, w]$ and the resulting fusion rules. The proofs of some of these properties are given in Appendix A. In Section 4, we investigate the fusion products $\mathsf{W} \times \mathsf{R}$ and $\mathsf{W} \times \mathsf{Q}$ in the case where $\mathsf{R}$ and $\mathsf{Q}$ are irreducible factors of a standard module $\mathsf{W}_{0,z}$ without defects. We define their fusion modules and obtain their decompositions as well as the corresponding fusion rules. Some of the arguments and technical calculations are relegated to Appendices B and C. The fusion products $\mathsf{Q} \times \mathsf{Q}$ of two quotient modules with zero defects are investigated in Section 5. The specifics of the decomposition of $\mathsf{Y}_{0,0,x,y,[z,w]}$ needed to obtain the decomposition and fusion rules are described in Appendix D. All the results presented up to this point for the fusion products arise from the new lattice prescription of fusion with the modules $\mathsf{Y}_{k,\ell,x,y,[z,w]}$. Section 6 then presents the CFT predictions for the fusion rules of the different modules in the continuum scaling limit, as modules over the tensor product $\mathsf{Vir} \otimes \overline{\mathsf{Vir}}$ of two copies of the Virasoro algebra. A comparison between the two reveals that all modules arising in the lattice fusion rule are predicted by the fusion rules of degenerate operators in CFT. Concluding comments are presented in Section 7.

## 2 The enlarged periodic Temperley–Lieb algebra and its modules

The enlarged periodic Temperley–Lieb algebra, also called the affine Temperley–Lieb algebra, was first introduced in the context of the spin-$\frac{1}{2}$ XXZ chain and the Potts model [7]. Its representation theory was subsequently investigated [8–11], revealing a wide range of indecomposable modules. In this section, we review the definitions of this algebra, of its standard modules, and of some of its useful elements. We also study in greater detail the special cases where the standard modules are indecomposable yet reducible with two composition factors, and give explicit bases for both the radical submodules and the quotient modules.

### 2.1 Definition of the algebra

The enlarged periodic Temperley–Lieb algebra $\mathcal{E}\mathsf{PTL}_N(\beta)$ is a unital associative algebra that has two equivalent presentations. The first is formulated in terms of generators and their relations, and the second is described in terms of families of diagrams satisfying certains rules for their multiplication. Let $N$ be an integer larger than 2. In terms of the generators, the algebra is defined as

$$\mathcal{E}\mathsf{PTL}_N(\beta) = \langle \Omega, \Omega^{-1}, e_1, e_2, \dots, e_N \rangle, \quad (11)$$

---

[5]In contrast with the above results for $\mathsf{W}_{k,x} \times \mathsf{W}_{\ell,y}$ and $\mathsf{W}_{k,x} \times \mathsf{Q}_{0,\varepsilon q^m}$, here all possible values of $[z, w]$ are considered. Thus the right-hand side of (10) gives the full set of indecomposable modules arising in the fusion $\mathsf{Q}_{0,\varepsilon_a q^k} \times \mathsf{Q}_{0,\varepsilon_b q^\ell}$.

with the relations

$$e_j^2 = \beta \, e_j \,, \qquad e_j \, e_{j\pm1} \, e_j = e_j \,, \qquad e_i \, e_j = e_j \, e_i \,, \qquad \text{for } |i-j| > 1 \,, \tag{12a}$$

$$\Omega \, e_j \, \Omega^{-1} = e_{j-1} \,, \qquad \Omega \, \Omega^{-1} = \Omega^{-1} \, \Omega = 1 \,, \qquad e_{N-1} e_{N-2} \cdots e_2 e_1 = \Omega^2 e_1 \,, \tag{12b}$$

where $1$ is the identity, $\beta$ is a free complex parameter, and the indices $i, j$ are taken modulo $N$.

For the diagrammatic definition, the generators are assigned *connectivities* as follows:

$$\Omega = \quad , \qquad \Omega^{-1} = \quad , \qquad 1 = \quad , \tag{13a}$$

$$e_j = \quad , \qquad \text{for } 1 \leqslant j \leqslant N-1 \,, \text{and} \qquad e_N = \quad . \tag{13b}$$

A connectivity is therefore a diagram drawn inside an annulus with $N$ nodes drawn on the inner perimeter and $N$ more nodes on the outer perimeter, where non-intersecting curves connect these nodes pairwise. On both the inner and outer perimeters, we label the nodes from 1 to $N$ in the counter-clockwise direction, and draw a dashed line tying the midpoints between the nodes 1 and $N$ on each of the perimeters.

The product $c_1 c_2$ of two diagrammatic diagrams is obtained by drawing $c_2$ inside $c_1$ and reading the new connectivity from the outer and inner perimeters. If contractible loops are created in the process, each one is removed and replaced by a multiplicative factor of $\beta$. Here are two examples of this product:

$$e_1 e_2 = \quad = \quad , \qquad (e_2)^2 = \quad = \beta \quad = \beta e_2 \,. \tag{14}$$

It is easy to show that the relations (12) are satisfied by this diagrammatic product. Moreover, one can obtain any connectivity with products of the generators. There are in fact an infinite number of connectivities. Indeed, the loop segments may wind around the annulus an arbitrary number of times. Equivalently, the words $\Omega^k$ with $k \in \mathbb{Z}$ can never be reduced to a shorter word. For $N$ even, the connectivities may also have non-contractible loops, namely loops that encircle the inner perimeter, as in this example:

$$(e_1 e_3 \cdots e_{N-1})(e_2 e_4 \cdots e_N) = \quad . \tag{15}$$

The $k$-th power of this word, with $k \in \mathbb{N}$, can never be simplified to a shorter word. In other words, the relations (12) do not allow one to remove non-contractible loops. In its diagrammatic definition, the algebra $\mathcal{E}\mathsf{PTL}_N(\beta)$ is the linear span of the connectivities, endowed with the above rule for their product.

The two definitions are equivalent for $N > 2$. Moreover, the diagrammatic definition is valid for $N = 1$ and $N = 2$ as well, which we take as the definition of $\mathcal{E}\mathsf{PTL}_N(\beta)$ in these cases. (Some of the relations in (12) then need to be modified for the two definitions to be equivalent.)

The parameter $\beta$ is referred to as the *loop weight*. We parameterise it as

$$\beta = -q - q^{-1}, \qquad q \in \mathbb{C}^{\times}. \tag{16}$$

Our focus in this paper is on the case where $q$ is not a root of unity.

Finally, we recall that the ordinary Temperley–Lieb algebra $\mathsf{TL}_N(\beta)$ is the subalgebra of $\mathcal{E}\mathsf{PTL}_N(\beta)$ defined as

$$\mathsf{TL}_N(\beta) = \langle e_1, e_2, \dots, e_{N-1} \rangle. \tag{17}$$

In terms of the diagrams, it is spanned by the connectivities that have no loop segment crossing the dashed line. By cutting along this dashed line, one obtains a circular strip that can be deformed into a rectangular box with $N$ nodes on its top and top segments, as is usual for depicting connectivities of $\mathsf{TL}_N(\beta)$.

## 2.2   The standard modules $\mathsf{W}_{k,z}(N)$

A *link state* for a standard module $\mathsf{W}_{k,z}(N)$ is a diagram drawn on a disc with $N$ nodes on its perimeter and a marked point in its interior. We assign to the nodes the labels $1, 2, \dots, N$ in the counter-clockwise direction. We also draw a dashed line between the marked point and the midpoint on the perimeter between the nodes $N$ and $1$. The diagram then consists of a collection of loop segments that either connect the nodes pairwise or tie them to the marked point in what is called a *defect*. Moreover, the defects are drawn in such a way that they do not cross the dashed line. Two link states are then considered to be identical if their nodes are connected in the same way and it is possible to deform the loop segments of the first state into those of the second state without passing over the marked point. The standard module $\mathsf{W}_{k,z}(N)$ of $\mathcal{E}\mathsf{PTL}_N(\beta)$ is built on the vector space spanned by the set $\mathsf{B}_k(N)$ of link states with $2k$ defects, with $k \in \frac{1}{2}\mathbb{Z}_{\geqslant 0}$ and $\frac{N}{2} - k \in \mathbb{Z}_{\geqslant 0}$. The set $\mathsf{B}_k(N)$ has the cardinality

$$|\mathsf{B}_k(N)| = \binom{N}{\frac{N-2k}{2}}, \tag{18}$$

which is also the dimension of the module $\mathsf{W}_{k,z}(N)$. Here are the sets of link states for $N = 4$:

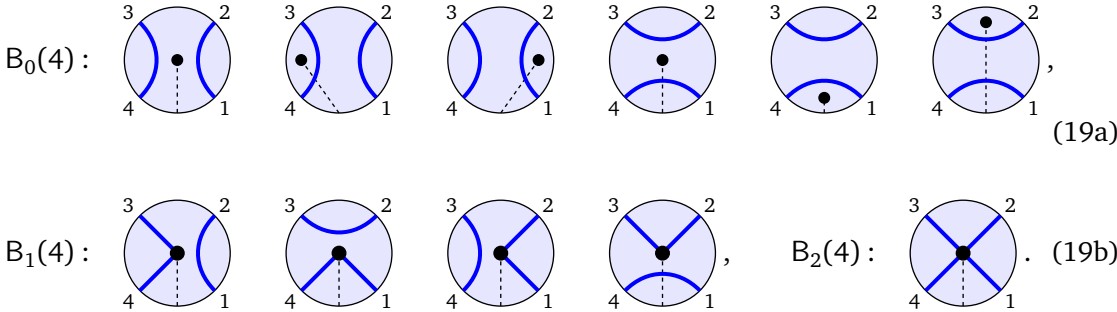

$$\tag{19a}$$

$$\tag{19b}$$

The standard action $c \cdot v$ of a connectivity $c \in \mathcal{E}\mathsf{PTL}_N(\beta)$ on the a link state $v \in \mathsf{W}_{k,z}(N)$ is obtained by drawing $v$ inside $c$. One then reads off the new link state on the outer perimeter. Any contractible loop formed in the process is erased and replaced by a multiplicative factor of $\beta$. For $k = 0$, there may also be non-contractible loops, namely loops that encircle the marked point. These are removed and replaced by the weight $\alpha = z + z^{-1}$. For $k > 0$, there are two extra rules. First, if two defects are connected, the result is automatically set to zero. Second,

there are extra weights $z$ and $z^{-1}$ that measure the winding of the defects around the annulus and arise each time a defect crosses the dashed line. If a walker traveling on a defect from the marked point to the perimeter crosses the dashed line, the result is multiplied by $z$ if the marked point is on the dashed line to its right, and $z^{-1}$ if it is to its left. This defects is then *unwound*, namely it is once again drawn in such a way that it does not cross the dashed line. Here are examples of the standard action:

$$e_1 \cdot \left( \begin{array}{c} 3 \\ 4 \end{array} \right) = \left( \begin{array}{c} 3 \\ 4 \end{array} \right) = \alpha \left( \begin{array}{c} 3 \\ 4 \end{array} \right), \quad e_4 \cdot \left( \begin{array}{c} 3 \\ 4 \end{array} \right) = \left( \begin{array}{c} 3 \\ 4 \end{array} \right) = z^{-1} \left( \begin{array}{c} 3 \\ 4 \end{array} \right). \quad (20)$$

Here is a summary of the local diagrammatic rules defining the standard action:

$$\left( \bullet \, \bigcirc \right) = \beta \left( \bullet \right), \qquad \left( \bullet \right) = \alpha \left( \bullet \right), \qquad (21a)$$

$$\left( \bullet \right) = z \left( \bullet \right), \qquad \left( \bullet \right) = z^{-1} \left( \bullet \right), \qquad \left( \bullet \right) = 0. \qquad (21b)$$

This action $c \cdot v$ is then extended linearly to all $c \in \mathcal{E}\mathsf{PTL}_N(\beta)$ and all $v \in \mathsf{W}_{k,z}(N)$.

The structure of the standard modules is known from the work of Graham and Lehrer [9]. First, $\mathsf{W}_{k,z}(N)$ is always indecomposable. Moreover, for all $q \in \mathbb{C}^\times$, there exist homomorphisms

$$\mathsf{W}_{\ell,\varepsilon q^{\sigma k}}(N) \to \mathsf{W}_{k,\varepsilon q^{\sigma \ell}}(N) \quad \text{for} \quad 0 \leqslant k < \ell \leqslant \tfrac{N}{2}, \qquad k, \ell \in \mathbb{Z} + \tfrac{N}{2}, \qquad \sigma, \varepsilon \in \{+1, -1\}. \quad (22)$$

These are the only non-trivial homomorphisms. This allows one to work out the module structure of $\mathsf{W}_{k,z}(N)$. If there exists no homormorphism of the form (22) into $\mathsf{W}_{k,z}(N)$, then it is irreducible. We refer to the corresponding values of $z$ as *generic*. In contrast, the values of $z$ of the form $\varepsilon q^{\sigma \ell}$ where there exists a non-trivial homomorphism into $\mathsf{W}_{k,z}(N)$ are called *non-generic*.

Let us describe the structure of the standard modules for $q$ not a root of unity and $z$ non-generic. In this case, the module $\mathsf{W}_{k,z}(N)$ is reducible and has two composition factors. Its structure is described by its Loewy diagram:

$$\mathsf{W}_{k,\varepsilon q^{\sigma \ell}}(N) \simeq \left[ \begin{array}{c} \mathsf{Q}_{k,\varepsilon q^{\sigma \ell}}(N) \\ \searrow \\ \mathsf{R}_{k,\varepsilon q^{\sigma \ell}}(N) \end{array} \right] \simeq \left[ \begin{array}{c} \mathsf{I}_{k,\varepsilon q^{\sigma \ell}}(N) \\ \searrow \\ \mathsf{I}_{\ell,\varepsilon q^{\sigma k}}(N) \end{array} \right]. \quad (23)$$

Here, $\mathsf{R}_{k,z}(N)$ and $\mathsf{Q}_{k,z}(N)$ are the *radical submodule* and the *quotient module* of $\mathsf{W}_{k,z}(N)$. For $q$ not a root of unity and $z$ non-generic, the radical $\mathsf{R}_{k,z}(N)$ is nonzero and both $\mathsf{R}_{k,z}(N)$ and $\mathsf{Q}_{k,z}(N)$ are irreducible modules. Moreover, we have the isomorphisms $\mathsf{R}_{k,\varepsilon q^{\sigma \ell}}(N) \simeq \mathsf{Q}_{\ell,\varepsilon q^{\sigma k}}(N) \simeq \mathsf{W}_{\ell,\varepsilon q^{\sigma k}}(N)$. Indeed, the value $z = \varepsilon q^{\sigma k}$ is generic for $\mathsf{W}_{\ell,z}(N)$ because $k < \ell$, so this module is irreducible. It is common to denote the modules $\mathsf{R}_{k,\varepsilon q^{\sigma \ell}}(N)$ and $\mathsf{Q}_{\ell,\varepsilon q^{\sigma k}}(N)$ by $\mathsf{I}_{\ell,\varepsilon q^{\sigma k}}(N)$, and likewise $\mathsf{Q}_{k,\varepsilon q^{\sigma \ell}}(N)$ by $\mathsf{I}_{k,\varepsilon q^{\sigma \ell}}(N)$, as in the right side of (23), to insist on their irreducibility and underline which modules are isomorphic. Here we keep a distinct notation for the radical and quotient modules, as we will find that they behave differently under fusion in Section 4.

## 2.3 Useful elements of the algebra

In this section, we recall the definition of certain useful elements of $\mathcal{E}\mathsf{PTL}_N(\beta)$: the Jones-Wenzl projectors $P_n$, the braid transfer matrices $\boldsymbol{F}$ and $\bar{\boldsymbol{F}}$, and the central element $\Omega^N$.

**Jones-Wenzl projectors.**   The Jones-Wenzl projectors $P_1, P_2, \ldots, P_N$ are elements of the ordinary Temperley–Lieb algebra $\mathsf{TL}_N(\beta) \subset \mathcal{E}\mathsf{PTL}_N(\beta)$ defined recursively as [17–19]

$$P_1 = 1, \qquad P_{n+1} = P_n + \frac{[n]}{[n+1]} P_n e_n P_n, \qquad \text{where} \qquad [k] = \frac{q^k - q^{-k}}{q - q^{-1}}. \tag{24}$$

With this definition, the loop weight is $\beta = -[2]$. We draw the Jones-Wenzl projectors as pink rectangles:

$$P_n = \boxed{\phantom{xx} n \phantom{xx}} \, . \tag{25}$$

They satisfy the relations

$$(P_n)^2 = P_n, \quad P_n e_j = e_j P_n = 0 \quad \text{for} \quad j = 1, 2, \ldots, n-1, \qquad e_n P_n e_n = -\frac{[n+1]}{[n]} P_{n-1} e_n. \tag{26}$$

One can also show that $P_n$ is invariant under both left-right reflections and vertical flips. As an element of $\mathcal{E}\mathsf{PTL}_N(\beta)$ acting on $\mathsf{W}_{k,z}(N)$, the projector $P_N$ vanishes on all but one link state: the unique link state $v_k(N) \in \mathsf{B}_k(N)$ with all of its $(N-2k)$ arcs crossing the dashed line — see (45) for the definition of $v_k(N)$. The Jones-Wenzl projectors also satisfy the relations

$$P_n = (1_1 \otimes P_{n-1}) \left( 1_n + \sum_{j=1}^{n-1} \frac{[n-j]}{[n]} e_1 e_2 \cdots e_j \right), \tag{27a}$$

$$P_n = (P_{n-1} \otimes 1_1) \left( 1_n + \sum_{j=1}^{n-1} \frac{[j]}{[n]} e_{n-1} e_{n-2} \cdots e_j \right), \tag{27b}$$

$$P_n = \left( 1_n + \sum_{j=1}^{n-1} \frac{[n-j]}{[n]} e_j e_{j-1} \cdots e_1 \right) (1_1 \otimes P_{n-1}), \tag{27c}$$

$$P_n = \left( 1_n + \sum_{j=1}^{n-1} \frac{[j]}{[n]} e_j e_{j+1} \cdots e_{n-1} \right) (P_{n-1} \otimes 1_1). \tag{27d}$$

Here $1_m$ denotes the identity on $m$ strands in $\mathsf{TL}_m(\beta) \subseteq \mathsf{TL}_N(\beta)$, with $1 \leqslant m \leqslant N$. Moreover, $c_1 \otimes c_2$ denotes the element in the subalgebra $\mathsf{TL}_n(\beta) \otimes \mathsf{TL}_{N-n}(\beta)$ of $\mathsf{TL}_N(\beta)$, with $1 \leqslant n \leqslant N-1$, where $c_1 \in \mathsf{TL}_n(\beta)$ and $c_2 \in \mathsf{TL}_{N-n}(\beta)$ are drawn side-by-side in the rectangular diagram.

**The central elements $F$ and $\bar{F}$.**   The braid transfer matrices $F$ and $\bar{F}$ are the elements of $\mathcal{E}\mathsf{PTL}_N(\beta)$ defined as

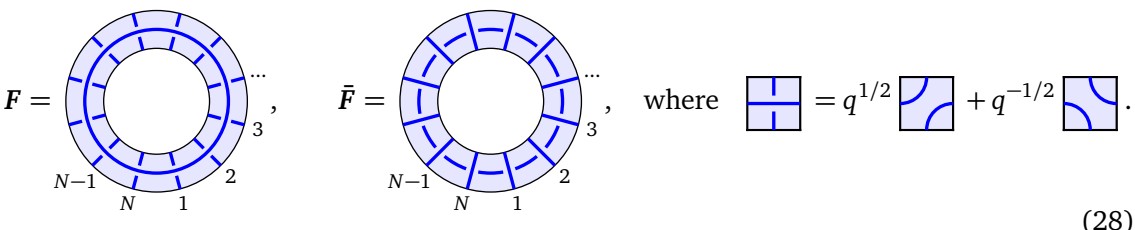

$$(28)$$

It follows [20] from the push-through property

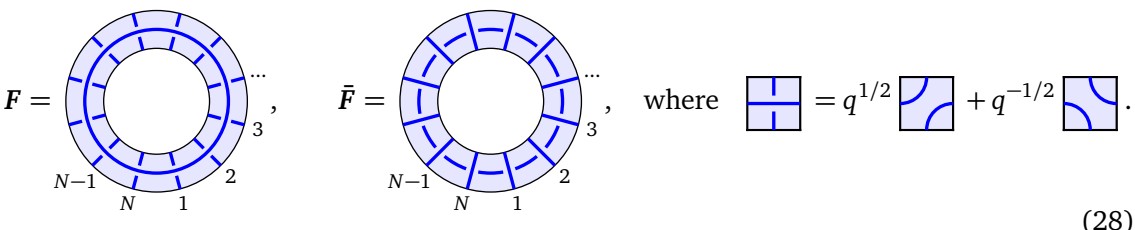

that these elements are in the center of $\mathcal{E}\mathsf{PTL}_N(\beta)$. On $\mathsf{W}_{k,z}(N)$, both $F$ and $\bar{F}$ act as multiples of the identity, namely

$$F \cdot v = \left( zq^k + \frac{1}{zq^k} \right) v, \qquad \bar{F} \cdot v = \left( \frac{z}{q^k} + \frac{q^k}{z} \right) v, \qquad \forall v \in \mathsf{W}_{k,z}(N). \tag{30}$$

**The central element $\Omega^N$.** The element $\Omega^N$ is also in the center of $\mathcal{E}\mathrm{PTL}_N(\beta)$. It acts as a multiple of the identity on the standard modules:

$$\Omega^N \cdot v = z^{2k} v, \qquad \forall v \in \mathsf{W}_{k,z}(N). \tag{31}$$

## 2.4 Bases for the quotient and the radical

In this section, we study the composition factors of $\mathsf{W}_{k,z}(N)$ in the case where $q$ is not a root of unity and $z$ is non-generic. We describe bases for these modules in terms of link states. The construction of a basis for the radical submodules was already given in [13], using the Jones-Wenzl projectors and the *insertion algorithm*. The construction for the quotient presented here is new. Before describing the general construction, we give three examples, each of them for $N = 4$. In these examples, we consider for simplicity the case where the twist parameter is set to $z = \varepsilon q^\ell$ with $\ell > 0$. The case $z = \varepsilon q^{-\ell}$ is easily obtained by changing $q$ to $q^{-1}$.

For the construction, it is useful to define an extra property of link states: their *crossing number*. For $v \in \mathsf{B}_k(N)$, its crossing number $r$ is the number of loop segments that intersect the dashed line tying the marked point to the outer perimeter. For instance, the first three states of $\mathsf{B}_1(4)$ in (19) have crossing numbers $r = 0$ whereas the last one has $r = 1$.

**Example 1: Bases for $\mathsf{R}_{0,\varepsilon q}(4)$ and $\mathsf{Q}_{0,\varepsilon q}(4)$.** For $z = \varepsilon q$, the weight of the non-contractible loops is $\alpha = -\varepsilon\beta$. A basis of the radical is

$$\mathsf{R}_{0,\varepsilon q}(4): \tag{32}$$

where the thick pink arc is the projector $P_2$. It was shown in [13] that the four-dimensional subspace of $\mathsf{W}_{0,\varepsilon q}(4)$ spanned by these states is indeed closed under the action of $\mathcal{E}\mathrm{PTL}_N(\beta)$. In this construction, the *inserted state* is

$$\cdots = 2 \,\cdots\, 1 - \frac{\alpha}{\beta} \, 2 \,\cdots\, 1 \, \Big|_{\alpha = -\varepsilon\beta} = 2 \,\cdots\, 1 + \varepsilon \, 2 \,\cdots\, 1, \tag{33}$$

and it is inserted on all four link states of $\mathsf{W}_{1,\varepsilon}(4) \simeq \mathsf{R}_{0,\varepsilon q}(4)$.

A basis for the quotient is

$$\mathsf{Q}_{0,\varepsilon q}(4): \tag{34}$$

namely it consists of the subset of link states of $\mathsf{B}_0(4)$ with crossing number $r = 0$. The action of $\mathcal{E}\mathrm{PTL}_N(\beta)$ on this basis includes an extra quotient relation, obtained by setting the right side of (33) to zero, namely

$$2 \,\cdots\, 1 \equiv -\varepsilon \, 2 \,\cdots\, 1. \tag{35}$$

As a result, if the action of $\mathcal{E}\mathrm{PTL}_N(\beta)$ creates a loop segment that crosses the dashed line, then one can move it across the marked point, at the cost of a sign $-\varepsilon$. Thus in the resulting quotient module, depending on the value of $\varepsilon$, the marked point either "commutes" or "anticommutes" with the loop segments.

**Example 2: Bases for $R_{0,\varepsilon q^2}(4)$ and $Q_{0,\varepsilon q^2}(4)$.** In this case, the weight of the non-contractible loops is

$$\alpha = z + z^{-1} = \varepsilon(q^2 + q^{-2}) = \varepsilon(\beta^2 - 2). \tag{36}$$

The radical is one-dimensional, with its unique state constructed from the projector $P_4$ as

$$R_{0,\varepsilon q^2}(4): \qquad \qquad . \tag{37}$$

From the explicit form of the projector, we find that this state reads

$$
\begin{aligned}
&= \quad + \frac{\alpha}{\beta^2 - 2} \quad - \frac{\alpha\beta}{\beta^2 - 2} \quad + \frac{\alpha}{\beta^2 - 2} \\
&\quad - \frac{\beta(\alpha^2 + \beta^2 - 2)}{(\beta^2 - 2)(\beta^2 - 1)} \quad + \frac{\alpha^2 + \beta^2 - 2}{(\beta^2 - 2)(\beta^2 - 1)} \Big|_{\alpha = \varepsilon(\beta^2 - 2)} \\
&= \quad + \varepsilon \quad - \varepsilon\beta \quad + \varepsilon \quad - \beta \quad + \quad .
\end{aligned}
\tag{38}
$$

A basis of the quotient module is

$$Q_{0,\varepsilon q^2}(4): \qquad \qquad . \tag{39}$$

It therefore consists of all the link states in $B_0(4)$ that have crossing numbers $r \in \{0, 1\}$. The quotient relation is obtained by setting (38) to zero, namely

$$
\equiv -\varepsilon \quad + \varepsilon\beta \quad - \varepsilon \quad + \beta \quad - \quad . \tag{40}
$$

Thus, if the action of $\mathcal{E}\mathsf{PTL}_N(\beta)$ on a state of the basis (39) produces the only link state in $B_0(4)$ with $r = 2$, this state is replaced by the linear combination of states with $r < 2$ given in the right-hand side of (40).

**Example 3: Bases for $R_{1,\varepsilon q^2}(4)$ and $Q_{1,\varepsilon q^2}(4)$.** The radical is also one-dimensional in this example:

$$R_{1,\varepsilon q^2}(4): \qquad \qquad . \tag{41}$$

Expanding the projector, we find

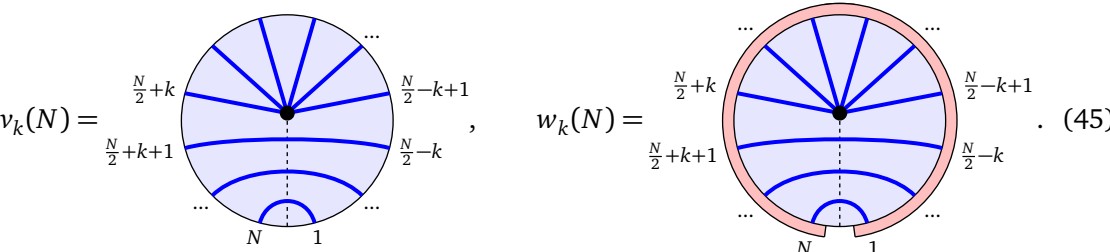

$$\left.
\begin{array}{c}
\end{array}
\right|_{z=\varepsilon q^2}$$

(42)

The basis for the quotient module is

$$Q_{1,\varepsilon q^2}(4):$$

(43)

It consists of the subset of link states of $B_1(4)$ with $r = 0$. The quotient relation is obtained by setting the last line of (42) to zero:

(44)

Thus if the action of $\mathcal{E}\mathrm{PTL}_N(\beta)$ creates the only link state in $B_1(4)$ with $r = 1$, it is replaced by this linear combination of states with $r = 0$.

**General construction: Bases for $R_{k,\varepsilon q^{\sigma\ell}}(N)$ and $Q_{k,\varepsilon q^{\sigma\ell}}(N)$.** In the general case, to construct the bases of the radical and quotient, we consider the states $v_k(N)$ and $w_k(N)$ in $W_{k,z}(N)$ defined as

$$v_k(N) = \qquad , \qquad w_k(N) = \qquad .$$

(45)

To construct a basis of $R_{k,\varepsilon q^{\sigma\ell}}(N)$, one applies the insertion algorithm to $B_\ell(N)$. For each link state $v$ in $B_\ell(N)$, we map it to a linear combination of link states of $B_k(N)$, by replacing the $2\ell$ defects of $v$ by the linear combination $w_k(2\ell)$. As shown in [13], for $z = \varepsilon q^{\sigma\ell}$, the action of $\mathcal{E}\mathrm{PTL}_N(\beta)$ on this basis is invariant, so this indeed produces a submodule of $W_{k,\varepsilon q^{\sigma\ell}}(N)$.

For $Q_{k,\varepsilon q^{\sigma\ell}}(N)$, we define its basis as spanned by the subset of link states of $B_k(N)$ with $r < \ell - k$. The action of $\mathcal{E}\mathrm{PTL}_N(\beta)$ on this basis includes an extra quotient relation, which is an equation on the inserted state: $w_k(2\ell) \equiv 0$. This relation can be written in terms of the complementary projector $1 - P_{2\ell}$ as $v_k(2\ell) \equiv (1 - P_{2\ell})v_k(2\ell)$, making it clear that the state $v_k(2\ell)$ whose crossing number is $r = \ell - k$ is mapped to a linear combination of states with $r < \ell - k$. The quotient relation allows us to rewrite any link state with crossing number $r \geqslant \ell - k$ as a linear combination of states in the basis.

The dimensions of these representations are

$$\dim \mathsf{R}_{k,\varepsilon q^{\sigma \ell}}(N) = |\mathsf{B}_\ell(N)| = \binom{N}{\frac{N-\ell}{2}}, \tag{46a}$$

$$\dim \mathsf{Q}_{k,\varepsilon q^{\sigma \ell}}(N) = |\mathsf{B}_k(N)| - |\mathsf{B}_\ell(N)| = \binom{N}{\frac{N-k}{2}} - \binom{N}{\frac{N-\ell}{2}}. \tag{46b}$$

This is clear from the homomorphisms (22) and the Loewy diagram (23), and can also be verified in terms of the cardinalities of the bases defined above. Interestingly, for the special case $\ell = k + 1$, the dimension of $\mathsf{Q}_{k,\varepsilon q^{\sigma(k+1)}}(N)$ is precisely equal to the dimension of the standard module $\mathsf{V}_k(N)$ of the usual Temperley–Lieb algebra with $2k$ defects. This generalises the same property already known for the vacuum module $\mathsf{V}(N)$ corresponding to $k = 0$.

## 3   The fusion modules of $\mathsf{W}_{k,x} \times \mathsf{W}_{\ell,y}$

In this section, we first review the definition of the modules $\mathsf{X}_{k,\ell,x,y,z}(N)$ that we defined previously in [13]. We then define the new modules $\mathsf{Y}_{k,\ell,x,y,[z,w]}(N)$ and describe their various properties.

### 3.1   The modules $\mathsf{X}_{k,\ell,x,y,z}(N)$

In [13], we defined a family of modules $\mathsf{X}_{k,\ell,x,y,z}(N)$ that we argued were suitable candidates for the fusion of two standard modules. In this section, we review the definition of these modules. Their vector space is spanned by link states drawn on a disc with $N$ nodes and with two marked points a and b. Here are three examples of such link states, respectively arising in $\mathsf{X}_{0,0,x,y,z}(12)$, $\mathsf{X}_{1,0,x,y,z}(12)$ and $\mathsf{X}_{\frac{3}{2},\frac{1}{2},x,y,z}(12)$:

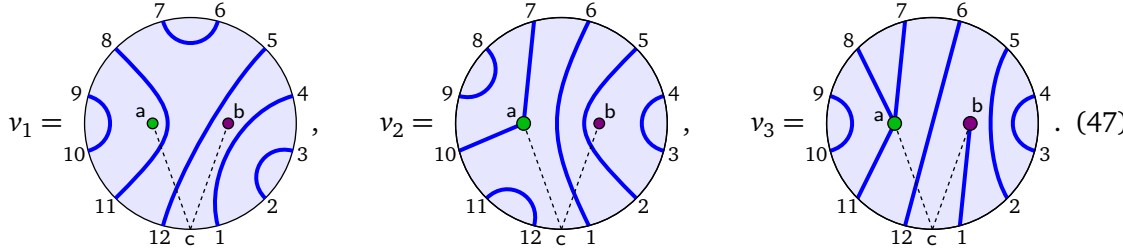

$$\tag{47}$$

The modules $\mathsf{X}_{k,\ell,x,y,z}(N)$ are defined for $N \geqslant 2$ with $k,\ell \in \frac{1}{2}\mathbb{Z}_{\geqslant 0}$ satisfying $\frac{N}{2} - k - \ell \in \mathbb{Z}_{\geqslant 0}$. A link state for $\mathsf{X}_{k,\ell,x,y,z}(N)$ is a diagram drawn on a disc with $N$ marked nodes, with non-intersecting curves connecting the points pairwise or tying them by defects to the points a and b. In these diagrams, we also label as c the midpoint on the outer perimeter between $N$ and 1, and draw the dashed segments ac and bc. The defects are always drawn in such a way as to not intersect the segments ac and bc.

One key property of these link states is their *depth p*, which measures the minimal number of loop segments that one must cross to travel from a to b. In the above examples, the depth is $p = 2$ for $v_1$ and $v_2$, and $p = 1$ for $v_3$.

In general, the module $\mathsf{X}_{k,\ell,x,y,z}(N)$ is constructed on sets of link states with a number $2r$ of defects that varies, with $r$ taking the values $|k-\ell|, |k-\ell|+1, \ldots, k+\ell$. Of these, $(r+k-\ell)$ are attached to the point a and $(r+\ell-k)$ are attached to the point b. Thus $k$ and $\ell$ count the maximal number of defects that can be attached to a and b respectively. If either $k = 0$ or $\ell = 0$, then $r$ only takes one value, and in this case one marked point has zero defects whereas the other has a fixed number of defects. If both $k$ and $\ell$ are non-zero, the number $r$ takes

more than one value, and thus the states do not all have the same total number of defects. In this last case, the basis of $X_{k,\ell,x,y,z}(N)$, as defined in [13], is such that the only link states with non-zero depth are those which have exactly $2k + 2\ell$ defects. Finally, one last feature of the basis regards link states of depth $p = 0$ in the case where one of the two marked points has no defects. Let us suppose that the marked point without defects is b. In this case, the only such link states that are part of the basis are those where b is positioned in such a way that the dashed segment bc does not cross a defect of a. The same principle applies if it is instead a that has no defects. To illustrate, here are the link states of $X_{1,0,x,y,z}(N)$:

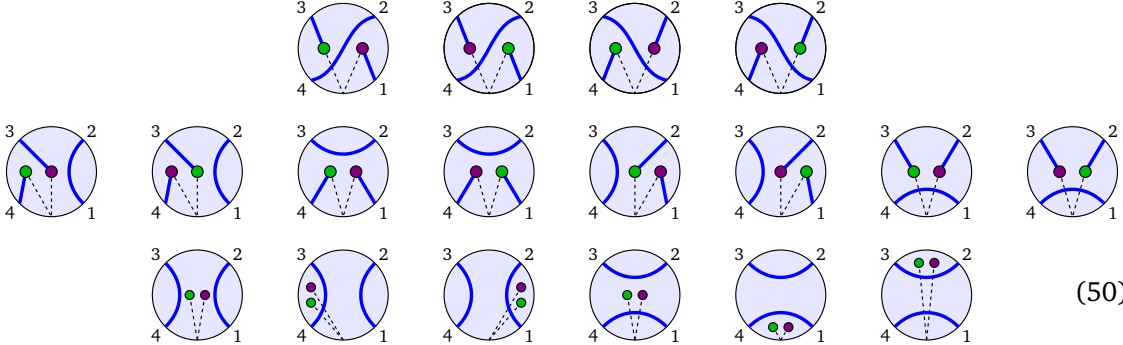

$$(48)$$

Because of the last rule, the state

$$v = \qquad (49)$$

is not an element of the basis of $X_{1,0,x,y,z}(4)$. As a second example, the basis states of $X_{\frac{1}{2},\frac{1}{2},x,y,z}(4)$ are

$$(50)$$

The dimension of $X_{k,\ell,x,y,z}(N)$ is

$$\dim X_{k,\ell,x,y,z}(N) = |B_{|k-\ell|}(N)| + \sum_{m=|k-\ell|+1}^{N/2} 2m\,|B_m(N)| = (\tfrac{N}{2} - |k-\ell| + 1)\binom{N}{\tfrac{N}{2} - |k-\ell|}. \quad (51)$$

We now describe the action of the algebra on this basis. First, the parameters $x$ and $y$ play a similar role for the action $c \cdot v$ in these modules, around the points a and b respectively, as they do in the standard modules $W_{k,x}$ and $W_{\ell,y}$. Indeed, depending on whether $k = 0$ or $k > 0$, the parameter $x$ either parameterises the weight $\alpha_a = x + x^{-1}$ of the loops encircling the point a, or it plays the role of a twist factor for the defects of a when they cross the dashed line ac. The same holds for the parameter $y$ for loops and defect windings across bc, with $\alpha_b = y + y^{-1}$. Moreover, if two defects of a given marked point are connected by the action of $\mathcal{E}PTL_N(\beta)$, the result is set to zero. The parameter $z$ is an extra variable that couples the two standard modules in their fusion, namely it either parameterises the weight $\alpha_{ab}$ of the loops encircling both marked points, or it couples to the winding of the defects of a across the line bc and the defects of b across the line ac. In the case $k, \ell > 0$, the total number of defects may decrease if defects of a connect to those of b. There is then a non-trivial rule involving a weight $\mu$ that

arises when two of these defects connect, and that depends on the respective positions of the dashed lines ac and bc. Here is an overview of the diagrammatic rules for $X_{k,\ell,x,y,z}(N)$:

$$\text{(diagrammatic rules)} \tag{52}$$

with

$$\alpha_{\mathrm{a}} = x + x^{-1}, \qquad \alpha_{\mathrm{b}} = y + y^{-1}, \qquad \alpha_{\mathrm{ab}} = z + z^{-1}, \qquad \mu = \frac{yz}{x}. \tag{53}$$

More generally, the twist factor assigned to a defect that winds around the marked points and possibly crosses the segments ac and bc multiple times is obtained as the product of the weights of the individual crossings. For example,

$$\text{(diagram)} = \frac{z}{x}\,\text{(diagram)}. \tag{54}$$

Moreover, with the above diagrammatic rules, the state $v$ defined in (49) evaluates to

$$v = \frac{z}{x}\,\text{(diagram)} \tag{55}$$

and is thus proportional to a state in the basis (48).

In [13], we argued that this diagrammatic action indeed produces a representation of $\mathcal{E}\mathsf{PTL}_N(\beta)$. We also obtained the decomposition

$$X_{k,\ell,x,y,z}(N) \simeq W_{k-\ell,z}(N) \oplus \bigoplus_{m=|k-\ell|+1}^{N/2} \bigoplus_{n=0}^{2m-1} W_{m,z^{(k-\ell)/m}e^{i\pi n/m}}(N), \tag{56}$$

for generic values of $z$. Here and below, we use the convention $W_{-m,z} = W_{m,z^{-1}}$ to define standard modules with negative defect numbers. In this way, the first term in the right side of (56) is well defined for $k < \ell$.

## 3.2 Definition of the modules $Y_{k,\ell,x,y,[z,w]}(N)$

Some of the diagrammatic rules defining the modules $X_{k,\ell,x,y,z}$ are not the most convenient for the study of fusion of connectivity operators. First, some of the rules in (52) allow one to displace a marked point across a defect (with some weights), which is not what one typically expects when computing correlation functions. Second, the link states of $X_{k,\ell,x,y,z}$ have at least $|k-\ell|$ defects, and the action of the algebra cannot reduce this number further. As we discuss at the end of Section 4.3, this turns out to be an issue for the associativity of fusion products. Finally, it is convenient to define modules where the weight $\alpha_{\mathrm{ab}}$ and the weights for

the unwinding of the defects across both marked points are parameterised by two independent variables $w$ and $z$, instead of by a unique variable $z$.

We now define a second family of modules $\mathsf{Y}_{k,\ell,x,y,[z,w]}(N)$, also built from link states with two marked points, that have better properties related to the above remarks. The new modules have some similarities and differences with the modules $\mathsf{X}_{k,\ell,x,y,z}(N)$, which we emphasise throughout.

Like for the modules $\mathsf{X}_{k,\ell,x,y,z}(N)$, the modules $\mathsf{Y}_{k,\ell,x,y,[z,w]}(N)$ are defined for $N \geqslant 2$ with $k, \ell \in \frac{1}{2}\mathbb{Z}_{\geqslant 0}$ and $\frac{N}{2} - k - \ell \in \mathbb{Z}_{\geqslant 0}$. The parameters $x$, $y$, $z$ and $w$ are non-zero complex variables that arise in the action of the algebra on the basis states. The vector space for $\mathsf{Y}_{k,\ell,x,y,[z,w]}(N)$ is spanned by link states on a disc with two marked points a and b, and $N$ nodes on the perimeter. The basis link states satisfy the following five properties:

(P1) The marked points a and b are on the right and the left, respectively. The dashed line ac leaves c on the right of the line bc, and these lines never cross.[6]

(P2) Loop segments may connect pairs of outer nodes, creating arches.

(P3) A defect can connect a to b.

(P4) A defect can connect a to itself only if the resulting curve separates the component of the disc containing b from the component adjacent to the boundary. The same holds for a ↔ b. In other words, the diagrams

$$\text{and} \tag{57}$$

are not set to zero.

(P5) A defect can connect a marked point to a node on the boundary of the disc. A defect connecting b to an outer node can cross the line ac at most once, and in this case, the point a must lie to its right along the dashed line. Such a defect may also cross the line bc, but only if the following condition is satisfied: a walker traveling on the defect from b to the outer node must cross the line ac before the line bc. Similarly, a defect attached to a can cross the line bc at most once, with the point a to its right along the dashed line. Moreover, a defect attached to a can never cross the line ac. Finally, the total number of defects crossing dashed lines can take the values $0, 1, \ldots, 2k + 2\ell - 1$.

Here are examples of link states in $\mathsf{Y}_{\frac{3}{2},\frac{1}{2},x,y,[z,w]}(12)$:



$$v_1 = \quad , \qquad v_2 = \quad , \qquad v_3 = \quad . \tag{58}$$

For $k \geqslant \ell$, the link states spanning the vector space of $\mathsf{Y}_{k,\ell,x,y,[z,w]}(N)$ split into four types:

(i) Link states with $2k$ and $2\ell$ defects attached to a and b respectively, and with the depth $p$ taking the values $1, 2, \ldots, \frac{N}{2} - k - \ell$.

---

[6]We sometimes place the marked points higher or lower in the diagrams for the states of $\mathsf{Y}_{k,\ell,x,y,[z,w]}$, but always keep a on the right and b on the left.

(ii) Link states with $(r + k - \ell)$ and $(r + \ell - k)$ defects attached to a and b respectively, with $r = k - \ell, k - \ell + 1, \ldots, k + \ell$, and with depth $p = 0$. These are obtained from link states of type (i) by connecting $(k + \ell - r)$ pairs of defects between a and b.

(iii) Link states where b has no defects and a has $2r$ defects, with $r = k - \ell - s$, and $0 < s < k - \ell$. These are obtained from link states of type (i) by connecting $2\ell$ pairs of defects between a and b, and then $s$ pairs of defects attached to a.

(iv) Link states with no defects, obtained from link states of type (i) by connecting $2\ell$ pairs of defects between a and b, and then $k - \ell$ pairs of defects attached to a. These link states are only allowed if $N$ is even and $z^{2(k-\ell)} = 1$. Otherwise, they are not included in the basis.

The dimension of $\mathsf{Y}_{k,\ell,x,y,[z,w]}(N)$ is larger than or equal to the dimension of $\mathsf{X}_{k,\ell,x,y,z}(N)$. Indeed, the modules $\mathsf{X}_{k,\ell,x,y,z}(N)$ only have link states of types (i) and (ii). For $k < \ell$, the link states also split into four types, following the above rules with $k \leftrightarrow \ell$ and a $\leftrightarrow$ b.

As a first example, here is the basis of $\mathsf{Y}_{1,0,x,y,[z,w]}(4)$ for $z^2 = 1$:

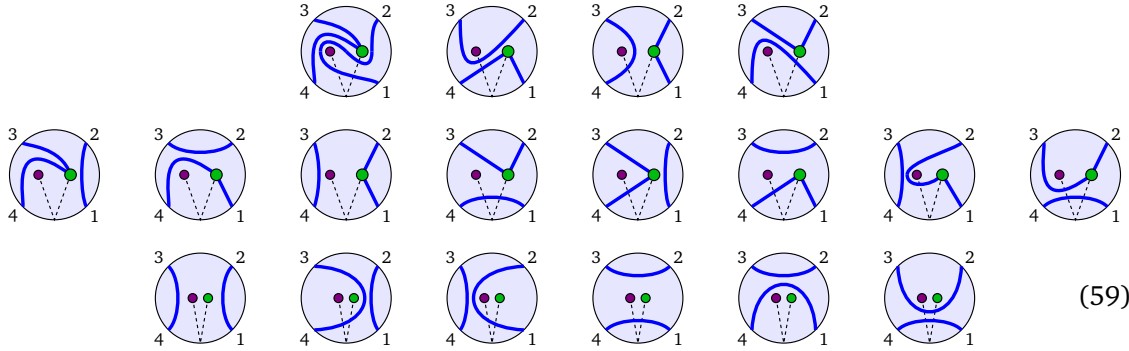

$$(59)$$

This is different from the basis for $\mathsf{X}_{1,0,x,y,z}(4)$. In this example, the first, second and third lines respectively depict the link states of types (i), (ii) and (iv). There are no link states of type (iii) in this case. Comparing this with (48), we see that the second line has twice the amount of link states for $\mathsf{Y}_{1,0,x,y,[z,w]}(4)$ and that the last line is absent for $\mathsf{X}_{1,0,x,y,z}(4)$, thus confirming that $\mathsf{Y}_{1,0,x,y,[z,w]}(4)$ is a larger representation. In particular, we see that the state (49) is a basis state of $\mathsf{Y}_{1,0,x,y,[z,w]}(4)$, but *not* of $\mathsf{X}_{1,0,x,y,z}(4)$. Moreover, if $z^2 \neq 1$, the basis of $\mathsf{Y}_{1,0,x,y,[z,w]}(4)$ is obtained from (59) by removing the states of the third line. As a second example, here is the basis of $\mathsf{Y}_{\frac{1}{2},\frac{1}{2},x,y,[z,w]}(4)$:

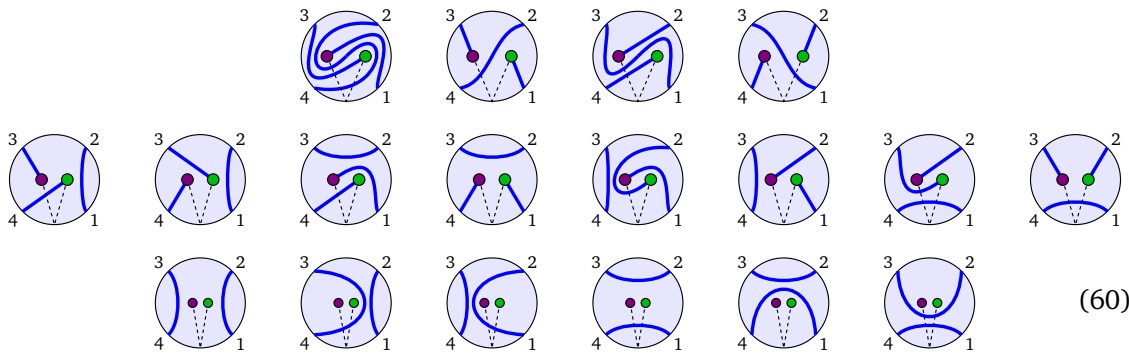

$$(60)$$

In this case, we readily see that there is a bijection with the basis states of $\mathsf{X}_{\frac{1}{2},\frac{1}{2},x,y,z}(4)$ in (50).

The action $c \cdot v$ of the algebra on this vector space is defined as before, namely one inserts the link state $v$ inside the connectivity $c$ and reads the results from the outer perimeter of the diagram. The rules for assigning the weights are

$$\text{(diagrams)} \tag{61}$$

as well as

$$\Omega^N \cdot v = z^{2(k-\ell)} v, \qquad \forall v \in \mathsf{Y}_{k,\ell,x,y,[z,w]}(N), \tag{62}$$

where the weights of the non-contractible loops are parameterised as

$$\alpha_{\mathsf{a}} = x + x^{-1}, \qquad \alpha_{\mathsf{b}} = y + y^{-1}, \qquad \alpha_{\mathsf{ab}} = w + w^{-1}. \tag{63}$$

Moreover, if $z^{2(k-\ell)} \neq 1$, the diagrammatic rule that allows pairs of defects attached to the same marked point to connect with unit weight is changed if its application produces a link state of type (iv). In this case, it instead yields a vanishing result. Thus it is only possible to produce a link state of type (iv) if $N$ is even and $z^{2(k-\ell)} = 1$.

We note that, for $k > 0$ or $\ell > 0$, applying the rules (61) for the action $c \cdot v$ may produce states whose defects do not satisfy the property (P5). But the property (62) then allows us to re-express these states in terms of basis link states satisfying this property.

These rules are different from those of the modules $\mathsf{X}_{k,\ell,x,y,z}(N)$ given in (52). First, we see that the defects attached to a given point may connect in pairs if they do so after encircling the second marked point. Second, the point a is always on the right of the point b. There is thus only one way that a defect can connect these two points. This defect is removed and replaced by a unit weight. There is thus no parameter $\mu$. Third, there are no rules that allow one to move a marked point across the dashed line attached to the other marked point. Finally, the parameter $w$ now parameterises $\alpha_{\mathsf{ab}}$, whereas $z$ only arises in the connection of pairs of defects. We also note that there are some coincidences between the two families of modules:

$$N \text{ even}: \qquad \mathsf{Y}_{0,0,x,y,[z,w]}(N) = \mathsf{X}_{0,0,x,y,w}(N), \tag{64a}$$

$$N \text{ odd}: \qquad \mathsf{Y}_{\frac{1}{2},0,x,y,z}(N) = \mathsf{X}_{\frac{1}{2},0,x,y,z}(N), \qquad \mathsf{Y}_{0,\frac{1}{2},x,y,z}(N) = \mathsf{X}_{0,\frac{1}{2},x,y,z}(N). \tag{64b}$$

We now make the following claim.

CLAIM 3.1 *The vector space* $\mathsf{Y}_{k,\ell,x,y,[z,w]}(N)$ *endowed with rules* (61) *and* (62) *for the action of* $\mathcal{E}\mathsf{PTL}_N(\beta)$ *produces a representation.*

This is a non-trivial claim which we leave without proof, and which we have verified for many small system sizes using a computer.

It stems from (62) that the eigenvalues of $\Omega$ are given by $\omega_{N/2,n}$ with $n \in \{0, 1, \ldots, N-1\}$, where

$$\omega_{m,n} = z^{(k-\ell)/m} e^{in\pi/m}. \tag{65}$$

We define the projectors over the corresponding eigenspaces of $\Omega$

$$\Pi_{m,n} = \frac{1}{N} \sum_{j=0}^{N-1} \omega_{m,n}^{-j} \Omega^j. \tag{66}$$

We also note that $Y_{k,\ell,x,y,[z,w]}$ does not depend on $z$ for $(k,\ell) = (0,0)$, whereas it has a $z$-dependence in all the other cases. Moreover, it depends on $w$ only if $N$ is even and $z^{2(k-\ell)} = 1$, and in this case the module is invariant under $w \mapsto w^{-1}$. We use the notation $Y_{k,\ell,x,y,[z,w]}(N)$ for all cases, but sometimes write $Y_{k,\ell,x,y,z}(N)$ when discussing the case where $N$ is odd.

The reason for the extra condition on $z$ for the link states of type (iv) to be in the vector space is clear. In our construction, $\Omega^N$ acts on these link states as the identity times a unit prefactor. If these link states were to be included and $z^{2(k-\ell)} \neq 1$, then (62) would no longer hold and in fact one could show that the result is no longer a representation of $\mathcal{E}\mathsf{PTL}_N(\beta)$. This is related to the issue discussed in [21] whereby it is impossible, for $N$ even, to simultaneously have a module with the relations $\Omega^N = \gamma 1$ with $\gamma \neq 1$, and $E = e_2 e_4 \ldots e_N \neq 0$, since $\Omega^N E = E$.

## 3.3 Properties of the modules $Y_{k,\ell,x,y,[z,w]}(N)$

The representation $Y_{k,\ell,x,y,[z,w]}(N)$ has many useful properties. The similar properties of the modules $X_{k,\ell,x,y,z}(N)$ were studied in [13], so here we keep the presentation brief and do not repeat all the proofs. Some of them are given in Appendix A.

**Cyclicity.** Consider the link state of maximal depth $p = \frac{N}{2} - k - \ell$ defined as

$$u_{k,\ell}(N) = \quad \text{\raisebox{-2em}{}} \qquad . \tag{67}$$

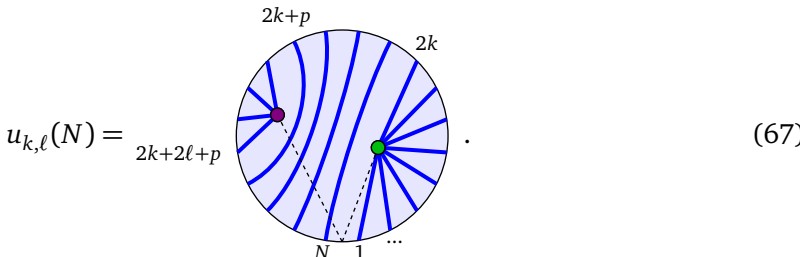

For $(k,\ell) \neq (0,0)$, the module $Y_{k,\ell,x,y,[z,w]}(N)$ is cyclic with respect to any link state of type (i) of maximal depth $p = \frac{N}{2} - k - \ell$. In other words, the entire module is generated from the action of $\mathcal{E}\mathsf{PTL}_N(\beta)$ on any such state. These states are $u_{k,\ell}(N)$ and its shifts $\Omega^j \cdot u_{k,\ell}(N)$. The same result holds for $k = \ell = 0$ if at least one of $\alpha_\mathsf{a}$ or $\alpha_\mathsf{b}$ is non-zero.

**Gluing map.** In [13], a gluing operator $g_z$ was defined as acting on pairs of standard modules $W_{k,x}(N_\mathsf{a})$ and $W_{\ell,y}(N_\mathsf{b})$, creating states in $X_{k,\ell,x,y,z}(N)$, and assigning to those states the action of $\mathcal{E}\mathsf{PTL}_N(\beta)$ with the rules (52). Here we introduce a gluing operator $g_{[z,w]}$ that acts similarly, but instead endows the vector space of the glued link states with the action of $Y_{k,\ell,x,y,[z,w]}(N)$. The gluing map

$$g_{[z,w]} : \; W_{k,x}(N_\mathsf{a}) \otimes W_{\ell,y}(N_\mathsf{b}) \to Y_{k,\ell,x,y,[z,w]}(N), \qquad N = N_\mathsf{a} + N_\mathsf{b}, \tag{68}$$

is a linear map defined as follows. For any pair of link states $(u,v) \in B_k(N_\mathsf{a}) \times B_\ell(N_\mathsf{b})$, the result of $g_{[z,w]}(u,v)$ is the link state with $N = N_\mathsf{a} + N_\mathsf{b}$ nodes, obtained by inserting the state $u$ on the nodes $(1,2,\ldots,N_\mathsf{a})$, and the state $v$ on the nodes $(N_\mathsf{a}+1, N_\mathsf{a}+2, \ldots, N_\mathsf{a}+N_\mathsf{b})$. Here is an example for $N_\mathsf{a} = 5$ and $N_\mathsf{b} = 6$:

$$g_{[z,w]}\left( \quad , \quad \right) = \quad . \tag{69}$$

By a slight abuse of notation, we define the gluing acting on any pair of submodules $M_a \subseteq W_{k,x}(N_a)$ and $M_b \subseteq W_{\ell,y}(N_b)$, as

$$g_{[z,w]}(M_a, M_b) = \left\{ c \cdot g_{[z,w]}(u, v) \,\middle|\, u \in M_a \,, v \in M_b \,, c \in \mathcal{E}\mathrm{PTL}(\beta) \right\}. \tag{70}$$

For $(k, \ell) \neq (0,0)$, from the above cyclicity property, we have

$$g_{[z,w]}\big(W_{k,x}(N_a), W_{\ell,y}(N_b)\big) = Y_{k,\ell,x,y,[z,w]}(N). \tag{71}$$

The same applies to $(k, \ell) = (0,0)$ if either $\alpha_a \neq 0$ or $\alpha_b \neq 0$. By construction, $g_{[z,w]}(M_a, M_b)$ is always a submodule of $Y_{k,\ell,x,y,[z,w]}(N)$.

**Dimensions.**   The dimensions of the representations are given by

$$N \text{ even:} \quad \dim Y_{k,\ell,x,y,[z,w]}(N) = \begin{cases} |B_0(N)| + \sum_{m=1}^{N/2} 2m|B_m(N)| = \left(\frac{N}{2} + 1\right)\binom{N}{\frac{N}{2}}, & z^{2(k-\ell)} = 1 \,, \\[2ex] \sum_{m=1}^{N/2} 2m|B_m(N)| = \frac{N}{2}\binom{N}{\frac{N}{2}}, & \text{otherwise,} \end{cases} \tag{72a}$$

$$N \text{ odd:} \quad \dim Y_{k,\ell,x,y,z}(N) = \sum_{m=1/2}^{N/2} 2m|B_m(N)| = \frac{N+1}{2}\binom{N}{\frac{N+1}{2}}. \tag{72b}$$

The argument leading to this dimension counting is developed in Appendix A.

**Filtrations.**   The action of the algebra can never increase the depth $p$. It may only reduce it or leave it unchanged. We denote by $Y^{(p)}_{k,\ell,x,y,[z,w]}(N)$ the submodule of $Y_{k,\ell,x,y,[z,w]}(N)$ spanned by link states of depth at most $p$, with $p = 0, 1, \ldots, \frac{N}{2} - k - \ell$. The submodule $Y^{(0)}_{k,\ell,x,y,[z,w]}(N)$ decomposes further. Indeed, the defects of states of zero depth may connect pairwise. However the action of the algebra cannot undo the corresponding processes. We then denote by $Y^{(0,r)}_{k,\ell,x,y,[z,w]}(N)$ the submodule of $Y^{(0)}_{k,\ell,x,y,[z,w]}(N)$ where the total number of defects of a and b is at most $2r$, with $r \leqslant k + \ell$. We have the filtrations

$$N \text{ even:} \quad Y^{(0,0)}_{k,\ell,x,y,[z,w]}(N) \subset Y^{(0,1)}_{k,\ell,x,y,[z,w]}(N) \subset \cdots \subset Y^{(0,k+\ell)}_{k,\ell,x,y,[z,w]}(N) = Y^{(0)}_{k,\ell,x,y,[z,w]}(N), \tag{73a}$$

$$N \text{ odd:} \quad Y^{(0,1/2)}_{k,\ell,x,y,z}(N) \subset Y^{(0,3/2)}_{k,\ell,x,y,z}(N) \subset \cdots \subset Y^{(0,k+\ell)}_{k,\ell,x,y,z}(N) = Y^{(0)}_{k,\ell,x,y,z}(N), \tag{73b}$$

and

$$Y^{(0)}_{k,\ell,x,y,[z,w]}(N) \subset Y^{(1)}_{k,\ell,x,y,[z,w]}(N) \subset \cdots \subset Y^{(N/2-k-\ell)}_{k,\ell,x,y,[z,w]}(N) = Y_{k,\ell,x,y,[z,w]}(N), \tag{73c}$$

for both parities of $N$. For $N$ even with $z^{2(k-\ell)} \neq 1$, the filtration (73a) holds with $Y^{(0,0)}_{k,\ell,x,y,[z,w]}(N) = 0$. We also define the quotient modules

$$M^{(0,r)}_{k,\ell,x,y,[z,w]}(N) = Y^{(0,r)}_{k,\ell,x,y,[z,w]}(N) \Big/ Y^{(0,r-1)}_{k,\ell,x,y,[z,w]}(N), \tag{74a}$$

$$M^{(p)}_{k,\ell,x,y,[z,w]}(N) = Y^{(p)}_{k,\ell,x,y,[z,w]}(N) \Big/ Y^{(p-1)}_{k,\ell,x,y,[z,w]}(N). \tag{74b}$$

The equivalence relations associated to the above filtrations will be useful in the subsequent discussions. We say that two elements $v_1, v_2$ in $Y_{k,\ell,x,y,z,w}(N)$ are equivalent at depth $p$ if their difference consists only of states of depth at most $p - 1$. We write this as

$$v_1 \equiv v_2 \, [[p-1]] \qquad \Longleftrightarrow \qquad v_1 - v_2 \in Y^{(p-1)}_{k,\ell,x,y,[z,w]}(N). \tag{75}$$

Similarly, we write

$$v_1 \equiv v_2 \, [[0, r-1]] \qquad \Longleftrightarrow \qquad v_1 - v_2 \in Y^{(0,r-1)}_{k,\ell,x,y,[z,w]}(N). \tag{76}$$

**Eigenvalues of $F$ and $\bar{F}$ and genericity of $[z, w]$.** Let us define

$$f_0 = \bar{f}_0 = w + w^{-1}, \qquad f_{m,n} = q^m \omega_{m,n} + q^{-m} \omega_{m,n}^{-1}, \qquad \bar{f}_{m,n} = q^{-m} \omega_{m,n} + q^m \omega_{m,n}^{-1}. \quad (77)$$

These are the eigenvalues of $F$ and $\bar{F}$ in the modules $\mathsf{W}_{0,w}(N)$ and $\mathsf{W}_{m,\omega_{m,n}}(N)$.

Repeating the arguments used in [13] for the module $\mathsf{X}_{k,\ell,x,y,z}(N)$, we find that the eigenvalues of $F$ in the quotient modules $\mathsf{M}^{(p)}_{k,\ell,x,y,[z,w]}$ and $\mathsf{M}^{(0,r)}_{k,\ell,x,y,[z,w]}$ are given by

$$\mathsf{M}^{(0,0)}_{k,\ell,x,y,[z,w]}(N): \qquad \{f_0\}, \tag{78a}$$

$$\mathsf{M}^{(0,r)}_{k,\ell,x,y,[z,w]}(N): \qquad \{f_{r,n} \mid n = 0, 1, \ldots, 2r-1\}, \qquad\qquad 0 < r \leqslant k+\ell, \quad (78b)$$

$$\mathsf{M}^{(p)}_{k,\ell,x,y,[z,w]}(N): \qquad \{f_{k+\ell+p,n} \mid n = 0, 1, \ldots, 2k+2\ell+2p-1\}, \qquad p \geqslant 1. \quad (78c)$$

The eigenvalues of $F$ in $\mathsf{Y}_{k,\ell,x,y,[z,w]}(N)$ are given by the union of those of its quotient modules $\mathsf{M}^{(0,r)}_{k,\ell,x,y,[z,w]}$ and $\mathsf{M}^{(p)}_{k,\ell,x,y,[z,w]}$. The eigenvalues of $\bar{F}$ in these modules are obtained from (78) by changing $q \to q^{-1}$.

We say that $[z, w]$ is generic for $\mathsf{Y}_{k,\ell,x,y,[z,w]}(N)$ if all the eigenvalues (78) of $F$ are distinct, and likewise for those of $\bar{F}$. To analyse the coincidence of these eigenvalues, we define

$$f(m, \omega) = q^m \omega + q^{-m} \omega^{-1}, \qquad \bar{f}(m, \omega) = q^m \omega^{-1} + q^{-m} \omega. \quad (79)$$

Recalling that we work with values of $q$ that are not roots of unity, we observe that, given a pair of modules $(\mathsf{W}_{m,\omega}, \mathsf{W}_{m',\omega'})$, the equalities $f(m, \omega) = f(m', \omega')$ and $\bar{f}(m, \omega) = \bar{f}(m', \omega')$ hold simultaneously if and only if $\omega$ and $\omega'$ are of the form

$$\omega = \varepsilon q^{\sigma m'}, \qquad \omega' = \varepsilon q^{\sigma m}, \qquad \sigma, \varepsilon \in \{+1, -1\}. \quad (80)$$

There are three kinds of non-generic values of $[z, w]$ for $\mathsf{Y}_{k,\ell,x,y,[z,w]}(N)$:

- The first kind arises for $N$ even with $z^{2(k-\ell)} = 1$ and $w = \varepsilon q^{\sigma m}$, where $m \in \{1, 2, \ldots, \frac{N}{2}\}$ and $\sigma, \varepsilon \in \{+1, -1\}$. In this case, the eigenvalues of $(F, \bar{F})$ coincide on the pair $(\mathsf{W}_{0,\varepsilon q^{\sigma m}}, \mathsf{W}_{m,\varepsilon})$. These eigenvalues are given by $f_0 = \bar{f}_0 = \varepsilon(q^m + q^{-m})$.

- The second kind arises for $N$ even with $z^{2(k-\ell)} = q^{2\sigma m m'}$, where $m, m'$ are distinct integers in $\{1, 2, \ldots, \frac{N}{2}\}$ and $\sigma \in \{+1, -1\}$. In this case, the eigenvalues of $(F, \bar{F})$ coincide on the two pairs $(\mathsf{W}_{m,q^{\sigma m'}}, \mathsf{W}_{m',q^{\sigma m}})$ and $(\mathsf{W}_{m,-q^{\sigma m'}}, \mathsf{W}_{m',-q^{\sigma m}})$. If $M = mm'$ is not a prime number, then there may exist several pairs of integers $\{m_i, m_i'\}$ such that $m_i, m_i' \in \{1, 2, \ldots, \frac{N}{2}\}$ and $m_i m_i' = M$. For each pair $\{m_i, m_i'\}$, the two pairs of modules $(\mathsf{W}_{m_i,q^{\sigma m_i'}}, \mathsf{W}_{m_i',q^{\sigma m_i}})$ and $(\mathsf{W}_{m_i,-q^{\sigma m_i'}}, \mathsf{W}_{m_i',-q^{\sigma m_i}})$ have coinciding eigenvalues of $(F, \bar{F})$. These eigenvalues are given by $(\varphi^\sigma_{m_i,m_i'}, \bar{\varphi}^\sigma_{m_i,m_i'})$ and $(-\varphi^\sigma_{m_i,m_i'}, -\bar{\varphi}^\sigma_{m_i,m_i'})$ respectively, where

$$\varphi^\sigma_{m,m'} = q^{m+\sigma m'} + q^{-m-\sigma m'}, \qquad \bar{\varphi}^\sigma_{m,m'} = q^{m-\sigma m'} + q^{-m+\sigma m'}. \quad (81)$$

  Note that all these pairs of eigenvalues are distinct, namely

$$\varepsilon_i(\varphi^\sigma_{m_i,m_i'}, \varphi^\sigma_{m_i,m_i'}) \neq \varepsilon_j(\varphi^\sigma_{m_j,m_j'}, \varphi^\sigma_{m_j,m_j'}), \qquad \text{if } \{m_i, m_i'\} \neq \{m_j, m_j'\} \text{ or } \varepsilon_i \neq \varepsilon_j. \quad (82)$$

- The third kind arises for $N$ odd with $z^{2(k-\ell)} = \varepsilon q^{2\sigma m m'}$, where $m, m'$ are distinct half-integers in $\{\frac{1}{2}, \frac{3}{2}, \ldots, \frac{N}{2}\}$ and $\sigma, \varepsilon \in \{+1, -1\}$. In this case, the eigenvalues of $(F, \bar{F})$ coincide on the pair $(\mathsf{W}_{m,\varepsilon q^{\sigma m'}}, \mathsf{W}_{m',\varepsilon q^{\sigma m}})$. Here, like for the second kind, several such pairs of modules can coexist, and they always correspond to distinct eigenvalues.

## 3.4 Decomposition of $\mathsf{Y}_{k,\ell,x,y,[z,w]}(N)$ and fusion rules for generic $[z,w]$

For generic values of $[z,w]$, the module $\mathsf{Y}_{k,\ell,x,y,[z,w]}(N)$ decomposes into a direct sum of irreducible standard modules as

$$N \text{ even:} \qquad \mathsf{Y}_{k,\ell,x,y,[z,w]}(N) \simeq \begin{cases} \mathsf{W}_{0,w}(N) \oplus \displaystyle\bigoplus_{m=1}^{N/2} \bigoplus_{n=0}^{2m-1} \mathsf{W}_{m,\omega_{m,n}}(N), & z^{2(k-\ell)} = 1, \\[2ex] \displaystyle\bigoplus_{m=1}^{N/2} \bigoplus_{n=0}^{2m-1} \mathsf{W}_{m,\omega_{m,n}}(N), & \text{otherwise,} \end{cases} \tag{83a}$$

$$N \text{ odd:} \qquad \mathsf{Y}_{k,\ell,x,y,z}(N) \simeq \bigoplus_{m=1/2}^{N/2} \bigoplus_{n=0}^{2m-1} \mathsf{W}_{m,\omega_{m,n}}(N). \tag{83b}$$

This is obtained using the same arguments as those used in [13] for $\mathsf{X}_{k,\ell,x,y,z}(N)$. We discuss this further in Appendix A. This decomposition has a natural diagrammatic interpretation:

$$\text{(84)}$$

For illustrative purposes, these diagrams are drawn here for $\mathsf{Y}_{k,\ell,x,y,[z,w]}(N)$ with $N$ even, $k > \ell$, and $z^{2(k-\ell)} = 1$. In these diagrams, only the loop segments that act as effective defects are drawn, and not the other arcs that are present and contribute to the total number $N$ of nodes. The splitting of the right-hand side in four contributions corresponds to the classification of link states in the types (i), (ii), (iii) and (iv) described in Section 3.2. For $z^{2(k-\ell)} \neq 1$, the only difference is that the last term with link states of type (iv) is absent. Thus in the decomposition, the effective number $2m$ of defects accounts for both the total number of defects and for the arcs lying between the two marked points that contribute to a non-zero depth $p$. The sums over $n$ account for the possible eigenvalues $(f_{m,n}, \bar{f}_{m,n})$ of $(F, \bar{F})$. Then the quotient modules $\mathsf{M}^{(p)}_{k,\ell,x,y,[z,w]}$ and $\mathsf{M}^{(0,r)}_{k,\ell,x,y,[z,w]}$ each correspond to one value of $m$ in this decomposition, namely

$$\mathsf{M}^{(0,0)}_{k,\ell,x,y,[z,w]}(N) \simeq \mathsf{W}_{0,w}(N), \tag{85a}$$

$$\mathsf{M}^{(0,r)}_{k,\ell,x,y,[z,w]}(N) \simeq \bigoplus_{n=0}^{2r-1} \mathsf{W}_{r,\omega_{r,n}}(N), \qquad 0 < r \leqslant k+\ell, \tag{85b}$$

$$\mathsf{M}^{(p)}_{k,\ell,x,y,[z,w]}(N) \simeq \bigoplus_{n=0}^{2m-1} \mathsf{W}_{m,\omega_{m,n}}(N), \qquad m = k+\ell+p, \qquad p \geqslant 1. \tag{85c}$$

From the decompositions (83), we see that

$$\mathsf{Y}_{k,\ell,x,y,[z,w]} \simeq \mathsf{Y}_{\ell,k,y,x,[z^{-1},w]}. \tag{86}$$

Let us define the elements of $\mathcal{E}\mathsf{PTL}_N(\beta)$

$$Q_0 = \prod_{m'=\iota}^{N/2} \prod_{n'=0}^{2m'-1} \frac{\boldsymbol{F} - f_{m',n'}}{f_0 - f_{m',n'}}, \tag{87a}$$

$$Q_{m,n} = \frac{\boldsymbol{F} - f_0}{f_{m,n} - f_0} \prod_{\substack{m'=\iota \\ (m',n') \neq (m,n)}}^{N/2} \prod_{n'=0}^{2m'-1} \frac{\boldsymbol{F} - f_{m',n'}}{f_{m,n} - f_{m',n'}}, \quad m = \iota, \iota+1, \dots, \tfrac{N}{2}, \quad n = 0, 1, \dots, 2m-1, \tag{87b}$$

where

$$\iota = \begin{cases} 0, & N \text{ even}, \\ \frac{1}{2}, & N \text{ odd}. \end{cases} \tag{88}$$

In $\mathsf{Y}_{k,\ell,x,y,[z,w]}(N)$ with $[z,w]$ generic, these elements act as projectors on each of the individual irreducible standard modules in the direct sum decomposition (83).

The definition of $\boldsymbol{F}$ in (28) expresses it as a sum of $2^N$ terms. Using this decomposition and acting on link states with the maximal number of defects and with depth $p = \frac{N}{2} - k - \ell$, we find

$$\boldsymbol{F} \cdot v \equiv \left( q^{N/2}\Omega + q^{-N/2}\Omega^{-1} \right) \cdot v \quad [[\tfrac{N}{2} - k - \ell - 1]], \qquad \forall v \in \mathsf{M}^{(N/2-k-\ell)}_{k,\ell,x,y,[z,w]}(N). \tag{89}$$

After simple manipulations, we obtain

$$\Pi_{N/2,v}\boldsymbol{F} \cdot v \equiv f_{N/2,v}\Pi_{N/2,v} \cdot v \quad [[\tfrac{N}{2} - k - \ell - 1]], \tag{90a}$$

$$\Pi_{N/2,v}Q_{N/2,n} \cdot v \equiv \delta_{v,n}\Pi_{N/2,n} \cdot v \quad [[\tfrac{N}{2} - k - \ell - 1]]. \tag{90b}$$

This implies that

$$Q_{N/2,n} \cdot v \equiv \Pi_{N/2,n} \cdot v \quad [[\tfrac{N}{2} - k - \ell - 1]], \qquad \forall v \in \mathsf{M}^{(N/2-k-\ell)}_{k,\ell,x,y,[z,w]}(N). \tag{91}$$

Therefore, the action of $Q_{N/2,n}$ and $\Pi_{N/2,n}$ in $\mathsf{Y}_{k,\ell,x,y,[z,w]}(N)$ on link states of depth $p = \frac{N}{2} - k - \ell$ is identical modulo states of smaller depths.

**Generic fusion rules.** Here we use the definition of a fusion rule of two modules presented in the Introduction and apply it to the fusion of two irreducible standard modules $\mathsf{W}_{k,x}(N_\mathrm{a})$ and $\mathsf{W}_{\ell,y}(N_\mathrm{b})$. For this, we consider the modules $\mathsf{Y}_{k,\ell,x,y,[z,w]}(N)$ on increasing values of $N = N_\mathrm{a} + N_\mathrm{b}$, for fixed values of the parities of $N_\mathrm{a}$ and $N_\mathrm{b}$. For a given indecomposable module $\mathsf{M}$, we write $\mathsf{W}_{k,x} \times \mathsf{W}_{\ell,y} \to \mathsf{M}$ if there exists a positive integer $N_0$ such that, for all $N \geqslant N_0$ with $N \equiv N_0 \bmod 2$, the module $\mathsf{M}(N)$ appears as a direct summand in the decomposition of $\mathsf{Y}_{k,\ell,x,y,[z,w]}(N)$ for some values of $[z,w]$. We also write $\mathsf{W}_{k,x} \times \mathsf{W}_{\ell,y} \to \{\mathsf{M}_1, \mathsf{M}_2, \mathsf{M}_3, \dots\}$, if each $\mathsf{M}_j$ is indecomposable and satisfies $\mathsf{W}_{k,x} \times \mathsf{W}_{\ell,y} \to \mathsf{M}_j$. The decompositions (83) then induce the fusion rules

$$\mathsf{W}_{k,x} \times \mathsf{W}_{\ell,y} \to \begin{cases} \{\mathsf{W}_{m,t} \mid m \in \mathbb{Z}_{\geqslant 0}, \, m+k+\ell \in \mathbb{Z}, \, \mathsf{W}_{m,t} \text{ is irreducible}\}, & k \neq \ell, \\ \{\mathsf{W}_{m,t} \mid m \in \mathbb{Z}_{\geqslant 0}, \, t^{2m} = 1, \, \mathsf{W}_{m,t} \text{ is irreducible}\}, & k = \ell. \end{cases} \tag{92}$$

Indeed, for these values of $m$ and $t$, and for $N$ large enough, there is always a choice of $[z,w]$ such that $\mathsf{W}_{m,t}(N)$, with $t$ fixed to a value where this module is irreducible, appears as a submodule of $\mathsf{Y}_{k,\ell,x,y,[z,w]}(N)$.

We stress that (92) does not give the *full* fusion rules for the product of two standard modules. Indeed, to get the full fusion rule $\mathsf{W}_{k,x} \times \mathsf{W}_{\ell,y}$, one must consider the decomposition of $\mathsf{Y}_{k,\ell,x,y,[z,w]}$ for all values of $[z,w]$. This also includes the non-generic ones, which we have

not considered here. This decomposition will include modules that are indecomposable yet reducible. As shown in [13] and further discussed in Appendix D for the case $k = \ell = 0$, this can include indecomposable modules that have three composition factors. We in fact present the full fusion rule for $W_{0,x} \times W_{0,y}$ in Section 5.1. We expect that the same kind of modules arise in general in the decomposition of $Y_{k,\ell,x,y,[z,w]}$ for non-generic values of $[z, w]$ and arbitrary values of $k$ and $\ell$. This analysis is however beyond the scope of the present paper.

# 4 The fusion modules of $W_{k,x} \times R_{0,\varepsilon q^m}$ and $W_{k,x} \times Q_{0,\varepsilon q^m}$

In this section, we focus on fusion products involving a standard module $W_{k,x}(N_a)$ and an irreducible module that appears in the decomposition of the standard module with zero defects $W_{0,\varepsilon q^{\sigma m}}(N_b)$. We discuss both the fusion with radical submodules $R_{0,\varepsilon q^{\sigma m}}(N_b)$ and with quotient modules $Q_{0,\varepsilon q^{\sigma m}}(N_b)$. Throughout this section, we focus on generic values of the fusion parameters $[z, w]$. We describe the construction of the fusion modules as well as their decomposition over the irreducible standard modules. After giving the general definitions in Section 4.1, we illustrate these concepts for the two simplest examples, namely $m = 1$ and $m = 2$, in Sections 4.2 and 4.3 respectively. Then in Section 4.4, we describe the constructions and decompositions in the general case.

## 4.1 Definitions and preliminaries

Let us first discuss the fusion product $W \times R$ between a standard module $W_{k,x}(N_a)$ and a radical submodule $R_{0,\varepsilon q^{\sigma m}}(N_b)$. It is defined as

$$\left(W_{k,x}(N_a) \times R_{0,\varepsilon q^{\sigma m}}(N_b)\right)_{[z,w]} = g_{[z,w]}\left(W_{k,x}(N_a), R_{0,\varepsilon q^{\sigma m}}(N_b)\right), \qquad N = N_a + N_b, \qquad (93)$$

where $g_{[z,w]}$ denotes the gluing operation on submodules of standard modules, as defined in (70). Indeed, because $R_{0,\varepsilon q^{\sigma m}}(N_b)$ is a submodule of $W_{0,\varepsilon q^{\sigma m}}(N_b)$, the action of $g_{[z,w]}$ is automatically defined on $W_{k,x}(N_a) \otimes R_{0,\varepsilon q^{\sigma m}}(N_b)$, namely it is obtained simply by restricting to states in the corresponding submodule. A basis of $R_{0,\varepsilon q^{\sigma m}}(N_b)$ is described in Section 2.4 in terms of linear combinations of link states of $W_{0,\varepsilon q^{\sigma m}}(N_b)$. Here is an example where the gluing $g_{[z,w]}$ acts on a pair of basis states of $W_{1,x}(4)$ and $R_{0,\varepsilon q}(4)$:

$$g_{[z,w]}\left( \cdots \right) = \cdots - \frac{\alpha_b}{\beta} \cdots . \qquad (94)$$

The action of the algebra $\mathcal{E}\mathsf{PTL}_N(\beta)$ on these states thus involves twist factors as well as weights for the various types of loops following the rules (61). With the definition (93), it is clear that $(W_{k,x}(N_a) \times R_{0,\varepsilon q^{\sigma m}}(N_b))_{[z,w]}$ is a module over $\mathcal{E}\mathsf{PTL}_N(\beta)$, and that it is a submodule of $(W_{k,x}(N_a) \times W_{0,\varepsilon q^{\sigma m}}(N_b))_{[z,w]}$. Obtaining its decomposition amounts to looking at (83) and figuring out which factors belong to the submodule. As will become clear in the next sections, this turns out to depend non-trivially on $x$ and $[z, w]$.

We now discuss fusion products $W \times Q$ involving a standard module $W_{k,x}(N_a)$ and a quotient module $Q_{0,\varepsilon q^{\sigma m}}(N_b)$. One can define the fusion module $(W_{k,x}(N_a) \times Q_{0,\varepsilon q^{\sigma m}}(N_b))_{[z,w]}$ in two equivalent ways. It can first be defined directly in terms of the diagrammatic action of the algebra on certain subsets of glued link states, which involves an extra quotient relation. Link states in $Y_{k,0,x,\varepsilon q^{\sigma m},[z,w]}$ have two crossing numbers, $r_a$ and $r_b$, counting the loop segments

crossing the dashed lines ac and bc respectively. It is then natural to consider as a basis for this fusion module the subset of link states in $Y_{k,0,x,\varepsilon q^{\sigma m},[z,w]}$ for which the crossing number associated to b satisfies $r_b < m$. There is however an extra subtlety arising in this definition, namely one must apply this procedure not to the entire module $Y_{k,0,x,\varepsilon q^{\sigma m},[z,w]}$, but instead to a certain submodule corresponding to an eigenspace of the central element $F$. This will already be clear in Section 4.2 in the discussion of the example $m = 1$.

The second definition of the fusion module for $W \times Q$ is

$$\left(W_{k,x}(N_a) \times Q_{0,\varepsilon q^{\sigma m}}(N_b)\right)_{[z,w]} = \left(W_{k,x}(N_a) \times W_{0,\varepsilon q^{\sigma m}}(N_b)\right)_{[z,w]} \Big/ \left(W_{k,x}(N_a) \times R_{0,\varepsilon q^{\sigma m}}(N_b)\right)_{[z,w]}. \quad (95)$$

The second definition is useful, as it allows us to deduce the decomposition of $W \times Q$ directly from the decompositions of $W \times W$ and $W \times R$. It is also natural from a CFT point of view. Indeed, fusing with $Q$ requires that one sets the radical states to zero, in the same way that the fusion of Virasoro modules where one is a degenerate conformal field requires studying modules with null states set to zero. We make the two following claims.

CLAIM 4.1 *The two definitions of the fusion modules of* $W \times Q$ *given above are equivalent.*

In Section 4.1, we show that this claim holds in the case $m = 1$.

CLAIM 4.2 *The decomposition of* $W_{k,x}(N_a) \times Q_{0,\varepsilon q^{\sigma m}}(N_b)$ *is independent of* $N = N_a + N_b$.

Equivalently, the decomposition of $W_{k,x}(N_a) \times R_{0,\varepsilon q^{\sigma m}}(N_b)$ can be read off directly from the minimal case $(N_a, N_b) = (2k, 2m)$ by changing the upper bound on $r$ of the direct sum of standard modules $W_{r,\omega_{r,n}}$ from $k + m$ to $\frac{N}{2}$. In Appendix B, we prove this claim in the case $m = 1$.

In the next sections, we present the construction and the arguments for $y = \varepsilon q^m$ with $m > 0$, understanding that the same arguments for $y = \varepsilon q^{-m}$ are obtained by substituting $q \mapsto q^{-1}$. For this discussion, it is also useful to define the operators

$$c_j = \vcenter{\hbox{}} \,, \qquad c_j^\dagger = \vcenter{\hbox{}} \,, \qquad j = 1, 2, \ldots, N \,. \quad (96)$$

These objects are not elements of $\mathcal{E}\mathsf{PTL}_N(\beta)$, as they have $N$ nodes on one perimeter and $N - 2$ on the other perimeter. Clearly, they are obtained by removing one of the two arches from the diagram $e_j$, either the inner or the outer one. They satisfy the relations

$$c_j^\dagger c_j = e_j \,, \qquad c_j c_j^\dagger = \beta \, 1 \,, \qquad c_j c_{j\pm1}^\dagger = 1 \,, \quad (97)$$

where $1$ stands for the identity in $\mathcal{E}\mathsf{PTL}_{N-2}(\beta)$, and the labels of $c_j$ and $c_j^\dagger$ are understood modulo $N$. The action of $c_j$ on a state in $Y_{k,\ell,x,y,[z,w]}(N)$ produces a state in $Y_{k,\ell,x,y,[z,w]}(N-2)$. In our calculations below, the outer arches of $e_j$ always act as spectators, and thus using $c_j$ and $c_j^\dagger$ instead of $e_j$ allows us to simplify the presentation of some of the arguments.

Let us also define the *seed states*

$$v_{k,m} = (1_{2k} \otimes P_{2m}) \cdot u_{k,0}(2k + 2m) = \vcenter{\hbox{}} \,, \quad (98)$$

where $u_{k,\ell}(N)$ is defined in (67), as well as the *reduced intermediate seed states*

$$\sigma_{k,m}^{(p)} = c_{2k-m+p+1} \ldots c_{2k-1} c_{2k} \cdot v_{k,m}, \qquad p = \max(m-2k,0), \ldots, m-1, m. \tag{99}$$

Each subsequent action of the operators $c_j$ reduces the number of nodes by two units, so that $\sigma_{k,m}^{(p)} \in \mathsf{Y}_{k,0,x,\varepsilon q^m,[z,w]}(2k+2p)$. Moreover, each application of $c_j$ reduces the maximal depth of the states by one unit. For $k \geqslant \frac{1}{2}$, these states can be represented diagrammatically as

$$\sigma_{k,m}^{(p)} = \quad$$ 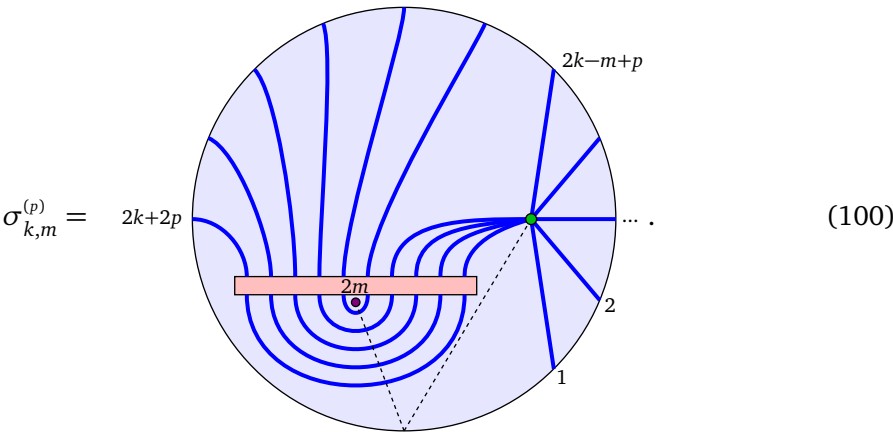 $$. \tag{100}$$

In this diagram, the number of defects of a attached to the projector is $m-p$, and the maximal depth of the link states composing this state is $p$. The states $\sigma_{k,m}^{(p)}$ will be useful in our investigation of the decomposition of $\mathsf{W} \times \mathsf{R}$ and $\mathsf{W} \times \mathsf{Q}$, namely they will allow us to determine which composition factors of $\mathsf{M}_{k,0,x,\varepsilon q^m,[z,w]}^{(p)}(N)$ are present in these modules. In our analysis, we will also define further intermediate states $\sigma_{k,m}^{(p)}$ for $0 \leqslant p < m-2k$, as well as states $\sigma_{k,m}^{(0,r)}$ that will allow us to perform the same analysis for the factors of $\mathsf{M}_{k,0,x,\varepsilon q^m,[z,w]}^{(0,r)}(N)$. Their definitions will depend non-trivially on the inequalities satisfied by $k$, $m$, $p$ and $r$ and will be given on a case-by-case basis. In particular, we investigate the states $\sigma_{0,m}^{(p)}$ in further detail in Appendix C. Finally, we note that the expressions that we derive in this section for the intermediate seed states are valid for $z, w \in \mathbb{C}^\times$, although here we only use them to elucidate properties of the fusion modules for $[z, w]$ generic. Their values for non-generic values of $[z, w]$ will be useful in Section 5.

## 4.2 A first example: the case $m = 1$

We set $m = 1$ for this entire section. We first discuss the fusion $\mathsf{W} \times \mathsf{R}$ and subsequently the fusion $\mathsf{W} \times \mathsf{Q}$. We present the construction for $N_a = 2k$, $N_b = 2$ and $N = 2k + 2$, with $k \in \mathbb{Z}_{\geqslant 0}$. The generalisation of the arguments to larger values of $N$ is discussed in Appendix B, confirming Claim 4.2 in this case.

**The fusion $\mathsf{W} \times \mathsf{R}$.** In the fusion module $(\mathsf{W}_{k,x}(N_a) \times \mathsf{R}_{0,\varepsilon q}(N_b))_{[z,w]}$, the gluing operator creates link states where the point a has $2k$ defects, whereas the point b has no defects and is associated to a Jones-Wenzl projector $P_2$. Both $\mathsf{W}_{k,x}(2k)$ and $\mathsf{R}_{0,\varepsilon q}(2)$ are one-dimensional modules. The gluing operator produces the seed state

$$v = v_{k,1} = \;\; \vcenter{\hbox{(diagram)}} \;\; = (1_{2k} \otimes P_2) \cdot u, \quad \text{with} \quad u = \;\; \vcenter{\hbox{(diagram)}} \;\; = u_{k,0}(2k+2).$$

$$\tag{101}$$

The state $v$ is a linear combination of the link states $u$ and $e_{2k+1} \cdot u$, with respective depths $p = 0$ and $p = 1$. Both $u$ and $e_{2k+1} \cdot u$ have $2k$ defects attached to a. The fusion module $(W_{k,x}(2k) \times R_{0,\varepsilon q}(2))_{[z,w]}$ is the submodule of $Y_{k,0,x,\varepsilon q,[z,w]}(N)$ generated by the action of $\mathcal{E}PTL_N(\beta)$ on the seed state $v$. The decomposition of $Y_{k,0,x,\varepsilon q,[z,w]}(N)$ is directly read off from (83) specialised to $\ell = 0$. Obtaining the decomposition of $(W_{k,x}(2k) \times R_{0,\varepsilon q}(2))_{[z,w]}$ then amounts to finding which of the factors are in the submodule generated by $v$.

With $N = 2k + 2$ and $\ell = 0$, the filtrations (73) read:

$$N \text{ even:} \quad Y^{(0,0)}_{k,0,x,\varepsilon q,[z,w]} \subset Y^{(0,1)}_{k,0,x,\varepsilon q,[z,w]} \subset \cdots \subset Y^{(0,k-1)}_{k,0,x,\varepsilon q,[z,w]} \subset Y^{(0)}_{k,0,x,\varepsilon q,[z,w]} \subset Y^{(1)}_{k,0,x,\varepsilon q,[z,w]}, \quad (102a)$$

$$N \text{ odd:} \quad Y^{(0,1/2)}_{k,0,x,\varepsilon q,z} \subset Y^{(0,3/2)}_{k,0,x,\varepsilon q,z} \subset \cdots \subset Y^{(0,k-1)}_{k,0,x,\varepsilon q,z} \subset Y^{(0)}_{k,0,x,\varepsilon q,z} \subset Y^{(1)}_{k,0,x,\varepsilon q,z}. \quad (102b)$$

For $N$ even, (102a) holds for $z^{2k} = 1$, but also for $z^{2k} \neq 1$ provided that we set $Y^{(0,0)}_{k,0,x,\varepsilon q,[z,w]} = 0$.

We proceed in three steps, acting repeatedly on $v$ with the algebra and examining whether nonzero states of type (i), (ii), and (iii-iv) can be generated. The action of the algebra on these states, in turn, respectively produces the factors appearing in the decomposition of the quotients modules $M^{(1)}_{k,0,x,\varepsilon q,[z,w]}$, $M^{(0)}_{k,0,x,\varepsilon q,[z,w]}$, and $\{M^{(0,r)}_{k,0,x,\varepsilon q,[z,w]}, r \leqslant k-1\}$, given in (85). We now discuss how the different families of states are produced by the action of $\mathcal{E}PTL_N(\beta)$:

- States of type (i): These are the states of depth $p = 1$, namely $u, \Omega \cdot u, \Omega^2 \cdot u, \ldots, \Omega^{2k+1} \cdot u$. Using the projectors $\Pi_{k+1,n}$ defined in (66), we construct the nonzero states $\Pi_{k+1,n} \cdot u$, with $n = 0, 1, \ldots, 2k+1$. These are eigenvectors of $\Omega$ with respective eigenvalues $\omega_{k+1,n}$. Let us also recall that $v \equiv u [[0]]$. In the quotient module $M^{(1)}_{k,0,x,\varepsilon q,[z,w]}(2k+2)$, the state $\Pi_{k+1,n} \cdot u$ spans the one-dimensional submodule $W_{k+1,\omega_n}(2k+2)$. In $Y_{k,0,x,\varepsilon q,[z,w]}(2k+2)$, the same one-dimensional submodule is spanned by the state $Q_{k+1,n} \cdot v$. Since, by (91), $Q_{k+1,n} \cdot v \equiv \Pi_{k+1,n} \cdot u [[0]]$, the state $Q_{k+1,n} \cdot v$ is clearly non-zero. This confirms that each of the factors $W_{k+1,\omega_n}(2k + 2)$ arises in the decomposition of the fusion module $(W_{k,x}(2k) \times R_{0,\varepsilon q}(2))_{[z,w]}$. This holds for all $x, z, w \in \mathbb{C}^\times$.

- States of type (ii): These are the states of depth $p = 0$ with $2k$ defects attached to the point a. We first note that

$$e_j \cdot v = 0, \qquad j \in \{1, 2, \ldots, 2k-1\} \cup \{2k+1\}. \quad (103)$$

For $k \geqslant \frac{1}{2}$, the action of $e_{2k}$ gives

$$e_{2k} \cdot v = c^\dagger_{2k} \sigma^{(0)}_{k,1}, \qquad \text{where} \qquad \sigma^{(0)}_{k,1} = c_{2k} \cdot v = (x^{-1}\Omega + \varepsilon 1) \cdot u_{k,0}(2k). \quad (104)$$

Here, $u_{k,0}(2k)$ is defined in (67) and is a state of maximal depth in $Y_{k,0,x,\varepsilon q,[z,w]}(2k)$. Applying $e_{2k+2}$ to $v$, we find that the result is also expressible in terms of $\sigma^{(0)}_{k,1}$:

$$e_{2k+2} \cdot v = \varepsilon x \, c^\dagger_{2k+2} \sigma^{(0)}_{k,1}. \quad (105)$$

Hence, whether the resulting action of the algebra yields a zero result depends only on the reduced intermediate state $\sigma^{(0)}_{k,1} \in M^{(0)}_{k,0,x,\varepsilon q,[z,w]}(2k)$. To see which of its composition factors are produced, we apply $\Pi_{k,n}$ and find

$$\Pi_{k,n} \cdot \sigma^{(0)}_{k,1} = (x^{-1}\omega_{k,n} + \varepsilon) \Pi_{k,n} \cdot u_{k,0}(2k), \qquad \omega_{k,n} = z \, e^{i\pi n/k}. \quad (106)$$

If $z^{2k} \neq (-\varepsilon x)^{2k}$, all the above vectors are nonzero and the action of the algebra on $\sigma^{(0)}_{k,1}$ generates the whole quotient module $M^{(0)}_{k,0,x,\varepsilon q,[z,w]}(2k)$, namely the direct sum of factors $W_{k,\omega_{k,n}}(2k)$ with $n \in \{0, 1, \ldots, 2k-1\}$. If instead

$$z = z_j = -\varepsilon x \, e^{-i\pi j/k}, \qquad \text{for some} \qquad j \in \{0, 1, \ldots, 2k-1\}, \quad (107)$$

then $z^{2k} = (-\varepsilon x)^{2k}$ and $\Pi_{k,j} \cdot \sigma_{k,1}^{(0)} = 0$. We conclude that, in this case, the action of the algebra on the seed state $v$ produces at depth $p = 0$ the subspace $\bigoplus_{0 \leqslant n \leqslant 2k-1, n \neq j} W_{k,\omega_{k,n}}(2k)$ of $M_{k,0,x,\varepsilon q,[z_j,w]}^{(0)}(2k)$. By using the insertion algorithm to reinsert the extra arc, we conclude that the action of the algebra on the seed state $v$ produces the factors $\bigoplus_{0 \leqslant n \leqslant 2k-1, n \neq j} W_{k,\omega_{k,n}}(2k+2)$ of type (ii). The missing factor is $W_{k,\omega_{k,j}}|_{z=z_j} = W_{k,-\varepsilon x}$.

The value $k = 0$ is special. In this case, the module always has states of depth $p = 0$ and the action of the algebra involves the parameter $w$. In this special case, we have

$$e_2 \cdot v = \left[(x + x^{-1}) + \varepsilon(w + w^{-1})\right] \; {}_2\!\!\bigcirc\!\!{}_1 \; . \tag{108}$$

If $w \neq -\varepsilon x^{\pm 1}$, then $e_2 \cdot v$ is a nonzero element of the submodule $Y_{0,0,x,\varepsilon q,[z,w]}^{(0)}(2) \simeq W_{0,w}(2)$. For generic values of $w$, this module is irreducible, and hence the action of the algebra on $e_2 \cdot v$ generates $W_{0,w}(2)$. If instead $w = \varepsilon x^{\pm 1}$, then $e_2 \cdot v = 0$ and this submodule is not produced by the action of the algebra on $v$.

- States of type (iii-iv): These states are only present for $k \geqslant 1$. They have $2r$ defects attached to point $\mathsf{a}$, with $r < k$. These states are obtained by acting with the algebra on $e_{2k} \cdot v$ in such a way as to reduce the number of defects attached to $\mathsf{a}$. We therefore define the reduced intermediate seed states

$$\sigma_{k,1}^{(0,k-1)} = c_{2k-1} \cdot c_{2k} \cdot v, \tag{109a}$$

$$\sigma_{k,1}^{(0,r)} = (c_0)^{k-1-r} \cdot \sigma_{k,1}^{(0,k-1)}, \qquad r < k-1. \tag{109b}$$

Using the graphical rules (61), we readily find that $\sigma_{k,1}^{(0,r)}$ is proportional to the seed state $u_{r,0}(2r)$ with a nonzero coefficient. The action of the algebra thus generates all the composition factors of $M_{k,0,x,\varepsilon q,[z,w]}^{(0,r)}(2r)$ given in (85). Using the insertion algorithm, we find that the corresponding submodules of $Y_{k,0,x,\varepsilon q,[z,w]}(2k+2)$ that are generated from the action of the algebra on $e_{2k} \cdot v$ are $\bigoplus_{n=0}^{2r-1} W_{r,\omega_{r,n}}(2k+2)$ for $0 < r < k$ and $W_{0,w}(N)$ for $r = 0$.

We recall that all of these arguments are presented here for $N_{\mathsf{a}} = 2k$, $N_{\mathsf{b}} = 2$ and $N = 2k+2$. In Appendix B, we argue that the results in fact extend to arbitrary larger values of $N_{\mathsf{a}}$, $N_{\mathsf{b}}$ and $N$ satisfying $N = N_{\mathsf{a}} + N_{\mathsf{b}}$, which is thus consistent with Claim 4.2. To summarise, we have for generic values of $[z,w]$

$$\left(W_{k,x}(N_{\mathsf{a}}) \times R_{0,\varepsilon q}(N_{\mathsf{b}})\right)_{[z,w]} \simeq \begin{cases} Y_{0,0,x,\varepsilon q,[z,w]}(N)/W_{0,-\varepsilon x}(N), & k = 0, \, w = -\varepsilon x^{\pm 1}, \\ Y_{k,0,x,\varepsilon q,[z,w]}(N)/W_{k,-\varepsilon x}(N), & k \geqslant \frac{1}{2}, \, z^{2k} = (-\varepsilon x)^{2k}, \\ Y_{k,0,x,\varepsilon q,[z,w]}(N), & \text{otherwise.} \end{cases} \tag{110}$$

Importantly, we recall that $R_{0,\varepsilon q} \simeq W_{1,\varepsilon}$ and compare (110) with the fusion

$$\left(W_{k,x}(N_{\mathsf{a}}) \times W_{1,\varepsilon}(N_{\mathsf{b}})\right)_{[z,w]} \simeq Y_{k,1,x,\varepsilon,[z,w]}(N). \tag{111}$$

It is thus clear that $W_{k,x} \times R_{0,\varepsilon q}$ and $W_{k,x} \times W_{1,\varepsilon}$ yield different results in general, in particular because $\omega_{m,n}$ takes different values in $Y_{k,0,x,\varepsilon q,[z,w]}(N)$ and $Y_{k,1,x,\varepsilon,[z,w]}(N)$. This shows that two modules that are isomorphic do not necessarily behave identically under the fusion defined with the modules $Y_{k,\ell,x,y,[z,w]}(N)$.

**The fusion W × Q.**   The natural way to define the fusion modules of $W_{k,x}(N_a)$ and $Q_{0,\varepsilon q}(N_b)$ is to start from the module $Y_{k,0,x,\varepsilon q,[z,w]}(N)$ and include the extra quotient relation (35) to the diagrammatic rules (61) that dictate the action of $\mathcal{E}PTL_N(\beta)$ on this module. In this context, this extra relation allows us to commute the marked point b across any loop segment at the cost of a sign $-\varepsilon$. As we shall see, this has non-trivial consequences.

Let us first describe the case $k = 0$ where there are no defects at all, as this case is the simplest. We construct a basis of link states made of the subset of link states of $Y_{0,0,x,\varepsilon q,[z,w]}(N)$ for which the marked point b is adjacent to the perimeter of the disc, between the nodes $N$ and 1. In other words, this basis consists of link states with $r_b = 0$. There is an obvious bijection between the link states in this subset and the states in $B_0(N)$, from which we conclude that the dimension of this module is $\binom{N}{N/2}$.

The action of $\mathcal{E}PTL_N(\beta)$ on this basis may produce states with $r_b > 0$, in which case the quotient relation (35) for $Q_{0,\varepsilon q}(N_b)$ is used to commute the marked point across the crossing loop segments and write the result in terms of states with $r_b = 0$. The extra quotient relation then implies that

$$\alpha_{ab} = \phantom{x} \equiv -\varepsilon \phantom{x} = -\varepsilon\,\alpha_a. \tag{112}$$

Thus with $\alpha_{ab} = w + w^{-1}$ and $\alpha_a = x + x^{-1}$, this identity holds only if $w = -\varepsilon x^{\pm 1}$. For all other values of $w$, endowing $Y_{0,0,x,\varepsilon q,[z,w]}$ with the extra relation does not produce a representation of $\mathcal{E}PTL_N(\beta)$. We then say that the result of this fusion product vanishes.

We still need to understand the decomposition of these modules in the case where they are non-zero. To answer this question, we define a new basis where the two marked points are adjacent. All the states in the new basis have depth $p = 0$. The change of bases is diagonal and simply requires that we commute b across the loop segments for each state, leading to a multiplicative factor $\varepsilon^{r_b}$. In the resulting representation, all the non-contractible loops that are produced encircle both marked points and have the weight $\alpha = w + w^{-1} = \varepsilon(x + x^{-1})$. This is true for both $w = -\varepsilon x$ and $w = -\varepsilon x^{-1}$. The resulting fusion module is thus isomorphic to $W_{0,w}(N)$. In summary, we have

$$\left(W_{0,x}(N_a) \times Q_{0,\varepsilon q}(N_b)\right)_{[z,w]} \simeq \begin{cases} W_{0,w}(N), & w = -\varepsilon x^{\pm 1}, \\ 0, & \text{otherwise.} \end{cases} \tag{113}$$

It is clear that the same decomposition is obtained using the definition (95) and the first line of (110), thus confirming Claim 4.1 in this case. To understand why this is the case, let us note that the relation (112) can be written alternatively as

$$\left. \phantom{xxx} \right|_{\alpha_b = -\varepsilon\beta} = \left. \alpha_{ab} - \frac{\alpha_a \alpha_b}{\beta} \right|_{\alpha_b = -\varepsilon\beta} = (w + w^{-1}) + \varepsilon(x + x^{-1}) \equiv 0. \tag{114}$$

Comparing with (108), we see that the diagram on the left side of (114) is equal to $c_2 \cdot v$, which is the intermediate seed state for $W_{0,x} \times R_{0,\varepsilon q}$. Thus the module W × Q constructed diagrammatically with the extra quotient relation acting on the link states is precisely the module obtained by quotienting W × W by W × R, as defined in (95).

We now discuss the case $k > 0$. The extra quotient relation (35) again allows us to commute the marked point b across the loop segments up to signs. It allows us to express any link state in $Y_{k,0,x,\varepsilon q,[z,w]}(2k + 2)$ in terms of a link state with crossing number $r_b = 0$, up to some

sign. The quotient relation then implies that

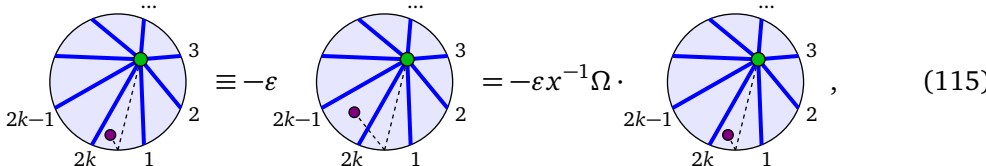 (115)

where for simplicity we have not drawn the extra spectator arc and the two nodes attached to it. Comparing with (104), we see that this relation can alternatively be written as

$$\sigma_{k,1}^{(0)} = (x^{-1}\Omega + \varepsilon 1) \cdot u_{k,0}(2k) \equiv 0. \tag{116}$$

The states $u_{k,0}(2k)$ and $\Omega \cdot u_{k,0}(2k)$ are distinct. Therefore for this equality to hold, the co-efficients of both of these states in (116) should be zero, however it is clear that they do not both vanish here. One then concludes that endowing the full representation $\mathsf{Y}_{k,0,x,\varepsilon q,[z,w]}$ with the extra quotient relation (35) never leads to a representation of the algebra $\mathcal{E}\mathsf{PTL}_N(\beta)$. To construct non-zero fusion products $\mathsf{W} \times \mathsf{Q}$ in this case, one must instead consider certain submodules of $\mathsf{Y}_{k,0,x,\varepsilon q,[z,w]}$ corresponding to specific eigenspaces of $F$ and endow these sub-modules with the extra quotient relation. It is however easier (yet equivalent) to perform this analysis on the quotient modules $\mathsf{M}_{k,0,x,\varepsilon q,[z,w]}^{(p)}$ and apply the projectors $\Pi_{k,n}$ associated to eigenspaces of $\Omega$. In the eigenspace corresponding to the projector $\Pi_{k,n}$, $\Omega$ acts as $\omega_{k,n}1$ and the relation (116) is replaced by

$$\Pi_{k,n} \cdot \sigma_{k,1}^{(0)} = (x^{-1}\omega_{k,n} + \varepsilon) u_{k,0}(2k) \equiv 0. \tag{117}$$

This is a single algebraic relation and it is satisfied for $z = z_n = -\varepsilon x\, e^{-i\pi n/k}$. The fusion module $\mathsf{W} \times \mathsf{Q}$ constructed in this way is thus non-zero for $z = z_n$, and it is clear from the construction that this module is precisely equal to the quotient of $\mathsf{W} \times \mathsf{W}$ by $\mathsf{W} \times \mathsf{R}$. The decomposition of $\mathsf{W} \times \mathsf{Q}$ can then be read off directly from (110). Using Claim 4.2 to extend the result to arbitrarily large values of $N_\mathsf{a}$ and $N_\mathsf{b}$, we have in summary

$$\left(\mathsf{W}_{k,x}(N_\mathsf{a}) \times \mathsf{Q}_{0,\varepsilon q}(N_\mathsf{b})\right)_{[z,w]} \simeq \begin{cases} \mathsf{W}_{0,-\varepsilon x}(N), & k = 0,\, w = -\varepsilon x^{\pm 1}, \\ \mathsf{W}_{k,-\varepsilon x}(N), & k \geqslant \frac{1}{2},\, z^{2k} = (-\varepsilon x)^{2k}, \\ 0, & \text{otherwise.} \end{cases} \tag{118}$$

Finally, the generic fusion rule reads

$$\mathsf{W}_{k,x} \times \mathsf{Q}_{0,\varepsilon q} \to \{\mathsf{W}_{k,-\varepsilon x}\}. \tag{119}$$

Thus we see that the vacuum module $\mathsf{V} = \mathsf{Q}_{0,-q}$ acts as the identity in the generic fusion rule, namely $\mathsf{W}_{k,x} \times \mathsf{V} \to \{\mathsf{W}_{k,x}\}$.

As will be clear in Section 4.4, the fusion modules for $\mathsf{W} \times \mathsf{Q}$ vanish for all $[z,w]$ except for a discrete set of values. The resulting fusion rule then involves only a finite set of modules. This is analogous to the fusion products in CFT involving one module whose null states are quotiented out. (The full analysis is given in Section 6.) In contrast, the fusion modules $\mathsf{W} \times \mathsf{W}$ obtained in (92) are always non-zero and the corresponding fusion rules involve a continuum of modules labeled by $[z,w]$.

## 4.3 A second example: the case $m = 2$

We set $m = 2$ for this entire section. Here, we only describe the fundamental case where $N_\mathsf{a} = 2k$, $N_\mathsf{b} = 4$ and $N = 2k + 4$ throughout. Following the ideas presented in Appendix B

for $m = 1$, it is not hard to show that the same constructions can be done for $m = 2$ with $N > 2k + 4$. This would prove Claim 4.2 for $m = 2$. However, we do not present this proof here and simply assume that Claim 4.2 holds.

We proceed as before, namely we first study the fusion $W \times R$ and subsequently $W \times Q$. Like in the previous section, there are two definitions for the fusion modules $W \times Q$. The first is obtained by applying the extra quotient relation (40) to a submodule of $Y_{k,0,x,\varepsilon q^2,[z,w]}$, and produces a representation only for certain values of $x$ and $[z, w]$. The second one defines $W \times Q$ directly as $(W \times W)/(W \times R)$. We assume that Claim 4.1 holds and thus take for granted that the two definitions are equivalent. Hereafter, we use the second definition to obtain the decomposition of $W \times Q$ directly.

The fusion module for $W_{k,x}(2k) \times R_{0,\varepsilon q^2}(4)$ is defined as

$$\left(W_{k,x}(2k) \times R_{0,\varepsilon q^2}(4)\right)_{[z,w]} = g_{[z,w]}\left(W_{k,x}(2k), R_{0,\varepsilon q^2}(4)\right). \tag{120}$$

It is thus the submodule of $Y_{k,0,x,\varepsilon q^2,[z,w]}(2k+4)$ defined from the action of $\mathcal{E}\mathrm{PTL}_N(\beta)$ on the seed state

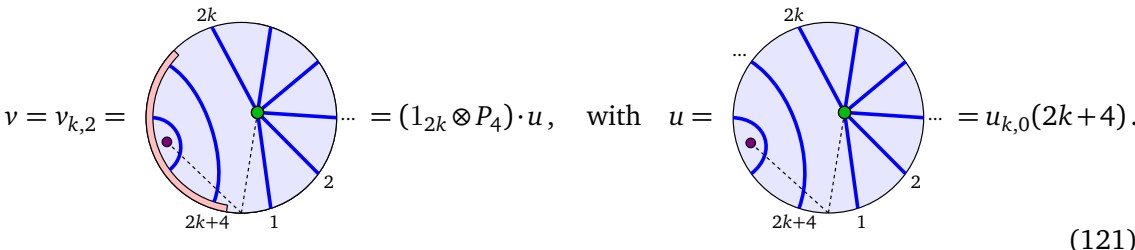

$$v = v_{k,2} = \quad = (1_{2k} \otimes P_4) \cdot u, \quad \text{with} \quad u = \quad = u_{k,0}(2k+4). \tag{121}$$

The decomposition of $Y_{k,0,x,\varepsilon q^2,[z,w]}$ is directly read off from (83) specialised to $\ell = 0$. To obtain the decomposition of $W \times R$, we must figure out which of the factors are part of the submodule. The arguments presented below are different for $k = 0$, $k = \frac{1}{2}$, $k = 1$ and $k \geq \frac{3}{2}$. We describe these four cases separately, and in each case we proceed like we did for $m = 1$: we act repeatedly on $v$ with the operators $e_j$, and examine whether nonzero states of types (i), (ii) and (iii-iv) can be produced.

**The case $k \geq \frac{3}{2}$.** In this case, the different types of states are produced as follows:

- States of type (i) with depth $p = 2$: These are the states $u, \Omega \cdot u, \ldots \Omega^{2k+3} \cdot u$. The projector $P_4$ is equal to $1$ plus some terms that involve the generators $e_j$, so as a result we have $v \equiv u_{k,0}(2k+4) \, [[1]]$. Using the same reasoning as for the states of maximal depth in the case $m = 1$, we readily deduce that all the modules $W_{k+2,\omega_{k+2,n}}$ with $n = 0, 1, \ldots, 2k+1$ are generated by the action of $\mathcal{E}\mathrm{PTL}_N(\beta)$ on $v$.

- States of type (i) with depth $p = 1$: The only generators that act on $v$ to produce a non-trivial state of depth $p = 1$ are $e_{2k}$ and $e_{2k+4}$. A simple computation shows that the resulting states $e_{2k} \cdot v$ and $e_{2k+4} \cdot v$ are proportional, and that the proportionality factor is non-zero. The corresponding intermediate reduced seed state is

$$\sigma_{k,2}^{(1)} = c_{2k} \cdot v. \tag{122}$$

Using the relation (27a) for the projector $P_4$, we find

$$\sigma_{k,2}^{(1)} = (1_{2k-1} \otimes P_3)(x^{-1}\Omega + \varepsilon 1) \cdot u_{k,0}(2k+2). \tag{123}$$

Using (27a) once more for $P_3$, we obtain

$$\sigma_{k,2}^{(1)} \equiv (x^{-1}\Omega + \varepsilon 1) \cdot u_{k,0}(2k+2) \, [[0]]. \tag{124}$$

To see which composition factors are produced, we apply the projectors onto eigenspaces of $\Omega$ and find

$$\Pi_{k+1,n} \cdot \sigma_{k,2}^{(1)} \equiv (x^{-1}\omega_{k+1,n} + \varepsilon)\,\Pi_{k+1,n} \cdot u_{k,0}(2k+2)\,[[0]], \quad \omega_{k+1,n} = z^{k/(k+1)}\,\mathrm{e}^{\mathrm{i}\pi n/(k+1)}, \quad (125)$$

with $n = 0, 1, \ldots, 2k+1$. For $z^{2k} \neq x^{2k+2}$, none of these states vanish. Using the insertion map and acting with the algebra on these non-zero states, we deduce that all the corresponding factors $\mathsf{W}_{k+1,\omega_{k+1,n}}$ appear in the decomposition of $\mathsf{W} \times \mathsf{R}$. In contrast, if $z$ is such that $z^{2k} = x^{2k+2}$, then exactly one of these states vanishes in $\mathsf{M}_{k,0,x,\varepsilon q^2,[z,w]}^{(1)}(2k+2)$. The corresponding value of $z$ is $z_j = (-\varepsilon x)^{(k+1)/k}\mathrm{e}^{-\mathrm{i}\pi j/k}$. In this case, we find that the factors $\mathsf{W}_{k+1,\omega_{k+1,n}}$ are all produced except the one corresponding to $n = j$. The corresponding missing factor is $\mathsf{W}_{k+1,\omega_{k+1,j}}|_{z=z_j} = \mathsf{W}_{k+1,-\varepsilon x}$.

- States of type (ii): These states have depth $p = 0$ and have $2k$ defects attached to a. The seed state for this sector is

$$\sigma_{k,2}^{(0)} = c_{2k-1}c_{2k} \cdot v = c_{2k-1} \cdot \sigma_{k,2}^{(1)}. \qquad (126)$$

Starting from the expression (123) and using the relation (27a) for $P_3$, we find

$$\sigma_{k,2}^{(0)} = (1_{2k-2} \otimes P_2)(x^{-1}\Omega + \varepsilon q 1)(x^{-1}\Omega + \varepsilon q^{-1}1) \cdot u_{k,0}(2k). \qquad (127)$$

Using the same argument as the one above for states of depth $p = 1$, we apply the projectors $\Pi_{k,n}$ to $\sigma_{k,2}^{(0)}$ and find that the result vanishes only for $\omega_{k,n} = -\varepsilon x q^{\pm 1}$. If $z$ is such that $\omega_{k,n} \neq -\varepsilon x q^{\pm 1}$, then all factors $\mathsf{W}_{k,\omega_{k,n}}$ with $n \in \{0, 1, \ldots, 2k-1\}$ are produced by the action of the algebra in the submodule generated by the seed states. If $z = z_j = -\varepsilon x q^{\pm 1}\mathrm{e}^{-\mathrm{i}\pi j/k}$, the submodule contains all factors $\mathsf{W}_{k,\omega_{k,n}}$ except the factor $\mathsf{W}_{k,\omega_{k,j}}|_{z=z_j} = \mathsf{W}_{k,-\varepsilon x q^{\pm 1}}$.

- States of type (iii): These states have $2k - 2$ defects attached to a. The corresponding seed state is given by

$$\sigma_{k,2}^{(0,k-1)} = c_{2k-2}c_{2k-1}c_{2k} \cdot v = c_{2k-2} \cdot \sigma_{k,2}^{(0)}. \qquad (128)$$

Applying $c_{2k-2}$ to (127), we obtain

$$\sigma_{k,2}^{(0,k-1)} = (x^{-1}\Omega + \varepsilon 1) \cdot u_{k-1,0}(2k-2). \qquad (129)$$

Applying $\Pi_{k-1,n}$ with $n = 0, 1, \ldots, 2k-3$, we find that the result is zero only if $\omega_{k-1,n} = -\varepsilon x$. Therefore, for $z = z_j = (-\varepsilon x)^{k/(k-1)}\mathrm{e}^{-\mathrm{i}\pi j/(k-1)}$, the submodule is made of all composition factors $\mathsf{W}_{k-1,\omega_{k-1,n}}$ except for the factor $\mathsf{W}_{k-1,\omega_{k-1,j}}|_{z=z_j} = \mathsf{W}_{k-1,-\varepsilon x}$. For the other values of $z$, all the composition factors $\mathsf{W}_{k-1,\omega_{k-1,n}}$ are produced.

- States of type (iii-iv): These states only exist for $k \geqslant 2$ and have $2r$ defects attached to a, with $r \leqslant k - 2$. The corresponding seed states are given by

$$\sigma_{k,2}^{(0,k-2)} = c_{2k-3}c_{2k-2}c_{2k-1}c_{2k} \cdot v = c_{2k-3} \cdot \sigma_{k,2}^{(0,k-1)}, \qquad (130a)$$

$$\sigma_{k,2}^{(0,r)} = (c_0)^{k-2-r} \cdot \sigma_{k,2}^{(0,k-2)}, \qquad r < k - 2. \qquad (130b)$$

Applying the rules (61) for the action of the algebra on $\mathsf{Y}_{k,0,x,\varepsilon q^2,[z,w]}$, we find

$$\sigma_{k,2}^{(0,r)} = u_{r,0}(2r), \qquad r \leqslant k - 2. \qquad (131)$$

Since these are the seed states of the modules $\mathsf{M}_{k,0,x,\varepsilon q^2,[z,w]}^{(0,r)}$, the entire corresponding module is generated by the action of the algebra on $v$, with all of its composition factors.

In summary, the fusion modules $W \times R$ decompose as

$$(W_{k,x} \times R_{0,\varepsilon q^2})_{[z,w]} \simeq \begin{cases} Y_{k,0,x,\varepsilon q^2,[z,w]}/W_{k\pm 1,-\varepsilon x}, & z^{2k} = (-\varepsilon x)^{2k\pm 2}, \\ Y_{k,0,x,\varepsilon q^2,[z,w]}/W_{k,-\varepsilon x q^{\pm 1}}, & z^{2k} = (-\varepsilon x q^{\pm 1})^{2k}, \\ Y_{k,0,x,\varepsilon q^2,[z,w]}, & \text{otherwise.} \end{cases} \tag{132}$$

Using (95), we directly deduce that the fusion modules $W \times Q$ decompose as

$$(W_{k,x} \times Q_{0,\varepsilon q^2})_{[z,w]} \simeq \begin{cases} W_{k\pm 1,-\varepsilon x}, & z^{2k} = (-\varepsilon x)^{2k\pm 2}, \\ W_{k,-\varepsilon x q^{\pm 1}}, & z^{2k} = (-\varepsilon x q^{\pm 1})^{2k}, \\ 0, & \text{otherwise.} \end{cases} \tag{133}$$

The decompositions (132) and (133) hold if the special values for $z^{2k}$ are distinct. Coincidences of these special values can only happen if $x^2 \in \{1, -1, q^{2k}, q^{-2k}\}$. If two of the special values coincide, the resulting module $W \times R$ is obtained by quotienting $Y_{k,0,x,\varepsilon q^2,[z,w]}$ by the two corresponding standard modules, and $W \times Q$ is the direct sum of these two standard modules. For instance, we have

$$(W_{k,q^k} \times Q_{0,\varepsilon q^2})_{[q^{k+1},w]} \simeq W_{k+1,-\varepsilon q^k} \oplus W_{k,-\varepsilon q^{k+1}}. \tag{134}$$

We conclude that, for $k \geqslant \frac{3}{2}$ and generic values of $x$, the decomposition (133) yields the generic fusion rules

$$W_{k,x} \times Q_{0,\varepsilon q^2} \to \{W_{k+1,-\varepsilon x}, W_{k,-\varepsilon x q}, W_{k,-\varepsilon x q^{-1}}, W_{k-1,-\varepsilon x}\}. \tag{135}$$

**The case $k = 1$.** Here, the analysis of states of types (i) and (ii) is the same as for $k \geqslant \frac{3}{2}$. Moreover, there are no states of type (iii). The states of (iv) exist only if $z^2 = 1$. In this case, the seed state reads

$$\sigma_{1,2}^{(0,0)} = c_0 \cdot \sigma_{1,2}^{(0)} = c_0 P_2 (x^{-1}\Omega + \varepsilon q 1)(x^{-1}\Omega + \varepsilon q^{-1} 1) \cdot u_{1,0}(2). \tag{136}$$

Using the fact that $\Omega^2 \cdot u_{1,0}(2) = u_{1,0}(2)$, we find

$$\sigma_{1,2}^{(0,0)} = (x + x^{-1}) + \varepsilon(w + w^{-1}). \tag{137}$$

Hence, the submodule $W_{0,w}$ is generated by the action of the algebra on $v$ and appears in the decomposition of the fusion product $W \times R$ only if $z^2 = 1$ and $w = -\varepsilon x^{\pm 1}$. Otherwise, it is absent. We conclude that the fusion module $(W_{1,x} \times Q_{0,\varepsilon q^2})_{[z,w]}$ decomposes as

$$(W_{1,x} \times Q_{0,\varepsilon q^2})_{[z,w]} \simeq \begin{cases} W_{2,-\varepsilon x}, & z^2 = (-\varepsilon x)^4, \\ W_{1,-\varepsilon x q^{\pm 1}}, & z^2 = (-\varepsilon x q^{\pm 1})^2, \\ W_{0,-\varepsilon x}, & z^2 = 1, w = -\varepsilon x^{\pm 1}, \\ 0, & \text{otherwise.} \end{cases} \tag{138}$$

This shows that the fusion rule (135) extends to $k = 1$.

**The case $k = \frac{1}{2}$.** In this case, there are only states of types (i) and (ii). The analysis of the states of type (i) is the same as for $k \geqslant \frac{3}{2}$. For states of type (ii), the seed state is

$$\sigma_{\frac{1}{2},2}^{(0)} = c_0 \cdot \sigma_{\frac{1}{2},2}^{(1)} = c_0 P_3 (x^{-1}\Omega + \varepsilon 1) \cdot u_{\frac{1}{2},0}(3). \tag{139}$$

We use the expression

$$P_3 = 1 + \frac{[2]}{[3]}(e_1 + e_2) + \frac{1}{[3]}(e_1 e_2 + e_2 e_1), \tag{140}$$

and find after some simplifications

$$\sigma^{(0)}_{\frac{1}{2},2} = (\varepsilon x z)^{-1}(z + \varepsilon q x)(z + \varepsilon q^{-1}x)(z + \varepsilon x^{-1})u_{\frac{1}{2},0}(1). \tag{141}$$

As before, the decompositions of the products $W \times R$ and $W \times Q$ follow directly. In particular, we have

$$(W_{\frac{1}{2},x} \times Q_{0,\varepsilon q^2})_z \simeq \begin{cases} W_{\frac{3}{2},-\varepsilon x}, & z = (-\varepsilon x)^3, \\ W_{\frac{1}{2},z}, & z \in \{-\varepsilon x q, -\varepsilon x q^{-1}, -\varepsilon x^{-1}\}, \\ 0, & \text{otherwise.} \end{cases} \tag{142}$$

Using the convention $W_{-k,z} = W_{k,z^{-1}}$, we see that the fusion rule (135) extends to $k = \frac{1}{2}$.

**The case $k = 0$.** Here, we compute separately the two non-trivial seed states, namely $\sigma^{(1)}_{0,2}$ and $\sigma^{(0)}_{0,2}$. An explicit calculation yields

$$\sigma^{(1)}_{0,2} = c_0 P_4 \cdot u_{0,0}(4) \equiv \left[ (x + x^{-1}) + \varepsilon(\Omega + \Omega^{-1}) \right] \cdot u_{0,0}(2) \, [[0]], \tag{143a}$$

$$\sigma^{(0)}_{0,2} = (c_0)^2 P_4 \cdot u_{0,0}(4) = (qwx)^{-2}(\varepsilon x + qw)(\varepsilon q x + w)(\varepsilon w x + q)(\varepsilon q w x + 1). \tag{143b}$$

The state $\Pi_{1,n} \cdot \sigma^{(1)}_{0,2}$ vanishes for $\omega_{1,n} = -\varepsilon x^{\pm 1}$. Because $\Omega^2$ acts in $Y_{0,0,x,\varepsilon q^2,[z,w]}(2)$ as the identity times a unit prefactor, this can only occur if $x^2 = 1$. Similarly, the seed state $\sigma^{(0)}_{0,2}$ vanishes for $w \in \{-\varepsilon x q^{\pm 1}, -\varepsilon x^{-1}q^{\pm 1}\}$. As a result, we obtain

$$(W_{0,x} \times Q_{0,\varepsilon q^2})_w \simeq \begin{cases} W_{0,-\varepsilon x q^{\pm 1}}, & w \in \{-\varepsilon x q^{\pm 1}, -\varepsilon x^{-1}q^{\pm 1}\}, \\ 0, & \text{otherwise,} \end{cases} \qquad x^2 \neq 1, \tag{144}$$

and

$$(W_{0,\varepsilon'} \times Q_{0,\varepsilon q^2})_w \simeq \begin{cases} W_{1,-\varepsilon\varepsilon'} \oplus W_{0,-\varepsilon\varepsilon'q}, & w = -\varepsilon\varepsilon' q^{\pm 1}, \\ W_{1,-\varepsilon\varepsilon'}, & \text{otherwise,} \end{cases} \qquad \varepsilon' \in \{+1,-1\}. \tag{145}$$

Thus we have the fusion rule

$$W_{0,x} \times Q_{0,\varepsilon q^2} \to \begin{cases} \{W_{0,-\varepsilon x q}, W_{0,-\varepsilon x q^{-1}}, W_{1,\varepsilon}, W_{1,-\varepsilon}\}, & x = \pm 1, \\ \{W_{0,-\varepsilon x q}, W_{0,-\varepsilon x q^{-1}}\}, & \text{otherwise.} \end{cases} \tag{146}$$

**Remarks on associativity.** We argue here the relevance of using the modules $Y_{k,\ell,x,y,[z,w]}$ instead of the modules $X_{k,\ell,x,y,z}$ to define the fusion of link states modules. With this in mind, let us denote the fusion of two modules $M$ and $N$ using the construction $X_{k,\ell,x,y,z}$ as $(M \times N)_X$. Repeating the arguments presented above, but using the modules $X_{k,\ell,x,y,z}$ instead, we find the fusion rules

$$(W_{k,x} \times W_{\ell,y})_X \to \{W_{m,t} \mid m + k + \ell \in \mathbb{Z}, m \geqslant |k - \ell|, W_{m,t} \text{ is irreducible}\}, \tag{147a}$$

$$(W_{k,x} \times Q_{0,\varepsilon q^2})_X \to \{W_{k+1,-\varepsilon x}, W_{k,-\varepsilon x q}, W_{k,-\varepsilon x q^{-1}}\}. \tag{147b}$$

The only difference between (135) and (147b) is that the module $W_{k-1,-\varepsilon x}$ is absent from the latter fusion rule. Moreover, we see that in the fusion of two standard modules, the integer $m$ has a lower bound that depends on $k$ and $\ell$.

To test whether the fusion rules associated to these two candidates can possibly be associative, it is then natural to assume that one can compute the triple fusion products $(M \times N) \times P$ and $M \times (N \times P)$ by first applying the fusion rules to fuse the pair of modules inside the parenthesis, and then by fusing each of the resulting composition factors with the third module.[7] Setting for simplicity $\varepsilon = -1$, we compute

$$\big((W_{k,x} \times Q_{0,-q^2})_\times \times W_{k+1,y}\big)_\times \to \big\{(W_{k+1,x} \times W_{k+1,y})_\times, (W_{k,xq} \times W_{k+1,y})_\times, (W_{k,xq^{-1}} \times W_{k+1,y})_\times\big\},$$

$$\big(W_{k,x} \times (Q_{0,-q^2} \times W_{k+1,y})_\times\big)_\times \to \big\{(W_{k,x} \times W_{k+2,y})_\times, (W_{k,x} \times W_{k+1,yq})_\times, (W_{k,x} \times W_{k+1,yq^{-1}})_\times\big\}. \quad (148)$$

Applying the fusion rules (147a) for $(W \times W)_\times$ to each element in the sets in the right-hand sides, we find that the first triple product allows for factors $W_{0,t}$, whereas the second does not. This shows that, under the above assumption for computing the fusion of three modules, the fusion $(M \times N)_\times$ cannot be associative. Repeating the calculation with the fusion built from the modules $Y_{k,\ell,x,y,[z,w]}$, we find that the same issue does not arise.

## 4.4 The general case $m \geqslant 1$

In this section, we investigate the fusion products $W_{k,x} \times R_{0,\varepsilon q^m}$ and $W_{k,x} \times Q_{0,\varepsilon q^m}$ for arbitrary values of $m$. We only present the arguments for $N_a = 2k$ and $N_b = 2m$, and assume that the decompositions of the module $W \times Q$ are identical for larger values, following Claim 4.2. The discussion below splits between four cases: $k > m-1$, $k = m-1$, $\frac{m}{2} \leqslant k < m-1$, and $0 \leqslant k < \frac{m}{2}$. Remarkably, the final result for the generic fusion rules $W \times Q$ is uniform over these cases and reads

$$W_{k,x} \times Q_{0,\varepsilon q^m} \to \Big\{W_{k+i-j,-\varepsilon x q^{i+j}} \,\Big|\, i,j \in \{-\tfrac{m-1}{2}, -\tfrac{m-3}{2}, \ldots, \tfrac{m-1}{2}\}\Big\}, \qquad k > 0, \quad (149)$$

where we recall that $W_{-k,x} = W_{k,x^{-1}}$ is used for the modules with negative defect numbers. The case $k = 0$ is special and discussed at the end of the section.

The strategy that we use is identical to the one presented previously for $m = 1$ and $m = 2$, namely we study the decomposition of the fusion products $W \times R$ and $W \times Q$ by evaluating the corresponding intermediate seed states $\sigma_{k,m}^{(p)}$ and $\sigma_{k,m}^{(0,r)}$. If such a state vanishes, then the corresponding composition factor is absent from the fusion product $W \times R$, but from (95) it is then present in $W \times Q$.

**The case $k > m-1$.** The set of intermediate seed states is $\{\sigma_{k,m}^{(p)}\}_{0 \leqslant p \leqslant m}$ and $\{\sigma_{k,m}^{(0,r)}\}_{r < k}$, obtained by acting with the operators $c_j$ on the seed state $v = v_{k,m}$ defined in (98). We now discuss the various types of states that can be produced.

- States of type (i) with depth $p = m$: In this case, using (27a) on $P_{2m}$, we readily find that the seed state $v$ satisfies $v \equiv u_{k,0}(2k+2m)\,[[m-1]]$. As a result, the projected states $\Pi_{k+m,n} \cdot v$ with $n = 0, 1, \ldots, 2k+2m-1$ are all non-zero. The action of the algebra on these states generates all the composition factors $W_{k+m,\omega_{k+m,n}}$ in $M_{k,0,x,\varepsilon q^m,[z,w]}^{(m)}$. This results lifts to $Y_{k,0,x,\varepsilon q^m,[z,w]}$ by noting that each state $Q_{k+m,n} \cdot v$ is also non-zero.

- States of type (i) with depths $1 \leqslant p \leqslant m-1$: These are generated by the intermediate seed states $\sigma_{k,m}^{(p)}$ defined in (100). These states admit a remarkably simple expression, given by the following proposition.

---

[7]Our construction of fusion is not guaranteed to satisfy this assumption, however fusion rules in CFT are expected to have this feature.

PROPOSITION 4.1 *The intermediate seed states are given by*

$$\sigma_{k,m}^{(p)} = (1_{2k-m+p} \otimes P_{m+p}) \left[ x^{p-m} \prod_{s=-\frac{m-p-1}{2}}^{\frac{m-p-1}{2}} (\Omega + \varepsilon x q^{2s} 1) \right] \cdot u_{k,0}(2k+2p). \tag{150}$$

PROOF. We first note that this result holds trivially for $p = m$. This will play the role of an initial condition in our inductive proofs on decreasing values of $p$. We introduce a set of auxiliary states

$$\sigma_{k,m,j}^{(p)} = (1_{2k-m+p} \otimes P_{m+p})(x^{-1}\Omega)^j \cdot u_{k,0}(2k+2p), \quad p = 1, 2, \ldots, m, \quad j = 0, 1, \ldots, m - p. \tag{151}$$

Using (27a) on $P_{m+p}$, we find that the action of $c_{2k-m+p}$ on $\sigma_{k,m,j}^{(p)}$ yields

$$c_{2k-m+p} \cdot \sigma_{k,m,j}^{(p)} = \frac{[2p+j]}{[m+p]} \sigma_{k,m,j+1}^{(p-1)} + \frac{\alpha_{\mathrm{b}}[p+j]}{[m+p]} \sigma_{k,m,j}^{(p-1)} + \frac{[j]}{[m+p]} \sigma_{k,m,j-1}^{(p-1)}, \tag{152}$$

for $j = 0, 1, \ldots, m - p$. This is a closed system of equations, namely it involves only the states defined in (151). We use it to show that $\sigma_{k,m}^{(p)}$ is of the form

$$\sigma_{k,m}^{(p)} = (1_{2k-m+p} \otimes P_{m+p}) A_{k,m}^{(p)}(\Omega) \cdot u_{k,0}(2k+2p), \tag{153}$$

where $A_{k,m}^{(p)}$ is a polynomial in $\Omega$ of degree $(m - p)$. In other words, (153) states that $\sigma_{k,m}^{(p)}$ is a linear combination of the auxiliary states $\sigma_{k,m,j}^{(p)}$. To prove this claim, we first remark that it holds trivially for $p = m$. Second, we assume that (153) holds for a given value of $p$. Because $\sigma_{k,m}^{(p-1)} = c_{2k-m+p} \cdot \sigma_{k,m}^{(p)}$, the relation (152) guarantees that (153) also holds for $p \to p - 1$, thus ending the inductive proof of this equation.

To determine $A_{k,m}^{(p)}(\Omega)$, we use (152) and find after some manipulations that it satisfies

$$A_{k,m}^{(p-1)}(\Omega) = \frac{x\,\Omega^{-1}}{q^{m+p} - q^{-m-p}} \left[ \varphi(q^p x^{-1}\Omega) A^{(p)}(q\,\Omega) - \varphi(q^{-p}x^{-1}\Omega) A^{(p)}(q^{-1}\Omega) \right], \tag{154}$$

where $\varphi(\omega) = \omega^2 + \alpha_{\mathrm{b}}\omega + 1$. For $\alpha_{\mathrm{b}} = \varepsilon(q^m + q^{-m})$,

$$A_{k,m}^{(p)}(\Omega) = x^{p-m} \prod_{s=-\frac{m-p-1}{2}}^{\frac{m-p-1}{2}} (\Omega + \varepsilon x q^{2s} 1), \tag{155}$$

is the unique solution of this polynomial recursion, satisfying the initial condition $A_{k,m}^{(m)}(\Omega) = 1$ and proved inductively for $p < m$. ∎

From (150), we have

$$\sigma_{k,m}^{(p)} \equiv \left[ x^{p-m} \prod_{s=-\frac{m-p-1}{2}}^{\frac{m-p-1}{2}} (\Omega + \varepsilon x q^{2s} 1) \right] \cdot u_{k,0}(2k+2p) \, [[p-1]]. \tag{156}$$

As a result, the state $\Pi_{k+p,n} \cdot \sigma_{k,m}^{(p)}$ vanishes in $\mathsf{M}_{k,0,x,\varepsilon q^m,[z,w]}^{(p)}$ if and only if $\omega_{k+p,n} = -\varepsilon x q^{2s}$, with $s \in \{-\frac{m-p-1}{2}, -\frac{m-p-3}{2}, \ldots, \frac{m-p-1}{2}\}$.

- States of type (ii): These states have depth $p = 0$ and $2k$ defects attached to a. The above derivation of $\sigma_{k,m}^{(p)}$ readily extends to $p = 0$ and yields

$$\sigma_{k,m}^{(0)} = (1_{2k-m} \otimes P_m) \left[ x^{-m} \prod_{s=-\frac{m-1}{2}}^{\frac{m-1}{2}} (\Omega + \varepsilon x q^{2s} 1) \right] \cdot u_{k,0}(2k). \tag{157}$$

The state $\Pi_{k,n} \cdot \sigma_{k,m}^{(0)}$ therefore vanishes in $\mathsf{M}_{k,0,x,\varepsilon q^m,[z,w]}^{(0)}$ if and only if $\omega_{k,n} = -\varepsilon x q^{2s}$, with $s \in \{-\frac{m-1}{2}, -\frac{m-3}{2}, \ldots, \frac{m-1}{2}\}$.

- States of type (iii): These states have $2r$ defects attached to a, with $k - m \leqslant r < k$. The corresponding intermediate seed states are defined as

$$\sigma_{k,m}^{(0,r)} = c_{k-m+r+1} \ldots c_{2k-1} c_{2k} \cdot v_{k,m}. \tag{158}$$

In order to give simple expressions for these states, we follow the same arguments as those used in the proof of Proposition 4.1. First, we define the auxiliary states

$$\sigma_{k,m,j}^{(0,r)} = (1_{k-m+r} \otimes P_{m-k+r}) \cdot (x^{-1} \Omega)^j \cdot u_{r,0}(2r), \qquad j = 0, 1, \ldots, m-k+r-1. \tag{159}$$

Second, using (27a) for the Jones-Wenzl projector, we find the simple relation

$$c_{k-m+r} \cdot \sigma_{k,m,j}^{(0,r)} = \frac{[j]}{[m-k+r]} \sigma_{k,m,j-1}^{(0,r-1)}, \qquad r = k-m+1, k-m+2, \ldots, k. \tag{160}$$

For $r = k$, this holds with the identification $\sigma_{k,m,j}^{(0,k)} = \sigma_{k,m,j}^{(0)}$. Moreover, because $\sigma_{k,m}^{(0,k-1)} = c_{2k-m} \cdot \sigma_{k,m}^{(0)}$, the initial condition for the induction argument below is the expression (157) for $\sigma_{k,m}^{(0)}$. Third, we use (160) to prove that

$$\sigma_{k,m}^{(0,r)} = (1_{k-m+r} \otimes P_{m-k+r}) A_{k,m}^{(0,r)}(\Omega) \cdot u_{r,0}(2r), \tag{161a}$$

$$A_{k,m}^{(0,r-1)}(\Omega) = \frac{x \Omega^{-1}}{q^{m-k+r} - q^{-m+k-r}} \left[ A_{k,m}^{(0,r)}(q \Omega) - A_{k,m}^{(0,r)}(q^{-1} \Omega) \right], \tag{161b}$$

where $A_{k,m}^{(0,r-1)}(\Omega)$ is a polynomial in $\Omega$ of degree $m - k + r$. Fourth, we prove inductively on decreasing values of $r$ that the solution is

$$\sigma_{k,m}^{(0,r)} = (1_{k-m+r} \otimes P_{m-k+r}) \left[ x^{-m+k-r} \prod_{s=-\frac{m-k+r-1}{2}}^{\frac{m-k+r-1}{2}} (\Omega + \varepsilon x q^{2s} 1) \right] \cdot u_{r,0}(2r). \tag{162}$$

We conclude that the state $\Pi_{r,n} \cdot \sigma_{k,m}^{(0,r)}$ vanishes in $\mathsf{M}_{k,0,x,\varepsilon q^m,[z,w]}^{(0,r)}$ if and only if $\omega_{r,n} = -\varepsilon x q^{2s}$, with $s \in \{-\frac{m-k+r-1}{2}, -\frac{m-k+r-3}{2}, \ldots, \frac{m-k+r-1}{2}\}$.

- States of type (iii-iv): These states exist only for $k \geqslant m$ and have $2r$ defects attached to a, with $r \leqslant k - m$. These are generated by the intermediate seed states

$$\sigma_{k,m}^{(0,k-m)} = c_{2k-2m+1} c_{2k-2m+2} \ldots c_{2k} \cdot v, \tag{163a}$$

$$\sigma_{k,m}^{(0,r)} = (c_0)^{k-m-r} \cdot \sigma_{k,m}^{(0,k-m)}, \qquad r < k-m. \tag{163b}$$

From (162), we have $\sigma_{k,m}^{(0,k-m)} = u_{k-m,0}(2k-2m)$. Then, using the graphical rules (61), we directly get

$$\sigma_{k,m}^{(0,r)} = u_{r,0}(2r), \qquad r \leqslant k-m. \tag{164}$$

These are the seed states for the quotient modules $\mathsf{M}_{k,0,x,\varepsilon q^m,[z,w]}^{(0,r)}$, and thus all the corresponding composition factors are generated by the action of the algebra on $v$.

The fusion modules $W \times R$ are the direct sums of all standard modules appearing in the decomposition (83) of $W \times W$, except those for which the corresponding seed states $\Pi_{k+p,n} \cdot \sigma_{k,m}^{(p)}$ or $\Pi_{r,n} \cdot \sigma_{k,m}^{(0,r)}$ vanish. In contrast, from (95), the fusion modules $W \times Q$ generally vanish, except in the cases where some seed states vanish, in which case they inherit the corresponding composition factors. The above results are then conveniently summarised as

$$(W_{k,x} \times Q_{0,\varepsilon q^m})_{[z,w]} \simeq \begin{cases} W_{k+p,-\varepsilon xq^{2s}} & \begin{cases} z^{2k} = (-\varepsilon xq^{2s})^{2k+2p}, \\ p \in \{-m+1, -m+2, \ldots, m-1\}, \\ s \in \{-\frac{m-|p|-1}{2}, -\frac{m-|p|-3}{2}, \ldots, \frac{m-|p|-1}{2}\}, \end{cases} \\ 0, & \text{otherwise.} \end{cases}$$
(165)

Hence, the allowed generic fusion rules are

$$W_{k,x} \times Q_{0,\varepsilon q^m} \to \left\{ W_{k+p,-\varepsilon xq^{2s}} \ \middle| \ \begin{array}{l} p \in \{-m+1, -m+2, \ldots, m-1\} \\ s \in \{-\frac{m-|p|-1}{2}, -\frac{m-|p|-3}{2}, \ldots, \frac{m-|p|-1}{2}\} \end{array} \right\}.$$
(166)

Changing variables to $i = s + \frac{p}{2}$ and $j = s - \frac{p}{2}$, we obtain (149).

**The case $k = m-1$.** The derivation is identical to the case $k > m-1$, except for states of type (iv). For $z^{2k} \neq 1$, there are no states of type (iv) in the basis of $Y_{k,0,x,\varepsilon q^m,[z,w]}$, so let us discuss the case $z^{2k} = 1$. From (162), we have

$$\sigma_{m-1,m}^{(0,1)} = P_2(x^{-1}\Omega + \varepsilon q1)(x^{-1}\Omega + \varepsilon q^{-1}1) \cdot u_{1,0}(2)$$

$$= \left(x^{-2}1 + \varepsilon(q+q^{-1})x^{-1}\Omega + 1\right) \cdot u_{1,0}(2) + \varepsilon \ {}_2\!\left(\vcenter{\hbox{\includegraphics{sphere}}}\right)\!{}_1 \ .$$
(167)

Hence, the action of $c_0$ yields

$$\sigma_{m-1,m}^{(0,0)} = c_0 \cdot \sigma_{m-1,m}^{(0,1)} = (x + x^{-1}) + \varepsilon(w + w^{-1}),$$
(168)

which vanishes for $w = -\varepsilon x^{\pm 1}$. As a result, the decomposition of the fusion module $W \times Q$ is

$$(W_{m-1,x} \times Q_{0,\varepsilon q^m})_{[z,w]} \simeq \begin{cases} W_{m-1+p,-\varepsilon xq^{2s}} & \begin{cases} z^{2m-2} = (-\varepsilon xq^{2s})^{2m+2p-2}, \\ p \in \{-m+2, -m+3, \ldots, m-1\}, \\ s \in \{-\frac{m-|p|-1}{2}, -\frac{m-|p|-3}{2}, \ldots, \frac{m-|p|-1}{2}\}, \end{cases} \\ W_{0,-\varepsilon x}, & z^{2m-2} = 1, w = -\varepsilon x^{\pm 1}, \\ 0, & \text{otherwise.} \end{cases}$$
(169)

We conclude that the fusion rule (149) also applies to $k = m-1$.

**The case $\frac{m}{2} \leqslant k < m-1$.** In this case, the intermediate seed states are

$$\sigma_{k,m}^{(p)} = c_{2k-m+p+1} c_{2k-m+p+2} \cdots c_{2k} \cdot v, \qquad 0 \leqslant p \leqslant m, \tag{170a}$$

$$\sigma_{k,m}^{(0,r)} = c_{k+r-m+1} c_{k+r-m+2} \cdots c_{2k} \cdot v, \qquad m-k \leqslant r \leqslant k, \tag{170b}$$

$$\sigma_{k,m}^{(0,r)} = (c_0)^{m-k-r} \cdot \sigma_{k,m}^{(0,m-k)}, \qquad 0 \leqslant r < m-k. \tag{170c}$$

The first two sequences of intermediate seed states are evaluated using the same polynomial recursions as in the case $k \geqslant m-1$. The results are given by (150) and (162), respectively. The calculation of the states $\sigma_{0,m}^{(0,r)}$ with $0 \leqslant r < m-k$ cannot be done as before by deriving recursion relations where $r$ varies. It is however possible to derive such relations where $m$ varies, similar to those presented for the states $\sigma_{0,m}^{(p)}$ in Appendix C. Here, we simply conjecture the results.

CONJECTURE 4.1 *The intermediate seed states satisfy*

$$\sigma_{k,m}^{(0,r)} \equiv x^{k-m-r}\, \Omega^{k-m+r} \left[ \prod_{s=-\frac{m-k+r-1}{2}}^{\frac{m-k+r-1}{2}} (\Omega + x\varepsilon q^{2s}1) \right] \left[ \prod_{s=-\frac{m-k-r-1}{2}}^{\frac{m-k-r-1}{2}} (\varepsilon x\Omega + q^{2s}1) \right] \cdot u_{r,0}(2r)\,[[0,r-1]], \quad (171)$$

*for* $0 < r < m-k$, *and*

$$\sigma_{k,m}^{(0,0)} = (xw)^{k-m} \prod_{s=-\frac{m-k-1}{2}}^{\frac{m-k-1}{2}} (w + \varepsilon x q^{2s})(\varepsilon x w + q^{2s}). \quad (172)$$

It follows from these conjectures that the fusion modules $W \times Q$ decompose as

$$(W_{k,x} \times Q_{0,\varepsilon q^m})_{[z,w]} \simeq \begin{cases} W_{k+p,-\varepsilon x q^{2s}} & \begin{cases} z^{2k} = (-\varepsilon x q^{2s})^{2k+2p}, \\ p \in \{-k+1, -k+2, \ldots, m-1\}, \\ s \in \{-\dfrac{m-|p|-1}{2}, -\dfrac{m-|p|-3}{2}, \ldots, \dfrac{m-|p|-1}{2}\}, \end{cases} \\ W_{k+p,-\varepsilon x^{-1} q^{2s}} & \begin{cases} z^{2k} = (-\varepsilon x^{-1} q^{2s})^{2k+2p}, \\ p \in \{-k+1, -k+2, \ldots, m-2k-1\}, \\ s \in \{-\dfrac{m-2k-p-1}{2}, -\dfrac{m-2k-p-3}{2}, \ldots, \dfrac{m-2k-p-1}{2}\}, \end{cases} \\ W_{0,-\varepsilon x q^{2s}} & \begin{cases} z^{2k} = 1,\ w = (-\varepsilon x q^{2s})^{\pm 1}, \\ s \in \{-\dfrac{m-k-1}{2}, -\dfrac{m-k-3}{2}, \ldots, \dfrac{m-k-1}{2}\}, \end{cases} \\ 0, & \text{otherwise.} \end{cases} \quad (173)$$

Using the convention $W_{-k,x} = W_{k,x^{-1}}$, the second line is equivalently written as

$$(W_{k,x} \times Q_{0,\varepsilon q^m})_{[z,w]} \simeq W_{k+p,-\varepsilon x q^{2s}}, \quad \text{for} \quad \begin{cases} z^{2k} = (-\varepsilon x^{-1} q^{2s})^{2k+2p}, \\ p \in \{-m+1, -m+2, \ldots, -k-1\}, \\ s \in \{-\frac{m-|p|-1}{2}, -\frac{m-|p|-3}{2}, \ldots, \frac{m-|p|-1}{2}\}, \end{cases} \quad (174)$$

where we operated the change of variables $(p,s) \to (-2k-p, -s)$. Hence, we find that the fusion rule (149) also holds for $\frac{m}{2} \leqslant k < m-1$.

**The case $0 \leqslant k < \frac{m}{2}$.** Here, the seed states are

$$\sigma_{k,m}^{(p)} = c_{2k-m+p+1} \ldots c_{2k} \cdot v, \qquad m-2k \leqslant p \leqslant m, \quad (175a)$$

$$\sigma_{k,m}^{(p)} = (c_0)^{p-m+2k} \cdot \sigma_{k,m}^{(m-2k)}, \qquad 0 \leqslant p < m-2k, \quad (175b)$$

$$\sigma_{k,m}^{(0,r)} = (c_0)^{k-r} \cdot \sigma_{k,m}^{(0)}, \qquad 0 \leqslant r \leqslant k, \quad (175c)$$

where we note that $\sigma_{k,m}^{(0)} = \sigma_{k,m}^{(0,k)}$. The first set obeys the same polynomial recursion as in the case $k \geqslant m-1$, with the solution given in (150). We again conjecture the results.

CONJECTURE 4.2 *The intermediate seed states satisfy*

$$\sigma_{k,m}^{(p)} \equiv x^{p-m}\, \Omega^{2k+p-m} \left[ \prod_{s=-\frac{m-p-1}{2}}^{\frac{m-p-1}{2}} (\Omega + \varepsilon x q^{2s}1) \right] \left[ \prod_{s=-\frac{m-2k-p-1}{2}}^{\frac{m-2k-p-1}{2}} (\varepsilon x\Omega + q^{2s}1) \right] \cdot u_{k,0}(2k+2p)\,[[p-1]],$$

$$(176a)$$

*for $0 < p < m - 2k$,*

$$\sigma_{k,m}^{(0,r)} \equiv x^{k-m-r}\,\Omega^{k-m+r} \left[ \prod_{s=-\frac{m-k+r-1}{2}}^{\frac{m-k+r-1}{2}} (\Omega + x\varepsilon q^{2s}1) \right] \left[ \prod_{s=-\frac{m-k-r-1}{2}}^{\frac{m-k-r-1}{2}} (\varepsilon x\Omega + q^{2s}1) \right] \cdot u_{r,0}(2r)\,[[0, r-1]], \tag{176b}$$

*for $0 < r \leqslant k$, and*

$$\sigma_{k,m}^{(0,0)} = (xw)^{k-m} \prod_{s=-\frac{m-k-1}{2}}^{\frac{m-k-1}{2}} (w + \varepsilon x q^{2s})(\varepsilon xw + q^{2s}). \tag{176c}$$

In Appendix C, we derive a set of recursion relations for these expressions for the special case $k = 0$. For $0 < k < \frac{m}{2}$, the above results yield exactly the same decomposition as the one given in (173). Thus in this case, the fusion rule (149) also applies.

For $k = 0$, the module does not depend on the parameter $z$. The decomposition of the fusion modules depends on the value of $x$, and is different if $x$ is one of the special values in the set

$$\mathcal{X} = \left\{ q^{2t}\,\omega_{r,n} \,\middle|\, r \in \{1, 2, \ldots, m-1\},\, n \in \{0, 1, \ldots, 2r-1\},\, t \in \{-\tfrac{m-r-1}{2}, -\tfrac{m-r-3}{2}, \ldots, \tfrac{m-r-1}{2}\} \right\}, \tag{177}$$

where we recall that $\omega_{r,n} = e^{i\pi n/r}$ in this case. For $x \notin \mathcal{X}$, the decomposition of the fusion module $W \times Q$ is

$$(W_{0,x} \times Q_{0,\varepsilon q^m})_w \simeq \begin{cases} W_{0,w} & \begin{cases} w^{\pm 1} = -\varepsilon x q^{2s}, \\ s \in \{-\tfrac{m-1}{2}, -\tfrac{m-3}{2}, \ldots, \tfrac{m-1}{2}\}, \end{cases} \\ 0 & \text{otherwise.} \end{cases} \tag{178}$$

For $x \in \mathcal{X}$, some of the factors in (176a) vanish. There may in fact be more than one such vanishing factors. For each $x = q^{2t}\omega_{r,n}$, we define the set of standard modules

$$\mathcal{E}_{r,n} = \left\{ W_{p,\omega_{p,n'}} \,\middle|\, \omega_{p,n'} = -\varepsilon\,\omega_{r,\pm n},\, p \in \{1, 2, \ldots, m-2|t|-1\},\, n' \in \{0, 1, \ldots, 2p-1\} \right\}. \tag{179}$$

The values $\omega_{p,n'}$ are those for which $\Pi_{p,n'} \cdot \sigma_{0,m}^{(p)} \equiv 0\,[[p-1]]$ for $x = q^{2t}\omega_{r,n}$. We then have

$$(W_{0,q^{2t}\omega_{r,n}} \times Q_{0,\varepsilon q^m})_w \simeq \begin{cases} W_{0,w} \oplus \bigoplus_{W \in \mathcal{E}_{r,n}} W & \begin{cases} w^{\pm 1} = -\varepsilon q^{2t+2s}\,\omega_{r,n}, \\ s \in \{-\tfrac{m-1}{2}, -\tfrac{m-3}{2}, \ldots, \tfrac{m-1}{2}\}, \end{cases} \\ \bigoplus_{W \in \mathcal{E}_{r,n}} W, & \text{otherwise.} \end{cases} \tag{180}$$

Thus, for the values of $x$ for which $W_{0,x}$ is irreducible, the fusion rule is

$$W_{0,x} \times Q_{0,\varepsilon q^m} \to \left\{ W_{0,-\varepsilon x q^{2s}} \,\middle|\, s \in \{-\tfrac{m-1}{2}, -\tfrac{m-3}{2}, \ldots, \tfrac{m-1}{2}\} \right\}. \tag{181}$$

This holds for both $x \in \mathcal{X}$ and $x \notin \mathcal{X}$. For $x \in \mathcal{X}$, there is the additional fusion rule

$$W_{0,q^{2t}\omega_{r,n}} \times Q_{0,\varepsilon q^m} \to \left\{ W_{p,\omega_{p,n'}} \,\middle|\, p \in \{1, 2, \ldots, m-2|t|-1\},\, \omega_{p,n'} = -\varepsilon\,\omega_{r,\pm n} \right\}. \tag{182}$$

# 5   The fusion modules of $\mathbf{Q}_{0,\varepsilon_a q^k} \times \mathbf{Q}_{0,\varepsilon_b q^\ell}$

The objective of this section is to study the fusion modules $\left(Q_{0,\varepsilon_a q^k} \times Q_{0,\varepsilon_b q^\ell}\right)_{[z,w]}$ for $k,\ell \in \mathbb{Z}_{\geqslant 1}$ and determine their structures. We will see that the natural diagrammatic definition of this fusion module is equivalent to the quotient of $Y_{0,0,\varepsilon_a q^k,\varepsilon_b q^\ell,[z,w]}$ by the sum of the two submodules $\left(W_{0,\varepsilon_a q^k} \times R_{0,\varepsilon_b q^\ell}\right)_{[z,w]}$ and $\left(R_{0,\varepsilon_a q^k} \times W_{0,\varepsilon_b q^\ell}\right)_{[z,w]}$. The fusion module $Q \times Q$ turns out to be a non-zero modules only for $w = -\varepsilon_a \varepsilon_b q^m$, where $m$ is an integer in a specific interval. These are non-generic values of $w$. Hence, we start in Section 5.1 by studying the decomposition of $Y = W \times W$ for non-generic $[z,w]$. Then in Section 5.2, we determine the structure of the submodule $W \times R$, for both generic and non-generic value of $[z,w]$. Finally, we study the fusion module $Q \times Q$ in Section 5.3.

## 5.1   The decomposition of $\mathbf{W}_{0,x} \times \mathbf{W}_{0,y}$ for non-generic $w$

We first recall that the module $Y_{0,0,x,y,[z,w]}$ is independent of $z$ and satisfies $Y_{k,\ell,x,y,[z,w]} = Y_{k,\ell,x,y,[z,w^{-1}]}$. As discussed in Section 3.3, the non-generic values of $w$ for $Y_{0,0,x,y,[z,w]}$ are of the form $w^{\pm 1} = \varepsilon q^m$ with $m \in \{1,2,\dots,\frac{N}{2}\}$. As stated in (64), the modules $Y_{0,0,x,y,[z,w]}$ and $X_{0,0,x,y,w}$ are identical. The decomposition of the latter for non-generic $w$ was studied in [13]. The coincidence of $F$ eigenvalues occurs between the standard modules $W_{0,w}$ and $W_{m,\varepsilon}$, with

$$f_0 = f_{m,n_\varepsilon} = \varepsilon(q^m + q^{-m}), \qquad n_\varepsilon = \left\{ \begin{array}{ll} 0, & \varepsilon = 1, \\ m, & \varepsilon = -1. \end{array} \right. \tag{183}$$

The module $Y_{0,0,x,y,[z,w=\varepsilon q^m]}$ thus decomposes as

$$Y_{0,0,x,y,[z,w=\varepsilon q^m]} \simeq Z_{0,0,x,y,[z,w=\varepsilon q^m]} \oplus \bigoplus_{\substack{r=1 \\ (r,n)\neq(m,n_\varepsilon)}}^{N/2} \bigoplus_{n=0}^{2r-1} W_{r,\omega_{r,n}}, \tag{184}$$

where $Z_{0,0,x,y,[z,w=\varepsilon q^m]}$ is the kernel of $(F - f_0 1)^2$ in $Y_{0,0,x,y,[z,w=\varepsilon q^m]}$. Here we make a stronger statement than the one presented in [13] for the structure of $Z_{0,0,x,y,[z,w=\varepsilon q^m]}$. For $N \geqslant 2m$, the submodule $Z_{0,0,x,y,[z,w=\varepsilon q^m]}$ has the structure

$$Z_{0,0,x,y,[z,w=\varepsilon q^m]} \simeq \left\{ \begin{array}{ll} \left[ \begin{array}{c} Q_{0,\varepsilon q^m} \\ \qquad\searrow \\ \qquad\qquad R_{0,\varepsilon q^m} \end{array} \right] \oplus W_{m,\varepsilon} & \left\{ \begin{array}{l} y = -\varepsilon x^{\pm 1} q^{2s}, \\ s \in \{-\frac{m-1}{2}, -\frac{m-3}{2}, \dots, \frac{m-1}{2}\}, \end{array} \right. \\[2em] \left[ \begin{array}{c} W_{m,\varepsilon} \\ \swarrow \\ Q_{0,\varepsilon q^m} \\ \qquad\searrow \\ \qquad\qquad R_{0,\varepsilon q^m} \end{array} \right] & \text{otherwise.} \end{array} \right. \tag{185}$$

The proof is given in Appendix D. Only the bottom case was studied in [13].

With this information, we are now able to state the *full* fusion rules for $W_{0,x} \times W_{0,y}$. For

convenience, let us define

$$P_{m,\varepsilon} = \begin{bmatrix} & W_{m,\varepsilon} \\ Q_{0,\varepsilon q^m} & \\ & R_{0,\varepsilon q^m} \end{bmatrix}. \tag{186}$$

For $y$ not of the form $y = -\varepsilon x^{\pm 1} q^k$ with $\varepsilon \in \{-1, +1\}$ and $k \in \mathbb{Z}$, we have

$$W_{0,x} \times W_{0,y} \to \{W_{m,t} \mid m \in \mathbb{Z}_{\geqslant 0}, \, t^{2m} = 1, \, W_{m,t} \text{ is irreducible}\}$$
$$\cup \{P_{m,\varepsilon'} \mid m \in \mathbb{Z}_{>0}, \, \varepsilon' \in \{-1, +1\}\}. \tag{187}$$

For $y = -\varepsilon x^{\pm 1} q^k$ with $\varepsilon \in \{-1, +1\}$ and $k \in \mathbb{Z}$, we instead have

$$W_{0,x} \times W_{0,y} \to \{W_{m,t} \mid m \in \mathbb{Z}_{\geqslant 0}, \, t^{2m} = 1, \, W_{m,t} \text{ is irreducible}\}$$
$$\cup \{W_{0,\varepsilon q^m} \mid m \in |k| + 1 + 2\mathbb{Z}_{\geqslant 0}\}$$
$$\cup \{P_{m,\varepsilon} \mid m \in \mathbb{Z}_{>0} \setminus (|k| + 1 + 2\mathbb{Z}_{\geqslant 0})\}. \tag{188}$$

## 5.2 The fusion modules of $W_{0,\varepsilon_a q^k} \times R_{0,\varepsilon_b q^\ell}$ for generic and non-generic $w$

Let $k, \ell \in \mathbb{Z}_{\geqslant 1}$ and $\varepsilon_a, \varepsilon_b \in \{-1, +1\}$. The fusion modules of $W \times R$ are defined in (93) using the gluing operation as

$$\left(W_{0,\varepsilon_a q^k}(N_a) \times R_{0,\varepsilon_b q^\ell}(N_b)\right)_{[z,w]} = g_{[z,w]}\left(W_{0,\varepsilon_a q^k}(N_a), R_{0,\varepsilon_b q^\ell}(N_b)\right). \tag{189}$$

These are modules over $\mathcal{E}\mathsf{PTL}_N(\beta)$ with $N = N_a + N_b$. We now describe the decomposition of the above modules, following the same arguments as those used in Section 4.4. The structure of the module $W \times R$ depends on two factors: (i) the vanishing of the intermediate seed states, and (ii) the decomposition of $Y_{0,0,\varepsilon_a q^k, \varepsilon_b q^\ell, [z,w]}$. The analysis detailed below is divided between certain cases depending on the value taken by $w$ (but not $z$, as this parameter does not arise in this module). It is convenient to introduce the notation

$$\varepsilon_{ab} = -\varepsilon_a \varepsilon_b. \tag{190}$$

For point (i), applying the same idea that led to (180), we find from (176) that the intermediate seed states satisfy

$$\sigma_{0,\ell}^{(0,0)} = 0 \quad \Longleftrightarrow \quad w^{\pm 1} = \varepsilon_{ab} q^m, \quad \text{with} \quad m \in \{k - \ell + 1, k - \ell + 3, \dots, k + \ell - 1\}, \tag{191a}$$

$$\Pi_{r,n} \cdot \sigma_{0,\ell}^{(r)} \equiv 0 \, [[r-1]] \quad \Longleftrightarrow \quad \varepsilon_{ab} = \exp(i\pi n/r), \quad \text{with} \quad \begin{cases} 1 \leqslant r \leqslant \ell - k - 1, \\ r + k + \ell \in 2\mathbb{Z} + 1. \end{cases} \tag{191b}$$

The latter holds for all values of $w$. This means that for all values of $w$ (and thus in all the cases examined below), certain standard modules are absent from the decomposition of the fusion module $W \times R$ if the conditions in (191b) are satisfied. We use the notation

$$\widetilde{\bigoplus_{r,n}} W_{r,\omega_{r,n}} = \begin{cases} \displaystyle\bigoplus_{\substack{r=1 \\ W_{r,\omega_{r,n}} \notin \{W_{s,\varepsilon_{ab}} \mid s \leqslant \ell-k-1, s+k+\ell \in 2\mathbb{Z}+1\}}}^{N/2} \bigoplus_{n=0}^{2r-1} W_{r,\omega_{r,n}}, & \ell > k+1, \\[2em] \displaystyle\bigoplus_{r=1}^{N/2} \bigoplus_{n=0}^{2r-1} W_{r,\omega_{r,n}}, & \text{otherwise}. \end{cases} \tag{192}$$

Regarding point (ii), an important difference for $w = \varepsilon_{\mathsf{ab}}q^m$ as in (191b) is that the decomposition of $\mathsf{Y}_{0,0,\varepsilon_{\mathsf{a}}q^k,\varepsilon_{\mathsf{b}}q^\ell,[z,w]}$ may contain indecomposable yet reducible factors. For $x = \varepsilon_{\mathsf{a}}q^k$ and $y = \varepsilon_{\mathsf{b}}q^\ell$, the condition for the top case of (185) can be expressed as

$$w^{\pm 1} = \varepsilon_{\mathsf{ab}}q^m, \qquad \text{with} \qquad m \in \left\{ |k-\ell|+1, |k-\ell|+3, \ldots, \left[ \tfrac{N}{2}-1 \text{ or } \tfrac{N}{2} \right] \right\}, \tag{193}$$

where the factor in the bracket is chosen so that it has the same parity as $|k-\ell|+1$. In contrast, if one sets $w^{\pm 1} = \varepsilon_{\mathsf{ab}}q^m$ where $m \in \{1, 2, \ldots, \tfrac{N}{2}\}$ but not in the above set, then the decomposition of $\mathsf{Y}_{0,0,\varepsilon_{\mathsf{a}}q^k,\varepsilon_{\mathsf{b}}q^\ell,[z,w]}$ is given by the bottom case of (185).

We now discuss the decomposition of $\left( \mathsf{W}_{0,\varepsilon_{\mathsf{a}}q^k} \times \mathsf{R}_{0,\varepsilon_{\mathsf{b}}q^\ell} \right)_{[z,w]}$ for the different values of $w$.

- Case 1: $w$ is *not* of the form $w^{\pm 1} = \varepsilon_{\mathsf{ab}}q^m$ with $m \in \{1, 2, \ldots, \tfrac{N}{2}\}$. In this case, the generic decomposition (83a) of $\mathsf{Y}_{0,0,\varepsilon_{\mathsf{a}}q^k,\varepsilon_{\mathsf{b}}q^\ell,[w,z]}$ applies. We consider separately two subcases.

  - Case 1a: $w \neq \varepsilon_{\mathsf{ab}}$. In this case, the state $\sigma_{0,\ell}^{(0,0)}$ is nonzero, so we find

  $$\left( \mathsf{W}_{0,\varepsilon_{\mathsf{a}}q^k} \times \mathsf{R}_{0,\varepsilon_{\mathsf{b}}q^\ell} \right)_{[z,w]} = \mathsf{W}_{0,w} \oplus \widetilde{\bigoplus_{r,n}} \mathsf{W}_{r,\omega_{r,n}}. \tag{194}$$

  - Case 1b: $w = \varepsilon_{\mathsf{ab}}$. In this case, $\sigma_{0,\ell}^{(0,0)}$ vanishes for $\ell \geqslant k+1$ and $k+\ell \in 2\mathbb{Z}+1$, but is non-zero otherwise. This yields

  $$\left( \mathsf{W}_{0,\varepsilon_{\mathsf{a}}q^k} \times \mathsf{R}_{0,\varepsilon_{\mathsf{b}}q^\ell} \right)_{[z,\varepsilon_{\mathsf{ab}}]} = \begin{cases} \widetilde{\bigoplus_{r,n}} \mathsf{W}_{r,\omega_{r,n}}, & \ell \geqslant k+1, \ k+\ell \in 2\mathbb{Z}+1, \\ \mathsf{W}_{0,\varepsilon_{\mathsf{ab}}} \oplus \widetilde{\bigoplus_{r,n}} \mathsf{W}_{r,\omega_{r,n}}, & \text{otherwise.} \end{cases} \tag{195}$$

- Case 2: $w^{\pm 1} = \varepsilon_{\mathsf{ab}}q^m$ with $m \in \{[1 \text{ or } 2], [3 \text{ or } 4], \ldots, |k-\ell|-1\}$, where the factors in the brackets are chosen to have the same parity as $|k-\ell|-1$. The bottom case of (185) applies, where $\mathsf{Z}_{0,0,x,y,[z,w]}$ has three composition factors. There are two subcases, depending on the values of $k$ and $\ell$.

  - Case 2a: $\ell > k+1$. In this case, we have $\Pi_{m,n_{\varepsilon_{\mathsf{ab}}}} \cdot \sigma_{0,\ell}^{(m)} \equiv 0 \, [[m-1]]$ and $\sigma_{0,\ell}^{(0,0)} = 0$, and therefore none of the factors of $\mathsf{Z}_{0,0,x,y,[z,w]}$ are produced. This yields

  $$\left( \mathsf{W}_{0,\varepsilon_{\mathsf{a}}q^k} \times \mathsf{R}_{0,\varepsilon_{\mathsf{b}}q^\ell} \right)_{[z,\varepsilon_{\mathsf{ab}}q^m]} \simeq \widetilde{\bigoplus_{r,n}} \mathsf{W}_{r,\omega_{r,n}}. \tag{196}$$

  We note that the factor $\mathsf{W}_{m,\varepsilon_{\mathsf{ab}}}$ that acts as the head of $\mathsf{Z}_{0,0,x,y,[z,w]}$ is absent from the direct sum because of the restriction over $(r,n)$ described in (192).

  - Case 2b: $\ell \leqslant k+1$. In this case, $\Pi_{m,n_{\varepsilon_{\mathsf{ab}}}} \cdot \sigma_{0,\ell}^{(m)} \not\equiv 0 \, [[m-1]]$. Thus the head of the module $\mathsf{Z}_{0,0,x,y,[z,w]}$ is produced by the action of the algebra on the seed state of the module $\left( \mathsf{W}_{0,\varepsilon_{\mathsf{a}}q^k} \times \mathsf{R}_{0,\varepsilon_{\mathsf{b}}q^\ell} \right)_{[z,w]}$. This implies that the entire module $\mathsf{Z}_{0,0,x,y,[z,w]}$ is generated. Moreover, none of the projected intermediate seed states in (191b) vanish, and therefore

  $$\left( \mathsf{W}_{0,\varepsilon_{\mathsf{a}}q^k} \times \mathsf{R}_{0,\varepsilon_{\mathsf{b}}q^\ell} \right)_{[z,\varepsilon_{\mathsf{ab}}q^m]} = \mathsf{Y}_{0,0,\varepsilon_{\mathsf{a}}q^k,\varepsilon_{\mathsf{b}}q^\ell,[z,\varepsilon_{\mathsf{ab}}q^m]}. \tag{197}$$

- Case 3: $w^{\pm 1} = \varepsilon_{\mathsf{ab}}q^m$ with $m \in \{1, 2, \ldots, \tfrac{N}{2}\}$ and either $m > k+\ell-1$ or $m+k+\ell \in 2\mathbb{Z}$. In this case, the intermediate states satisfy $\Pi_{m,n_{\varepsilon_{\mathsf{ab}}}} \cdot \sigma_{0,\ell}^{(m)} \not\equiv 0 \, [[m-1]]$ and $\sigma_{0,\ell}^{(0,0)} \neq 0$. We thus deduce that

$$\left( \mathsf{W}_{0,\varepsilon_{\mathsf{a}}q^k} \times \mathsf{R}_{0,\varepsilon_{\mathsf{b}}q^\ell} \right)_{[z,\varepsilon_{\mathsf{ab}}q^m]} \simeq \mathsf{Z}_{0,0,\varepsilon_{\mathsf{a}}q^k,\varepsilon_{\mathsf{b}}q^\ell,[z,\varepsilon_{\mathsf{ab}}q^m]} \oplus \widetilde{\bigoplus_{\substack{r,n \\ (r,n) \neq (m,n_{\varepsilon_{\mathsf{ab}}})}}} \mathsf{W}_{r,\omega_{r,n}}. \tag{198}$$

There are thus two restrictions on the values of $(r, n)$ in this case. The decomposition of $Z_{0,0,\varepsilon_a q^k, \varepsilon_b q^\ell, [z, \varepsilon_{ab} q^m]}$ is given by the top case of (185) for $m + k + \ell \in 2\mathbb{Z} + 1$, and by the bottom case for $m + k + \ell \in 2\mathbb{Z}$.

- Case 4: $w^{\pm 1} = \varepsilon_{ab} q^m$ with $m \in \{|k - \ell| + 1, |k - \ell| + 3, \ldots, k + \ell - 1\}$. The top case of (185) applies, and the intermediate seed states satisfy $\Pi_{m, n_{\varepsilon_{ab}}} \cdot \sigma_{0,\ell}^{(m)} \not\equiv 0$ [[$m - 1$]] and $\sigma_{0,\ell}^{(0,0)} = 0$. Hence we have

$$\left( W_{0, \varepsilon_a q^k} \times R_{0, \varepsilon_b q^\ell} \right)_{[z, \varepsilon_{ab} q^m]} \simeq W_{m, \varepsilon_{ab}} \oplus \bigoplus_{\substack{r, n \\ (r,n) \neq (m, n_{\varepsilon_{ab}})}} \widetilde{W}_{r, \omega_{r,n}} = \bigoplus_{r,n} \widetilde{W}_{r, \omega_{r,n}}. \tag{199}$$

We note that, for $\ell \leqslant k$, we have $\left( W_{0, \varepsilon_a q^k} \times R_{0, \varepsilon_b q^\ell} \right)_{[z, w]} = Y_{0,0, \varepsilon_a q^k, \varepsilon_b q^\ell, [z, w]}$ in all the cases except for Case 4.

## 5.3 The decomposition of Q × Q

In this section, we investigate the fusion $Q_{0, \varepsilon_a q^k} \times Q_{0, \varepsilon_b q^\ell}$ of two quotient modules with zero defects. The corresponding fusion modules are defined diagrammatically by endowing the module $Y_{0,0,\varepsilon_a q^k, \varepsilon_b q^\ell, [z, w]}$ with two extra quotient relations. To obtain the decomposition of this module, it will also be useful to formulate its definition in terms of the fusion modules for W × W, W × R and R × W that we already investigated.

Let us first illustrate the diagrammatic construction for the simplest example with $k = \ell = 1$. In this case, the action of the algebra $\mathcal{E}PTL_N(\beta)$ on the link states of $Y_{0,0,x,y,[z,w]}$ involves two quotient relations that allow us to commute the points a and b across loop segments, at the cost of signs $-\varepsilon_a$ and $-\varepsilon_b$ respectively. The argument presented in (112) generalises directly. We deduce that the fusion module is non-zero only if $\alpha_{ab} = -\varepsilon_b \alpha_a = -\varepsilon_a \alpha_b$, or equivalently for $w^{\pm 1} = \varepsilon_{ab} q$. As a basis of this module, one can choose the subset of link states of $Y_{0,0,x,y,[z,w]}$ that have both marked points adjacent and located close to the perimeter of the disc, between the nodes $N$ and 1. There is a direct bijection between this basis and the basis of the vacuum module V. The action of the algebra on this Q × Q module assigns weights $\alpha_{ab} = \varepsilon_{ab} \beta$ to loops encircling the two points a and b. As a result, we directly conclude that

$$\left( Q_{0, \varepsilon_a q} \times Q_{0, \varepsilon_b q} \right)_{[z, w]} = \begin{cases} Q_{0, w}, & w^{\pm 1} = \varepsilon_{ab} q, \\ 0, & \text{otherwise.} \end{cases} \tag{200}$$

As a second example, we consider the case where $k = 1$ and $\ell > 1$. In this case, only the marked point a can be moved freely across loop segments. Repeating again the argument in (112), we deduce that the fusion module is non-zero only for $w^{\pm 1} = \varepsilon_{ab} q^\ell$. By moving a across loop segments, we can restrict our focus to the subset of link states of $Y_{0,0,x,y,[z,w]}$ where a is directly adjacent to b. There is a direct bijection between this set and the set $B_0(N)$ with no defects and one marked point. The action of the algebra on these link states follows the action in $W_{0,w}$, except that it also involves the second quotient relation. This second relation allows us to express any link state with $r_b \geqslant \ell$ in terms of link states with $r_b < \ell$. As a result, the basis can be restricted further, so as to include only link states with $r_b < \ell$. The standard module $W_{0,w}$ with $w^{\pm 1} = \varepsilon_{ab} q^\ell$ is reducible, with its structure as given in (23). The extra quotient relation is precisely the one that maps the radical $R_{0,w}$ to zero, leaving only the quotient module $Q_{0,w}$. We thus conclude that

$$\left( Q_{0, \varepsilon_a q} \times Q_{0, \varepsilon_b q^\ell} \right)_{[z, w]} = \begin{cases} Q_{0, w}, & w^{\pm 1} = \varepsilon_{ab} q^\ell, \\ 0, & \text{otherwise.} \end{cases} \tag{201}$$

The cases with $k, \ell > 1$ are more complicated, as the resulting fusion modules are non-zero for more than one values of $w$. The diagrammatic definition of the fusion modules for $Q \times Q$ is equivalently written as

$$\left(Q_{0,\varepsilon_a q^k} \times Q_{0,\varepsilon_b q^\ell}\right)_{[z,w]} = Y_{0,0,\varepsilon_a q^k,\varepsilon_b q^\ell,[z,w]} \Big/ \left[\left(W_{0,\varepsilon_a q^k} \times R_{0,\varepsilon_b q^\ell}\right)_{[z,w]} + \left(R_{0,\varepsilon_a q^k} \times W_{0,\varepsilon_b q^\ell}\right)_{[z,w]}\right]. \quad (202)$$

The sum of $W \times R$ and $R \times W$, as vector spaces, is clearly a submodule of $W \times W$. Taking the quotient as in (202) thus amounts to setting to zero in $Y_{0,0,\varepsilon_a q^k,\varepsilon_b q^\ell,[z,w]}$ all states in this larger submodule. To determine the structure of the module $Q \times Q$, one should in principle consider the two seed states

$$u = (P_{2k} \otimes 1_{2\ell})\Omega^{-k} \cdot u_{0,0}(2k + 2\ell), \qquad v = (1_{2k} \otimes P_{2\ell})\Omega^{-k} \cdot u_{0,0}(2k + 2\ell), \quad (203)$$

and study the action of the algebra on arbitrary linear combinations of these two states. This analysis is certainly tedious, and here we instead take a shortcut to obtain the decomposition.

From the last remark of Section 5.2, it readily follows that, for Cases 1, 2 and 3, at least one of the fusion modules $\left(W_{0,\varepsilon_a q^k} \times R_{0,\varepsilon_b q^\ell}\right)_{[z,w]}$ or $\left(R_{0,\varepsilon_a q^k} \times W_{0,\varepsilon_b q^\ell}\right)_{[z,w]}$ is isomorphic to $Y_{0,0,\varepsilon_a q^k,\varepsilon_b q^\ell,[z,w]}$. Thus it is clear from (202) that the corresponding fusion module $Q \times Q$ vanishes in these cases. Hence, the only values of $w$ where $Q \times Q$ is potentially nonzero are those for Case 4, namely

$$w^{\pm 1} = \varepsilon_{ab} q^m, \qquad \text{with} \qquad m \in \{|k - \ell| + 1, |k - \ell| + 3, \dots, k + \ell - 1\}. \quad (204)$$

Here, the system size is assumed to satisfy the inequality $\frac{N}{2} \geqslant k + \ell - 1$, so that all the values of $m$ in (204) are in the set $\{1, 2, \dots, \frac{N}{2}\}$.

For Case 4, both $\left(W_{0,\varepsilon_a q^k} \times R_{0,\varepsilon_b q^\ell}\right)_{[z,w]}$ and $\left(R_{0,\varepsilon_a q^k} \times W_{0,\varepsilon_b q^\ell}\right)_{[z,w]}$ decompose as in (199). Each factor $W_{r,\omega_{r,n}}$ with $r = 1, 2, \dots, \frac{N}{2}$ and $n = 0, 1, \dots, 2r - 1$ appears in at least one of the modules $W \times R$ or $R \times W$. As a result, these factors also arise in $W \times R + R \times W$. For $(r, n) \neq (m, n_{\varepsilon_{ab}})$, the module $W_{r,\omega_{r,n}}$ arises exactly once in the decomposition of $Y_{0,0,\varepsilon_a q^k,\varepsilon_b q^\ell,[z,w]}$. As a result, their multiplicity in $W \times R + R \times W$ is also equal to one. In contrast, $Y_{0,0,\varepsilon_a q^k,\varepsilon_b q^\ell,[z,w]}$ has two copies of $W_{m,\varepsilon_{ab}}$. Indeed, the composition factors $R_{0,\varepsilon q^m}$ and $W_{m,\varepsilon}$ arising in the top case of (185) are isomorphic to the same irreducible module $I_{m,\varepsilon}$. The fusion modules for $W \times R$ and $R \times W$ have identical decompositions, with one copy of the module $I_{m,\varepsilon}$. But these may not involve the same copy of $I_{m,\varepsilon}$, out of the two copies arising in $Y_{0,0,\varepsilon_a q^k,\varepsilon_b q^\ell,[z,w]}$. Moreover, the factor $Q_{0,\varepsilon_{ab} q^m}$ never arises, as the action of the algebra on both $u$ and $v$ produces intermediate seed states that vanish for all values of $w$ associated to Case 4. As a result, we conclude that

$$\left(W_{0,\varepsilon_a q^k} \times R_{0,\varepsilon_b q^\ell}\right)_{[z,w=\varepsilon_{ab} q^m]} + \left(R_{0,\varepsilon_a q^k} \times W_{0,\varepsilon_b q^\ell}\right)_{[z,w=\varepsilon_{ab} q^m]} \subseteq 2\, W_{m,\varepsilon_{ab}} \oplus \bigoplus_{\substack{r=1 \\ (r,n)\neq(m,n_{\varepsilon_{ab}})}}^{N/2} \bigoplus_{n=0}^{2r-1} W_{r,\omega_{r,n}}. \quad (205)$$

To establish an equality and obtain the decomposition of $(W \times R + R \times W)$, we need only determine whether the multipicity of $W_{m,\varepsilon_{ab}}$ is equal to 1 or 2. This leaves two possibilities for the decomposition of $Q \times Q$:

$$\left(Q_{0,\varepsilon_a q^k} \times Q_{0,\varepsilon_b q^\ell}\right)_{[z,w=\varepsilon_{ab} q^m]} \simeq \left[\begin{array}{c} Q_{0,\varepsilon_{ab} q^m} \\ \searrow \\ R_{0,\varepsilon_{ab} q^m} \end{array}\right], \quad \text{or} \quad \left(Q_{0,\varepsilon_a q^k} \times Q_{0,\varepsilon_b q^\ell}\right)_{[z,w=\varepsilon_{ab} q^m]} \simeq Q_{0,\varepsilon_{ab} q^m}. \quad (206)$$

We now show that the leftmost decomposition is not possible. Let us assume without loss of generality that $k \leqslant \ell$. If $Q \times Q$ had the leftmost structure, this would imply that $Q \times Q$ and $Q \times W$

are both isomorphic to $W_{0,w}$. The module $Q \times W$ has a well-defined basis whose elements are written in terms of linear combinations of link states of $Y_{0,0,\varepsilon_a q^k, \varepsilon_b q^\ell, [z,w]}$, for which the crossing number $r_b$ is not restricted in terms of $\ell$. By writing $Q \times Q$ as $(Q \times W)/(Q \times R)$, one effectively imposes an extra quotient relation to the action of $W_{0,w}$, that allows one to express all link states with $r_b \geqslant \ell$ in terms of link states with $r_b < \ell$. This implies that a basis for $Q \times Q$ is necessarily strictly smaller than a basis of $Q \times W$, thus excluding the leftmost option in (206). We conclude that the fusion module $Q \times Q$ has the structure

$$\left(Q_{0,\varepsilon_a q^k} \times Q_{0,\varepsilon_b q^\ell}\right)_{[z,w]} \simeq \begin{cases} Q_{0,w} & \begin{cases} w = \varepsilon_{ab} q^m, \\ m \in \{|k-\ell|+1, |k-\ell|+3, \ldots, k+\ell+1\}, \end{cases} \\ 0, & \text{otherwise.} \end{cases} \tag{207}$$

The multiplicity of $W_{m,\varepsilon_{ab}}$ in $(W \times R + R \times W)$ is therefore equal to 2. The corresponding fusion rule is

$$Q_{0,\varepsilon_a q^k} \times Q_{0,\varepsilon_b q^\ell} \to \left\{ Q_{0,\varepsilon_{ab} q^m} \mid m \in \{|k-\ell|+1, |k-\ell|+3, \ldots, k+\ell+1\}\right\}. \tag{208}$$

## 6 The CFT interpretation

In this section, we study the scaling limit of the dense loop model and its description with a logarithmic bulk CFT. We use known facts about the scaling limit of the standard modules to derive CFT predictions for fusion products involving at least one quotient module $Q_{0,\varepsilon q^m}$ with $m \in \mathbb{Z}_{\geqslant 1}$. We compare these with our lattice derivations in the previous sections of the fusion rules $W \times Q$ and $Q \times Q$.

### 6.1 The dense loop model and its scaling limit

In this section, we consider the dense loop model on the square lattice with its configurations weighted by the Boltzmann weights $\beta^n$ where $n$ is the number of closed loops and $\beta = -q - q^{-1} \in (-2, 2)$ is the loop weight. For a homogeneous system defined on a cylinder with a circumference of $N$ sites, the evolution operator is the transfer matrix $T$. It is the element of $\mathcal{E}\mathsf{PTL}_N(\beta)$ defined as

$$T = \quad\text{where}\quad \boxed{\phantom{x}} = \diagup\!\diagdown + \diagdown\!\diagup. \tag{209}$$

By expanding each tile as the sum of two contributions, one obtains a set of $2^N$ connectivities of $\mathcal{E}\mathsf{PTL}_N(\beta)$. The transfer matrix is then the sum of these diagrams, each one assigned a unit weight.

Let us consider the action of $T$ and of the shift operator $\Omega$ on an $\mathcal{E}\mathsf{PTL}_N(\beta)$-module $\mathsf{M}(N)$. Moreover, let $\Lambda_\alpha(N)$ be the eigenvalues of $T$, with labels $\alpha = 0, 1, 2, \ldots$ conveniently chosen such that $|\Lambda_0(N)| \geqslant |\Lambda_1(N)| \geqslant |\Lambda_2(N)| \geqslant \ldots$ . For large $N$, these leading eigenvalues of $T$ are known to behave as [22, 23]

$$\log \Lambda_\alpha(N) = -N f_\infty - \frac{2\pi}{N}\left(h_\alpha + \bar{h}_\alpha - \frac{c}{12}\right) + o(\tfrac{1}{N}), \tag{210}$$

and the corresponding eigenvalues of $\Omega$ are of the form $\exp(\mathrm{i}P_\alpha)$ with

$$P_\alpha = \frac{2\pi}{N}(h_\alpha - \bar{h}_\alpha) + \pi k_\alpha, \qquad k_\alpha \in \{0, 1\}. \tag{211}$$

Here $f_\infty$ is the bulk free energy and it is independent of $\alpha$. The gap $\log \Lambda_0(N) - \log \Lambda_\alpha(N)$ for these leading eigenvalues thus scales as $\frac{1}{N}$. In the scaling limit, the dense loop model is described by an effective logarithmic CFT whose symmetry algebra is the tensor product $\mathrm{Vir} \otimes \overline{\mathrm{Vir}}$ of two copies of the Virasoro algebra. The eigenstates for which the gap scales as $\frac{1}{N}$ become the conformal states in this underlying CFT. In (210), the term proportional to $\frac{1}{N}$ is the first finite-size correction. It depends on the central charge

$$c = 1 - 6(b^{-1} - b)^2, \qquad \text{where} \qquad q = \exp(-\mathrm{i}\pi b^2), \qquad 0 < b < 1. \tag{212}$$

It also involves the conformal weights $h_\alpha$ and $\bar{h}_\alpha$, namely the eigenvalues of the operators $L_0$ and $\bar{L}_0$ in a module $\mathsf{M}^\infty$ over $\mathrm{Vir} \otimes \overline{\mathrm{Vir}}$ interpreted as the scaling limit of $\mathsf{M}(N)$.

The character of a $\mathrm{Vir} \otimes \overline{\mathrm{Vir}}$-module $\mathsf{Z}$ is defined as

$$\chi_{\mathsf{Z}}(\tau) = \mathrm{Tr}_{\mathsf{Z}}\left[q^{L_0 - c/24}\,\bar{q}^{\bar{L}_0 - c/24}\right], \qquad q = e^{2\mathrm{i}\pi\tau}, \quad \bar{q} = e^{-2\mathrm{i}\pi\bar{\tau}}, \tag{213}$$

in terms of a fixed modular parameter $\tau = \tau_1 + \mathrm{i}\tau_2$, with $\tau_1, \tau_2 \in \mathbb{R}$ and $\tau_2 > 0$. Similarly, let $\mathsf{M}(N)$ be an $\mathcal{E}\mathsf{PTL}_N(\beta)$-module. One defines its finite-size character as

$$\chi_{\mathsf{M}(N)}(\tau) = \mathrm{Tr}_{\mathsf{M}(N)}\left[\Omega^{M_1}\,\boldsymbol{T}^{M_2}\right], \tag{214}$$

where the integers $M_1$ and $M_2$ are given by $M_1 = \lfloor N\tau_1 \rfloor$ and $M_2 = \lfloor N\tau_2 \rfloor$. In the scaling limit, $N$ is sent to infinity with $\tau$ kept fixed, and therefore $M_1, M_2$ are also sent to infinity. Eigenvalues $\Lambda_\alpha(N)$ for which the gap $\log \Lambda_0(N) - \log \Lambda_\alpha(N)$ is larger than $\mathcal{O}(\frac{1}{N})$ have contributions to the finite-size character $\chi_{\mathsf{M}(N)}(\tau)$ that vanish as $N \to \infty$. Thus, one generally expects the convergence of the finite-size characters to the conformal characters as

$$\chi_{\mathsf{M}(N)}(\tau) \xrightarrow{N\to\infty} \chi_{\mathsf{M}^\infty}(\tau). \tag{215}$$

## 6.2 The scaling limit of the standard modules

For the standard modules, the finite-size characters are known [24] to converge to the infinite sesquilinear form of Verma characters

$$\chi_{\mathsf{W}_{k,\exp(\mathrm{i}\pi\mu)}(N)}(\tau) \xrightarrow{N\to\infty} \sum_{p\in\mathbb{Z}} (-1)^{pM} \chi_{\mathsf{V}(h_{\mu+p,k})\otimes\bar{\mathsf{V}}(h_{\mu+p,-k})}(\tau), \qquad \mu \in \mathbb{R}, \tag{216}$$

where the Kac notation for the conformal weights is

$$h_{r,s} = \frac{(b^{-1}r - bs)^2 - (b^{-1} - b)^2}{4}, \tag{217}$$

the Verma module over $\mathrm{Vir}$ with conformal weight $h$ is denoted by $\mathsf{V}(h)$, its chiral character is

$$\chi_{\mathsf{V}(h)} = \mathrm{Tr}_{\mathsf{V}(h)}(q^{L_0 - c/24}), \tag{218}$$

and

$$M = \begin{cases} 0, & M_1 + M_2 \in 2\mathbb{Z}, \\ 1, & M_1 + M_2 \in 2\mathbb{Z} + 1. \end{cases} \tag{219}$$

We therefore consider separately the scaling limit over odd and even values of $M_1 + M_2$.

We remark that the right side of (216) is unchanged under the transformation $\mu \to \mu + 1$, up to an overall sign $(-1)^M$. This can be understood from the lattice construction. Indeed, let us denote by $T_{k,x}$ and $\Omega_{k,x}$ the matrix representatives of $\boldsymbol{T}$ and $\Omega$ in $\mathsf{W}_{k,x}$. Let us also define the parity of a connectivity of $\mathcal{E}\mathsf{PTL}_N(\beta)$ as the parity of the number of loop segments that

cross the dashed line. With this definition, $T$ and $\Omega$ are made of odd connectivities only. It is then easy to construct a change of basis $S$ over $W_{k,x}$ such that

$$S\,T_{k,-x}S^{-1} = -T_{k,x}\,, \qquad S\,\Omega_{k,-x}S^{-1} = -\Omega_{k,x}\,. \tag{220}$$

This change of basis is diagonal and maps each link state $v$ to $(-1)^r v$ where $r$ is the crossing number of $v$. We thus conclude that the finite-size character also picks up the sign $(-1)^M$ under the transformation $\mu \to \mu + 1$. Thus the behaviour of (216) under $\mu \to \mu + 1$ is consistent at the lattice level with the symmetries of $T$ and $\Omega$.

For $M = 1$, certain terms in the right side of (216) have negative coefficients. As a result, this expression cannot be obtained as the scaling limit of a character of the form (213), as in (215), because all the terms in (213) have positive coefficients. While there may be a way to modify (213) so that this convergence holds also for $M = 1$, here we choose to continue the analysis by focusing on the scaling limit with $M = 0$. The arguments below are presented with this assumption. The results for the fusion rules are then entirely blind to the sign $\varepsilon$ of the twist $x$.

Let us first consider standard modules $W_{k,x}$ where $x = \exp(i\pi\mu)$ with $\mu \notin b^2\mathbb{Z} + \mathbb{Z}$. This means that $x$ is not of the form $\varepsilon q^m$ with $m \in \mathbb{Z}$ and $\varepsilon \in \{+1, -1\}$. In this case, for $p \in \mathbb{Z}$, the dimensions $h_{\mu+p,k}$ and $h_{\mu+p,-k}$ cannot be written as $h_{r,s}$ with $r, s$ positive integers, and each term in the sum in the right side of (216) is the product of two irreducible Virasoro characters. This suggests that the scaling limit of $W_{k,x}$ is

$$W_{k,\exp(i\pi\mu)}(N) \xrightarrow{N\to\infty} W_{k,\exp(i\pi\mu)}^{\infty} = \bigoplus_{p\in\mathbb{Z}} V(h_{\mu+p,k}) \otimes \bar{V}(h_{\mu+p,-k})\,, \qquad \mu \notin b^2\mathbb{Z} + \mathbb{Z}\,. \tag{221}$$

The situation is different for the standard modules $W_{k,\varepsilon}$ with $\varepsilon \in \{-1, +1\}$. In this case, the conformal dimensions appearing in the right side of (216) are of the form $(h_{p,k}, h_{p,-k})$ with $p, k \in \mathbb{Z}$. The modules $V(h_{p,k})$ with $p > 0$ and $V(h_{p,-k})$ with $p < 0$ are reducible modules. For $p > 0$, the terms $\chi_{V(h_{p,k})\otimes\bar{V}(h_{p,-k})}(\tau)$ and $\chi_{V(h_{-p,k})\otimes\bar{V}(h_{-p,-k})}(\tau)$ in the sum of characters belong to a reducible, indecomposable $(\mathrm{Vir} \otimes \overline{\mathrm{Vir}})$-module that is diamond-shaped [25].

Finally, let us consider a standard module $W_{0,\varepsilon q^m}$ with $m \in \mathbb{Z}_{\geqslant 1}$. Its structure, read from (5), is

$$W_{0,\varepsilon q^m} \simeq \begin{bmatrix} Q_{0,\varepsilon q^m} & \\ & \searrow \\ & R_{0,\varepsilon q^m} \end{bmatrix}\,, \qquad R_{0,\varepsilon q^m} \simeq W_{m,\varepsilon}\,. \tag{222}$$

From (216), one has

$$\chi_{W_{0,\varepsilon q^m}(N)} \xrightarrow{N\to\infty} |\chi_{V(h_{0,m})}|^2 + \sum_{p=1}^{\infty}\left(|\chi_{V(h_{p,m})}|^2 + |\chi_{V(h_{p,-m})}|^2\right)\,, \tag{223a}$$

$$\chi_{W_{m,\varepsilon}(N)} \xrightarrow{N\to\infty} |\chi_{V(h_{0,m})}|^2 + \sum_{p=1}^{\infty}\left(\chi_{V(h_{p,m})}\bar{\chi}_{V(h_{p,-m})} + \chi_{V(h_{p,-m})}\bar{\chi}_{V(h_{p,m})}\right)\,, \tag{223b}$$

where we used the properties

$$h_{r-mb^2,s} = h_{r,s+m}\,, \qquad h_{-r,-s} = h_{r,s}\,. \tag{224}$$

From (223), we deduce that the characters for the quotient modules scale to

$$\chi_{Q_{0,\varepsilon q^m}(N)} \xrightarrow{N\to\infty} \sum_{p=1}^{\infty}\left|\chi_{V(h_{p,m})} - \chi_{V(h_{p,-m})}\right|^2\,. \tag{225}$$

For generic values of the central charge $c$, the structure of $\mathsf{V}(h_{p,m})$ as a module over Vir is

$$
\mathsf{V}(h_{p,m}) \simeq \left[ \begin{array}{c} \mathsf{K}(h_{p,m}) \\ \searrow \\ \mathsf{R}(h_{p,m}) \end{array} \right], \qquad \mathsf{R}(h_{p,m}) \simeq \mathsf{V}(h_{p,-m}), \tag{226}
$$

for all positive integers $p, m$. Here $\mathsf{R}(h_{p,m})$ is the radical of $\mathsf{V}(h_{p,m})$, namely the space of vectors generated by the singular vector at level $h_{p,-m} = h_{p,m} + pm$. It follows that

$$
\chi_{\mathsf{Q}_{0,\varepsilon q^m}(N)}(\tau) \xrightarrow{N \to \infty} \sum_{p=1}^{\infty} \chi_{\mathsf{K}(h_{p,m}) \otimes \bar{\mathsf{K}}(h_{p,m})}(\tau). \tag{227}
$$

Because the Virasoro modules in the right side of (227) are irreducible, the above equation suggests that the scaling limit of $\mathsf{Q}_{0,\varepsilon q^m}$ is

$$
\mathsf{Q}_{0,\varepsilon q^m}(N) \xrightarrow{N \to \infty} \mathsf{Q}_{0,\varepsilon q^m}^{\infty} = \bigoplus_{r=1}^{\infty} \mathsf{K}(h_{r,m}) \otimes \bar{\mathsf{K}}(h_{r,m}). \tag{228}
$$

## 6.3 The fusion products $\mathsf{W} \boxtimes \mathsf{Q}$ and $\mathsf{Q} \boxtimes \mathsf{Q}$

We denote by $\mathsf{M} \boxtimes \mathsf{N}$ the CFT fusion of two Virasoro modules $\mathsf{M}$ and $\mathsf{N}$. We use the same symbol $\boxtimes$ to denote the fusion of modules over Vir and over $\mathrm{Vir} \otimes \overline{\mathrm{Vir}}$. We are interested in computing the fusion products $\mathsf{W}_{k,x}^{\infty} \boxtimes \mathsf{Q}_{0,\varepsilon q^m}^{\infty}$ and $\mathsf{Q}_{0,\varepsilon_a q^k}^{\infty} \boxtimes \mathsf{Q}_{0,\varepsilon_b q^\ell}^{\infty}$. Starting with $\mathsf{W}_{k,x}^{\infty} \boxtimes \mathsf{Q}_{0,\varepsilon q^m}^{\infty}$, we first compute separately $\mathsf{V}(h_{\mu+p,k}) \boxtimes \mathsf{K}(h_{r,m})$ and $\bar{\mathsf{V}}(h_{\mu+p,-k}) \boxtimes \bar{\mathsf{K}}(h_{r,m})$ with $p \in \mathbb{Z}$ and $r \in \mathbb{Z}_{\geqslant 1}$, and then take the direct sum over $r$ and $p$ of the tensor product of the results.

Let us start with the special case $r = 1$. The central charge is taken to be generic, namely $b^2 \notin \mathbb{Q}$ in (212). We also fix $\mu$ and $k$ so that $\mathsf{V}(h_{\mu+p,k})$ and $\bar{\mathsf{V}}(h_{\mu+p,-k})$ are irreducible Virasoro modules, for all $p \in \mathbb{Z}$. In this case, standard CFT arguments yield the chiral fusion rules

$$
\mathsf{V}(h_{\mu+p,k}) \boxtimes \mathsf{K}(h_{1,m}) = \bigoplus_{i=-\frac{m-1}{2}}^{\frac{m-1}{2}} \mathsf{V}(h_{\mu+p,k+2i}), \tag{229a}
$$

$$
\bar{\mathsf{V}}(h_{\mu+p,-k}) \boxtimes \bar{\mathsf{K}}(h_{1,m}) = \bigoplus_{j=-\frac{m-1}{2}}^{\frac{m-1}{2}} \bar{\mathsf{V}}(h_{\mu+p,-k+2j}). \tag{229b}
$$

It follows from (224) that

$$
h_{\mu+p,k+2i} = h_{\mu+p-(i+j)b^2,k+i-j}, \qquad h_{\mu+p,-k+2j} = h_{\mu+p-(i+j)b^2,-k-i+j}. \tag{230}
$$

Using these relations, we obtain the fusion product

$$
\left( \mathsf{V}(h_{\mu+p,k}) \otimes \bar{\mathsf{V}}(h_{\mu+p,-k}) \right) \boxtimes \left( \mathsf{K}(h_{1,m}) \otimes \bar{\mathsf{K}}(h_{1,m}) \right)
$$

$$
= \bigoplus_{i,j=-\frac{m-1}{2}}^{\frac{m-1}{2}} \mathsf{V}(h_{\mu+p-(i+j)b^2,k+i-j}) \otimes \bar{\mathsf{V}}(h_{\mu+p-(i+j)b^2,-k-i+j}). \tag{231}
$$

Summing over $p \in \mathbb{Z}$ gives

$$
\mathsf{W}_{k,x}^{\infty} \boxtimes \left( \mathsf{K}(h_{1,m}) \otimes \bar{\mathsf{K}}(h_{1,m}) \right) = \bigoplus_{i,j=-\frac{m-1}{2}}^{\frac{m-1}{2}} \mathsf{W}_{k+i-j,xq^{i+j}}^{\infty}. \tag{232}
$$

We note that the right-hand side exactly matches the scaling limit of the possible outcomes of the fusion rule for $\mathsf{W}_{k,z} \times \mathsf{Q}_{0,\varepsilon q^m}$ that we obtained from our lattice construction of fusion in (149).

For the other components of $\mathsf{Q}_{0,\varepsilon q^m}^\infty$, a similar argument yields

$$\left(\mathsf{V}(h_{\mu+p,k}) \otimes \bar{\mathsf{V}}(h_{\mu+p,-k})\right) \boxtimes \left(\mathsf{K}(h_{r,m}) \otimes \bar{\mathsf{K}}(h_{r,m})\right)$$
$$= \bigoplus_{u,v=-\frac{r-1}{2}}^{\frac{r-1}{2}} \bigoplus_{i,j=-\frac{m-1}{2}}^{\frac{m-1}{2}} \mathsf{V}(h_{\mu+p+2u-(i+j)b^2,k+i-j}) \otimes \bar{\mathsf{V}}(h_{\mu+p+2v-(i+j)b^2,-k-i+j}), \quad (233)$$

from which we deduce that

$$\mathsf{W}_{k,x}^\infty \boxtimes \mathsf{Q}_{0,\varepsilon q^m}^\infty = \bigoplus_{p\in\mathbb{Z}} \bigoplus_{r=1}^{\infty} \bigoplus_{u,v=-\frac{r-1}{2}}^{\frac{r-1}{2}} \bigoplus_{i,j=-\frac{m-1}{2}}^{\frac{m-1}{2}} \mathsf{V}(h_{\mu+p+2u-(i+j)b^2,k+i-j}) \otimes \bar{\mathsf{V}}(h_{\mu+p+2v-(i+j)b^2,-k-i+j}). \quad (234)$$

This is the final result for the fusion of a standard module $\mathsf{W}_{k,x}^\infty$ and a quotient module $\mathsf{Q}_{0,\varepsilon q^m}^\infty$, as Virasoro modules. Our lattice results (149) reveal that the terms with $u = v$ all arise in the lattice fusion rules, producing precisely the right-hand side of (232). In contrast, the other terms with $u \neq v$ are not produced, and in fact their direct sum cannot be reorganised as a direct sum of standard modules. The product $\mathsf{M} \boxtimes \mathsf{N}$ in fact gives the list of all the modules that may appear in the fusion, and a given lattice realisation of the fusion of these modules may produce only a subset in this list. In other words, one expects that

$$\mathsf{M} \times \mathsf{N} \subseteq \mathsf{M} \boxtimes \mathsf{N}. \quad (235)$$

Our results (149) and (233) are consistent with this expectation.

Similarly, to compute $\mathsf{Q}_{0,\varepsilon_a q^k}^\infty \boxtimes \mathsf{Q}_{0,\varepsilon_b q^\ell}^\infty$, we use (228) and the chiral CFT fusion rules

$$\mathsf{K}(h_{r,k}) \boxtimes \mathsf{K}(h_{s,\ell}) = \bigoplus_{u=\frac{|r-s|+1}{2}}^{\frac{r+s-1}{2}} \bigoplus_{m=\frac{|k-\ell|+1}{2}}^{\frac{k+\ell-1}{2}} \mathsf{K}(h_{2u,2m}), \quad (236)$$

valid for generic central charges, and find

$$\mathsf{Q}_{0,\varepsilon_a q^k}^\infty \boxtimes \mathsf{Q}_{0,\varepsilon_b q^\ell}^\infty = \bigoplus_{r,s=1}^{\infty} \bigoplus_{u,v=\frac{|r-s|+1}{2}}^{\frac{r+s-1}{2}} \bigoplus_{m,n=\frac{|k-\ell|+1}{2}}^{\frac{k+\ell-1}{2}} \mathsf{K}_{2u,2m} \otimes \bar{\mathsf{K}}_{2v,2n}. \quad (237)$$

This can be compared to the result (207), obtained from the lattice fusion for $\mathsf{Q} \times \mathsf{Q}$ constructed from the modules $\mathsf{Y}$. We see that the set of modules that arise in the lattice fusion rules are all included in the right side of (237). They correspond to the terms with $2u = 2v \in \mathbb{Z}_{\geqslant 1}$ and $m = n \in \left\{ \frac{|k-\ell|+1}{2}, \frac{|k-\ell|+3}{2}, \ldots, \frac{k+\ell-1}{2} \right\}$.

## 7 Conclusion

In this paper, we proposed a new candidate for the fusion $\mathsf{W}_{k,x} \times \mathsf{W}_{\ell,y}$ of standard modules over the algebra $\mathcal{E}\mathsf{PTL}_N(\beta)$, with $\beta = -q - q^{-1}$. It is constructed in terms of new families of modules $\mathsf{Y}_{k,\ell,x,y,[z,w]}$, defined with link states with two marked points. These modules have both similarities and differences with the modules $\mathsf{X}_{k,\ell,x,y,z}$ that we constructed previously in [13]. A first difference is that pairs of defects originating from a given marked point may connect together in the modules $\mathsf{Y}_{k,\ell,x,y,[z,w]}$ if they encircle the second marked point, whereas

this is not allowed for the modules $\mathsf{X}_{k,\ell,x,y,z}$. A second difference is in the assignment of the weights in terms of the complex parameters $x$, $y$, $z$, and $w$, with in particular some subtleties in the way that the parameters $z$ and $w$ enter the definition. Focusing first on generic values of $z$ and $w$, we obtained the decomposition of $\mathsf{Y}_{k,\ell,x,y,[z,w]}$ over the irreducible standard modules, and used it to deduce the generic fusion rules for the fusion $\mathsf{W}_{k,x} \times \mathsf{W}_{\ell,y}$ of pairs of standard modules. The result is particularly simple, namely this fusion product can produce any irreducible standard module $\mathsf{W}_{m,t}$ with $m + k + \ell \in \mathbb{Z}$.

We also considered fusion products involving the irreducible factors that arise in the decomposition of the standard module $\mathsf{W}_{\ell,y}$ in the case where $q$ is not a root of unity and $y$ is non-generic. In this case, $\mathsf{W}_{\ell,y}$ has two composition factors: a radical submodule $\mathsf{R}_{\ell,y}$ and a quotient submodule $\mathsf{Q}_{\ell,y}$. Setting $\ell = 0$, we constructed the fusion products $(\mathsf{W}_{k,x} \times \mathsf{R}_{0,y})_{[z,w]}$ and $(\mathsf{W}_{k,x} \times \mathsf{Q}_{0,y})_{[z,w]}$, and obtained their decompositions. Although $\mathsf{R}_{0,y}$ and $\mathsf{Q}_{0,y}$ are both irreducible modules, their fusion products with a standard modules are drastically different. The modules $\mathsf{W} \times \mathsf{R}$ are always non-zero, whereas the modules $\mathsf{W} \times \mathsf{Q}$ are zero except if $x$ and $[z, w]$ satisfy certain equalities.

All of the analysis in fact relies on certain diagrams involving Jones-Wenzl projectors that we called *reduced intermediate seed states*. Their calculation is technical and requires that we use the recursion relations satisfied by the projectors to derive certain linear relations that can be solved for the intermediate seed states. We gave the full proofs of these results in a number of cases, however some cases are still left as conjectures. The arguments allowing us to obtain the decompositions of the products $\mathsf{W} \times \mathsf{R}$ and $\mathsf{W} \times \mathsf{Q}$ only require us to know whether the corresponding intermediate states vanish or not.

We believe that our construction of fusion using the modules $\mathsf{Y}_{k,\ell,x,y,[z,w]}$ is a promising candidate as a lattice version of the fusion of connectivity operators in logarithmic bulk CFT, to study correlation functions of the dense loop model with homogeneous Boltzmann weights. In the continuum scaling limit, the modules $\mathsf{Q}$ are believed to scale to an infinite direct sum of modules of the form $\mathsf{K} \otimes \bar{\mathsf{K}}$, over $\mathrm{Vir} \otimes \overline{\mathrm{Vir}}$, where $\mathsf{K}$ and $\bar{\mathsf{K}}$ are irreducible modules over $\mathrm{Vir}$ and $\overline{\mathrm{Vir}}$ respectively. We obtained the CFT fusion rules $\mathsf{W} \boxtimes \mathsf{Q}$ and $\mathsf{Q} \boxtimes \mathsf{Q}$ with this assumption and using the known fusion of degenerate and non-generate primary fields in CFT. Remarkably, this yields a result that is consistent with the fusion rules obtained using our lattice discretization with the modules $\mathsf{Y}_{k,\ell,x,y,[z,w]}$. The similar lattice discretization with $\mathsf{X}_{k,\ell,x,y,z}$ would lead to different results and would involve fewer terms. Assuming that the fusion rules for $(\mathsf{M} \times \mathsf{N}) \times \mathsf{P}$ are obtained by fusing each of the modules arising in the fusion rules $\mathsf{M} \times \mathsf{N}$ with $\mathsf{P}$ (and similarly for $\mathsf{M} \times (\mathsf{N} \times \mathsf{P})$), we showed that the fusion rules obtained using the modules $\mathsf{X}_{k,\ell,x,y,z}$ cannot be associative. In contrast, the fusion defined with the modules $\mathsf{Y}_{k,\ell,x,y,[z,w]}$ does not seem incompatible with associativity. Moreover, we found an example in (111) that showed that the prescription for fusion with the modules $\mathsf{Y}_{k,\ell,x,y,[z,w]}$ is incompatible with isomorphisms, a somewhat puzzling feature. It would be interesting to see how our construction of fusion is related to the Virasoro fusion rules in the presence of $\mathrm{O}(n)$ symmetry that were investigated recently [26].

An important unresolved question about our construction of fusion using the modules $\mathsf{Y}_{k,\ell,x,y,[z,w]}$ regards its algebraic definition. It is in fact easy to see that our diagrammatic prescription for the fusion of modules is not equivalent to the algebraic constructions proposed in [14, 15] and [16]. There are however other interesting candidates that deserve further investigations.

For $N$ odd, one such candidate is

$$\mathsf{V}(N_{\mathsf{a}}) \times \mathsf{W}(N_{\mathsf{b}}) \overset{?}{=} \mathsf{uaTL}_N(\beta, \gamma) \otimes_{\mathsf{TL}_{N_{\mathsf{a}}}(\beta) \otimes \mathsf{TL}_{N_{\mathsf{b}}}(\beta)} \big(\mathsf{V}(N_{\mathsf{a}}) \otimes \mathsf{W}(N_{\mathsf{b}})\big), \qquad N = N_{\mathsf{a}} + N_{\mathsf{b}}, \quad (238)$$

where $\mathsf{uaTL}_N(\beta, \gamma)$ is the *uncoiled affine Temperley–Lieb algebra* defined in [21], namely the algebra obtained by quotienting $\mathcal{E}\mathsf{PTL}_N(\beta)$ by the extra relation $\Omega^N = \gamma \mathbf{1}$. Interestingly, this

proposed definition involves the tensor product over the subalgebra $\mathsf{TL}_{N_\mathsf{a}}(\beta) \otimes \mathsf{TL}_{N_\mathsf{b}}(\beta)$ of $\mathsf{uaTL}_N(\beta, \gamma)$, and thus respectively considers $\mathsf{V}(N_\mathsf{a})$ and $\mathsf{W}(N_\mathsf{a})$ as modules over $\mathsf{TL}_{N_\mathsf{a}}(\beta)$ and $\mathsf{TL}_{N_\mathsf{b}}(\beta)$, even though they are also modules over the larger algebras $\mathcal{E}\mathsf{PTL}_{N_\mathsf{a}}(\beta)$ and $\mathcal{E}\mathsf{PTL}_{N_\mathsf{b}}(\beta)$. If $\mathsf{V}(N_\mathsf{a})$ and $\mathsf{W}(N_\mathsf{a})$ are finite-dimensional modules, the fusion modules defined by (238) are guaranteed to be finite-dimensional, because the uncoiled algebras are themselves finite-dimensional. An other candidate definition is obtained from (238) by replacing $\mathsf{uaTL}_N(\beta, \gamma)$ by *the uncoiled periodic Temperley–Lieb algebra* $\mathsf{upTL}_N(\beta, \gamma)$, also defined in [21], whereby the subalgebra $\{e_1, e_2, \ldots, e_N\}$ of $\mathcal{E}\mathsf{PTL}_N(\beta)$ is endowed by an extra relation that imposes an upper bound on the winding of the curves around the annulus. It remains an open question whether one of these proposals, when applied to the fusion of two standard modules, correctly reproduces the diagrammatic rules (61) of the modules $\mathsf{Y}_{k,\ell,x,y,[z,w]}$.

For $N$ even, two uncoiled affine Temperley–Lieb algebras were defined in [21]: $\mathsf{uaTL}_N^{(1)}(\beta, \alpha)$ and $\mathsf{uaTL}_N^{(2)}(\beta, \gamma)$. These differ in the choice of the quotient relations: (i) $E \Omega E = \alpha E$ and $\Omega^N = 1$ for $\mathsf{uaTL}_N^{(1)}(\beta, \alpha)$, and (ii) $E = 0$ and $\Omega^N = \gamma 1$ for $\mathsf{uaTL}_N^{(2)}(\beta, \gamma)$, where $E = e_2 e_4 \ldots e_N$. For $N$ even, one can then construct separate fusion modules of the form (238) for these two algebras. This would explain why our construction of the modules $\mathsf{Y}_{k,\ell,x,y,[z,w]}$ is so peculiar in terms of the parameters $[z, w]$ and the presence of link states of type (iv), as it attempts to embody simultaneously these two algebraic constructions. For $N$ even, there are also two uncoiled periodic Temperley–Lieb algebras, $\mathsf{upTL}_N^{(1)}(\beta, \alpha)$ and $\mathsf{upTL}_N^{(2)}(\beta, \gamma)$. More work will be needed before we can say which of these uncoiled algebras, if any, are the best to study the correlation functions of connectivity operators.

One can expect such a prescription for the fusion of two arbitrary modules to be defined as a bi-functor acting on categories of modules over the algebra. We hope that studying the properties of such a bi-functor would resolve some of the remaining problems with the fusion of standard modules presented in this paper in terms of the modules $\mathsf{Y}_{k,\ell,x,y,[z,w]}$, in particular its incompatibility with isomorphisms. We hope to return to this problem soon.

## Acknowledgments

The authors thank Alexis Langlois-Rémillard for his comments on the manuscript.

**Funding information**  AMD was supported by the EOS Research Project, project number 30889451. He thanks Ghent University for support and hospitality during the early stages of this work.

## A  Dimension counting and decomposition of $\mathbf{Y}_{k,\ell,x,y,[z,w]}(N)$

In this appendix, we sketch the arguments leading to the dimension counting in (72) and to the decomposition (83) of the modules $\mathsf{Y}_{k,\ell,x,y,[z,w]}(N)$.

To determine the dimension of $\mathsf{Y}_{k,\ell,x,y,[z,w]}(N)$, we proceed as follows. First, the link states of type (iv), namely those with no defects, are in one-to-one correspondence with the basis elements of $\mathsf{B}_0(N)$ — one simply needs to merge a and b into a single marked point. For the states of types (i), (ii) and (iii), we consider the set of *effective defects*, comprising the defects attached to a and b, and the endpoints of through lines. We call $2m$ the number of effective defects. To each link state of type (i), (ii) or (iii), we associate the state of $\mathsf{B}_m(N)$, constructed by merging a and b and turning every effective defect into a defect attached to the marked

point. To illustrate, the three states in (58) map to



$$v_1 \rightarrow \qquad , \qquad v_2 \rightarrow \qquad , \qquad v_3 \rightarrow \qquad . \tag{A.1}$$

As another example, the basis of $\mathsf{Y}_{1,0,x,y,[z,w]}(4)$ in (59) maps to

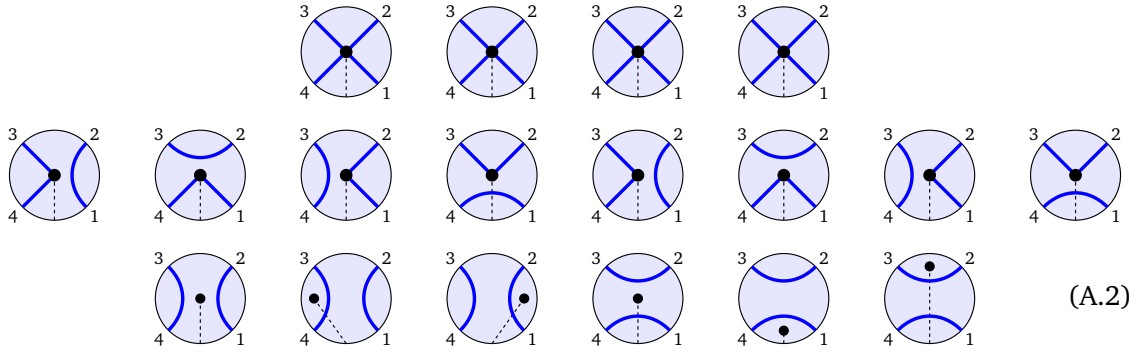

$$\tag{A.2}$$

One can argue that each state in $\mathsf{B}_m(N)$ has $2m$ pre-images under the map, because the effective defects can be assigned to a, b and through lines in a single possible order, up to $2m$ cyclic permutations. Summing over $m$ readily yields (72). A similar argument was presented in [13] to compute the dimensions of the modules $\mathsf{X}_{k,\ell,x,y,z}(N)$.

To prove the decomposition (83), we focus on the quotient modules $\mathsf{M}^{(0,r)}_{k,\ell,x,y,[z,w]}(N)$ and $\mathsf{M}^{(p)}_{k,\ell,x,y,[z,w]}(N)$. Let us first consider the module $\mathsf{M}^{(p)}_{k,\ell,x,y,[z,w]}(N = 2m)$ where $m = k+\ell+p > 0$ and $p \geqslant 0$. In this case, the module has dimension $2m$ and, from (89), we see that the braid transfer matrix acts in this module as

$$\boldsymbol{F} \cdot v \equiv (q^m \Omega + q^{-m} \Omega^{-1}) \cdot v\, [[p-1]], \qquad \forall v \in \mathsf{M}^{(p)}_{k,\ell,x,y,[z,w]}(2m). \tag{A.3}$$

As a result, the states $\Pi_{m,n} \cdot u_{k,\ell}(2m)$ for $n = 0, 1, \ldots, 2m-1$ are eigenvectors of $\boldsymbol{F}$ with respective eigenvalues $f_{m,n} = q^m \omega_{m,n} + q^{-m} \omega_{m,n}^{-1}$. For generic values of $[z, w]$, these eigenvalues are all distinct. We conclude that

$$\mathsf{M}^{(p)}_{k,\ell,x,y,[z,w]}(2m) = \bigoplus_{n=0}^{2m-1} \mathsf{W}_{m,\omega_{m,n}}(2m). \tag{A.4}$$

For $N > 2m$, we use the insertion trick to define homomorphisms

$$\Phi_{m,n} : \mathsf{W}_{m,\omega_{m,n}}(N) \rightarrow \mathsf{M}^{(p)}_{k,\ell,x,y,[z,w]}(N), \qquad n = 0, 1, \ldots, 2m-1, \tag{A.5}$$

as follows. For each link state $v \in \mathsf{W}_{m,\omega_{m,n}}$, $\Phi_{m,n}(v)$ is obtained by replacing the $2m$ defects of $v$ by the linear combination $\Pi_{m,n} \cdot u_{k,\ell}(2m)$. It is easy to see that the set of states $\{\Phi_{m,n}(v), v \in \mathsf{W}_{m,\omega_{m,n}}(N)\}$ is closed under the action of the $\mathcal{E}\mathsf{PTL}_N(\beta)$, and the map $\Phi_{m,n}$ is indeed a homomorphism. The same argument was used in [13] to obtain the decomposition of the modules $\mathsf{X}_{k,\ell,x,y,z}(N)$. This shows that each irreducible standard module $\mathsf{W}_{m,\omega_{m,n}}(N)$, with $n = 0, 1, \ldots, 2m-1$, arises as a composition factor of $\mathsf{M}^{(p)}_{k,\ell,x,y,[z,w]}(N)$. Because the sum of the dimensions of these factors exhausts the dimension of $\mathsf{M}^{(p)}_{k,\ell,x,y,[z,w]}(N)$, we conclude that

$$\mathsf{M}^{(p)}_{k,\ell,x,y,[z,w]}(N) = \bigoplus_{n=0}^{2m-1} \mathsf{W}_{m,\omega_{m,n}}(N). \tag{A.6}$$

In the special case where $k = \ell = m = 0$ and $z^{2(k-\ell)} = 1$, the decomposition is instead

$$\mathsf{M}^{(0)}_{0,0,x,y,[z,w]}(N) = \mathsf{W}_{0,w}(N). \tag{A.7}$$

This is obtained by noting that there is a trivial isomorphism between the link states of the two modules, defined by simply merging the adjacent points a and b into a single marked point.

To obtain the decomposition of the modules $\mathsf{M}^{(0,r)}_{k,\ell,x,y,[z,w]}(N)$, we note that

$$\mathsf{M}^{(0,r)}_{k,\ell,x,y,[z,w]}(N) = \mathsf{M}^{(0,r)}_{k',\ell',x,y,[z,w]}(N), \tag{A.8}$$

for some $k' < k$ and $\ell' < \ell$ satisfying $r = k' + \ell'$. Here $k'$ and $\ell'$ in fact count the number of defects of a and b, respectively, that are attached to the outer nodes, in $\mathsf{M}^{(0,r)}_{k,\ell,x,y,[z,w]}(N)$. The above statement then simply says that it is equivalent to consider a quotient module where some defects of a and b have connected, and a module where these defects were absent in the first place. The definitions of the two modules are indeed exactly the same. The resulting module $\mathsf{M}^{(0,r)}_{k',\ell',x,y,[z,w]}(N)$ can be equivalently written as $\mathsf{M}^{(0)}_{k',\ell',x,y,[z,w]}(N)$, and its decomposition can be read off from (A.6) with $p = 0$ and with $m = k' + \ell' = r$. Finally, we note that all the composition factors of the quotient modules are irreducible and non-isomorphic for generic $[z, w]$. The decomposition of $\mathsf{Y}_{k,\ell,x,y,[z,w]}(N)$ for generic $[z, w]$ is then obtained by taking the direct sum of the factors of all its quotient modules, yielding (83).

# B The fusion $\mathsf{W}_{k,x} \times \mathsf{R}_{0,\varepsilon q}$ for arbitrary system sizes

In this section, we prove Claim 4.2 for $m = 1$, namely we show that the result (118) for the fusion $\mathsf{W}_{k,x}(N_a) \times \mathsf{Q}_{0,\varepsilon q}(N_b)$, derived in Section 4.2 for $N_a = 2k$ and $N_b = 2$, holds for arbitrarily larger system sizes.

**The case $k = 0$.** In this case, $N$ is even. The gluing operator for $\big(\mathsf{W}_{0,x}(N_a) \times \mathsf{R}_{0,\varepsilon q}(N_b)\big)_{[z,w]}$ produces the seed state

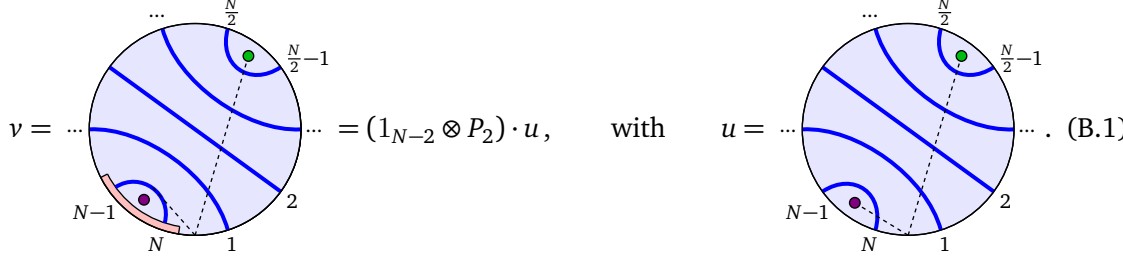

$$v = \quad = (1_{N-2} \otimes P_2) \cdot u, \qquad \text{with} \qquad u = \quad . \tag{B.1}$$

The state $v$ is a linear combination of two link states, $u$ and $e_{N-1} \cdot u$, of respective depths $p = \frac{N}{2}$ and $p = \frac{N}{2} - 1$.

We now investigate which factors in (83) belong in the submodule $\big(\mathsf{W}_{0,x}(N_a) \times \mathsf{R}_{0,\varepsilon q}(N_b)\big)_{[z,w]}$, starting with the factors $\mathsf{W}_{N/2,\omega_{N/2,n}}(N)$. From the properties of $P_2$, we have $e_{N-1} \cdot v = 0$. It is also not hard to see that

$$e_j \cdot v \equiv 0 \; [[\tfrac{N}{2} - 1]], \qquad j = 1, 2, \dots, N. \tag{B.2}$$

As a result, the states $\Pi_{N/2,n} \cdot v$ with $n = 0, 1, \dots, N-1$ generate all the one-dimensional modules $\mathsf{W}_{N/2,\omega_{N/2,n}}(N)$ in $\mathsf{M}^{(N/2)}_{0,0,x,\varepsilon q,[z,w]}(N)$. These modules are also in $\big(\mathsf{W}_{0,x}(N_a) \times \mathsf{R}_{0,\varepsilon q}(N_b)\big)_{[z,w]}$, where they are spanned by the states $Q_{N/2,n} \cdot v$. These states are indeed non-zero:

$$Q_{N/2,n} \cdot v = Q_{N/2,n} (1_{N-2} \otimes P_2) \cdot u = Q_{N/2,n} \cdot u \neq 0. \tag{B.3}$$

At the second equality, we used the fact that $Q_{N/2,n}$ evaluates to zero when acting on link states with depths $p < \frac{N}{2}$, because the submodules $\mathsf{Y}^{(p)}_{0,0,x,\varepsilon q,[z,w]}(N)$ do not contain $\mathsf{W}_{N/2,\omega_{N/2,n}}(N)$ as a composition factor. The resulting state $Q_{N/2,n} \cdot u$ is necessarily nonzero, because otherwise it would also vanish in $\mathsf{Y}_{0,0,x,\varepsilon q,[z,w]}(N)$, implying that $\mathsf{W}_{N/2,\omega_{N/2,n}}(N)$ is not a factor in this module and leading to a contradiction.

It is in fact a general feature of the fusion of a standard module with a radical submodule that the factors of maximal depth are always present in the submodule $\mathsf{W} \times \mathsf{R}$. To build states of lower depth, we define

$$v^{(1)} = \varepsilon \, \Omega \, e_{N/2} e_{N/2+1} \cdots e_{N-3} e_{N-2} \cdot v = \; \cdots \tag{B.4}$$

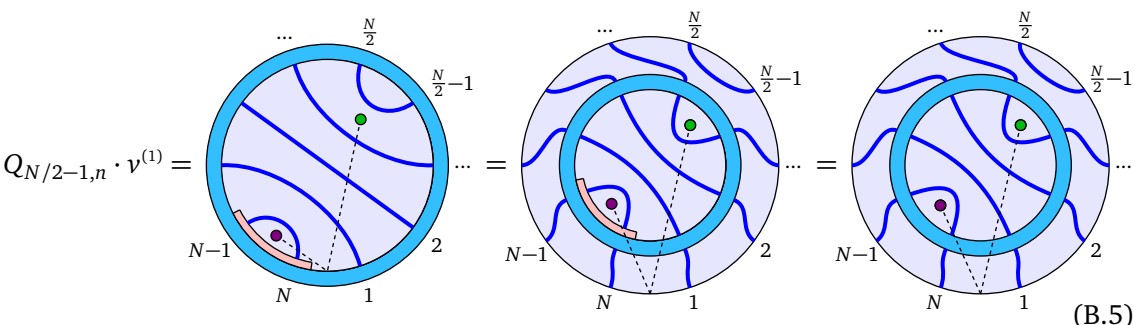

The state $v^{(1)}$ is a sum of two link states, of respective depths $\frac{N}{2} - 1$ and $\frac{N}{2} - 2$. Drawing the projector $Q_{N/2-1,n}$ as a blue ring, we apply it to $v^{(1)}$ and find

$$Q_{N/2-1,n} \cdot v^{(1)} = \quad \cdots \quad = \quad \cdots \quad = \quad \cdots \tag{B.5}$$

At the second equality, we used the push-through property (29). This results in a diagram with an inserted state consisting of a projector $Q_{N/2-1,n} \in \mathcal{E}\mathsf{PTL}_{N-2}(\beta)$ acting on a state with $N-2$ nodes, similar to $v$ but with the maximal depth one unit smaller. Then at the third equality, the Jones-Wenzl projector could be removed for the same reason as in (B.3). In the final expression, the state inside the blue ring is the state $u$ defined in (B.1), but with $N \mapsto N-2$. The resulting state is nonzero, as it was in (B.3). The state on the right side of (B.5) belongs to the submodule isomorphic to $\mathsf{W}_{N/2-1,\omega_{N/2-1,n}}(N)$. For generic $z$, $\mathsf{W}_{N/2-1,\omega_{N/2-1,n}}(N)$ is irreducible, and hence $\mathcal{E}\mathsf{PTL}_N(\beta)$ generates the entire corresponding submodule in $\mathsf{Y}_{0,0,x,\varepsilon q,[z,w]}(N)$.

The process can be iterated by defining the states

$$v^{(j)} = \varepsilon \, \Omega \, e_{N/2} e_{N/2+1} \cdots e_{N-3} e_{N-2} \cdot v^{(j-1)}, \qquad j = 1, 2, \ldots, \frac{N}{2} - 1, \tag{B.6}$$

with $v^{(0)} = v$. The state $v^{(j)}$ is obtained from $v^{(j-1)}$ by simply moving the point a one step closer to the point b. It is then a linear combination of two states, of depths $p = \frac{N}{2} - j$ and $p = \frac{N}{2} - j - 1$. This state is non-zero for $j = 1, 2, \ldots, \frac{N}{2} - 1$. This is true for $j = 0, 1, \ldots, \frac{N}{2} - 1$. This implies that each of the composition factors $\mathsf{W}_{m,\omega_{m,n}}(N)$ arises as a factor in the fusion $\left( \mathsf{W}_{0,x}(N_\mathsf{a}) \times \mathsf{R}_{0,\varepsilon q}(N_\mathsf{b}) \right)_{[z,w]}$, for $m = 1, 2, \ldots, \frac{N}{2}$ and $n = 0, 1, \ldots, 2m-1$.

The sector of depth $p = 0$ requires a separate treatment. First, we remark that any state in $\left( \mathsf{W}_{0,x}(N_\mathsf{a}) \times \mathsf{R}_{0,\varepsilon q}(N_\mathsf{b}) \right)_{[z,w]}$ with a non-zero component of depth $p > 0$ is a linear combination

of terms of the form

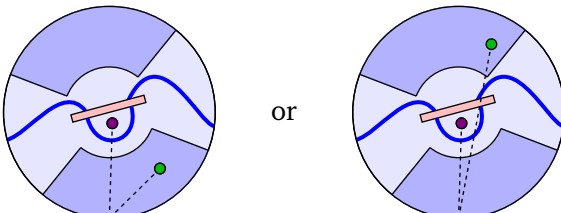

where the darker region denotes an arbitrary connectivity of the nodes inside this region. In other words, the point a is either on one side of the projector or on the other. Indeed, this is true for the states produced from the gluing. The action of the generators $e_j$ and $\Omega^{\pm 1}$ then simply moves around the two nodes of the outer perimeter connected to the projector $P_2$. This is true as long as the depth $p$ remains non-zero.

The only way to produce states of depth $p = 0$ is to connect the two loop segments connected to the projector, in such a way that the point a lies inside the region closed in the process. This produces two possible diagrams, which are equal up to a sign:

$$\left.\left(\bigodot\right)\right|_{\alpha_b = -\varepsilon\beta} = \varepsilon \left.\left(\bigodot\right)\right|_{\alpha_b = -\varepsilon\beta} = (w + w^{-1}) + \varepsilon(x + x^{-1}). \qquad (\text{B.7})$$

This is the same diagram that we computed in (114) for the special case $N_a = 2k$ and $N_b = 2$. For $w = -\varepsilon x^{\pm 1}$, this process produces a zero result. In this case, the states of depth $p = 0$ cannot be produced, and thus $W_{0,w}(N)$ is absent from the corresponding fusion module $W \times R$. In contrast, for $w \neq -\varepsilon x^{\pm 1}$, this process produces non-zero states of depth $p = 0$. The action of $\mathcal{E}PTL_N(\beta)$ then produces the entire $p = 0$ submodule, isomorphic to $W_{0,w}(N)$. As a result, for arbitrary values of $N_a$ and $N_b$ with $N = N_a + N_b$, the fusion products $W \times R$ and $W \times Q$ decompose as

$$\left(W_{0,x}(N_a) \times R_{0,\varepsilon q}(N_b)\right)_{[z,w]} \simeq \begin{cases} Y_{0,0,x,\varepsilon q,[z,w]}(N)\Big/W_{0,-\varepsilon x}(N), & w = -\varepsilon x^{\pm 1}, \\ Y_{0,0,x,\varepsilon q,[z,w]}(N), & \text{otherwise}, \end{cases} \qquad (\text{B.8})$$

and

$$\left(W_{0,x}(N_a) \times Q_{0,\varepsilon q}(N_b)\right)_{[z,w]} \simeq \begin{cases} W_{0,-\varepsilon x}(N), & w = -\varepsilon x^{\pm 1}, \\ 0, & \text{otherwise}. \end{cases} \qquad (\text{B.9})$$

**The case $k > 0$.** In this case, the seed state is

$$v = \begin{array}{c} \includegraphics{seed} \end{array} . \qquad (\text{B.10})$$

The argument to obtain the decomposition of $W \times R$ is then completely analogous to the one presented above for $k = 0$. Indeed, one first finds that the composition factors generated from states of maximal depth $p = \frac{N}{2} - k$ are all produced by the action of $\mathcal{E}PTL_N(\beta)$ on the seed state $v$. One then defines a sequence of states $v^{(j)}$ for $j = 0, 1, \ldots, \frac{N}{2} - k - 1$, with $v^{(0)} = v$, such

that the maximal depth $p$ decreases by one unit each time $j$ is increased. These states turn out to be all non-zero, implying that the composition factors of the corresponding quotient modules $M^{(p)}_{k,0,x,\varepsilon q,[z,w]}$ with $p \geqslant 1$ are all present in the fusion product $W \times R$. The special case of depth $p = 0$ is again treated separately, and the needed relation is (104) instead of (108). The resulting decomposition for $k > 0$ and generic $[z, w]$ is

$$\left(W_{k,x}(N_{\mathsf{a}}) \times R_{0,\varepsilon q}(N_{\mathsf{b}})\right)_{[z,w]} \simeq \begin{cases} \mathsf{Y}_{k,0,x,\varepsilon q,[z,w]}(N)\Big/ \mathsf{W}_{k,-\varepsilon x}(N), & z^{2k} = (-\varepsilon x)^{2k}, \\ \mathsf{Y}_{k,0,x,\varepsilon q,[z,w]}(N), & \text{otherwise.} \end{cases} \tag{B.11}$$

## C  The reduced intermediate seed states $\sigma^{(p)}_{0,m}$

In this appendix, we investigate more closely the reduced intermediate seed states $\sigma^{(p)}_{0,m}$ corresponding to the case $k = 0$:

$$\sigma^{(p)}_{0,m} = \qquad \qquad \qquad . \tag{C.1}$$

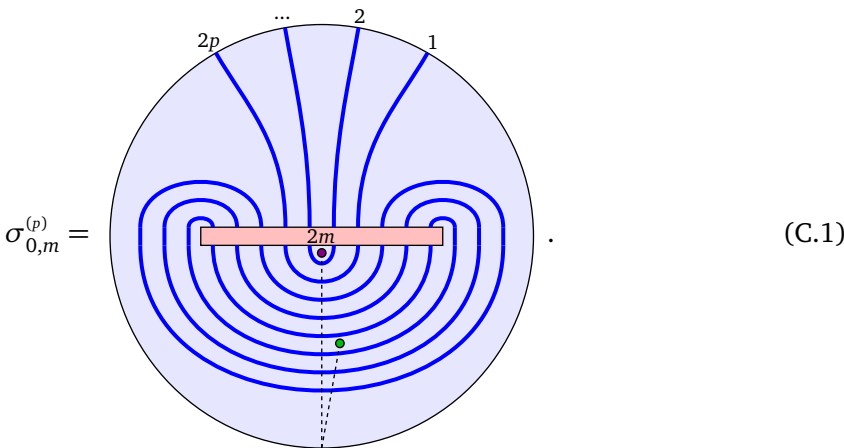

We show that these diagrams satisfy certain recursion relations that determine them exactly. In contrast with the states $\sigma^{(p)}_{k,m}$ and $\sigma^{(0,r)}_{k,m}$ with $k > m-1$, we are not able to obtain a recursion relation for $\sigma^{(p)}_{0,m}$ where the depth $p$ varies. Instead, we consider a fixed value of $p$ and derive recursion relations where $m$ varies. The following proposition gives those relations in the case $p \geqslant 1$.

PROPOSITION C.1 *For $1 \leqslant p \leqslant m$, the state $\sigma^{(p)}_{0,m}$ satisfies*

$$\sigma^{(p)}_{0,m} \equiv X^{(p)}_m(\Omega, x, y) \cdot u_{0,0}(2p) \quad [[p-1]], \tag{C.2}$$

*where $X^{(p)}_m(\omega, x, y)$ is a polynomial in the variables $\alpha_{\mathsf{a}} = x + x^{-1}$, $\alpha_{\mathsf{b}} = y + y^{-1}$ and $\alpha_\omega = \omega + \omega^{-1}$, that satisfies the recursion relation for $m \geqslant p + 1$*

$$X^{(p)}_{m+1} = \left(\alpha_{\mathsf{a}} + \frac{\alpha_p \alpha_{\mathsf{b}} \alpha_\omega}{\{m\}\{m+1\}}\right) X^{(p)}_m + \frac{(\alpha_p^2 - \{m\}^2)(\alpha_{\mathsf{b}}^2 - \{m\}^2)(\alpha_\omega^2 - \{m\}^2)}{(q - q^{-1})^2 [2m-1][2m+1]\{m\}^2} X^{(p)}_{m-1}, \tag{C.3}$$

*and the initial conditions*

$$X^{(p)}_p = 1, \qquad X^{(p)}_{p+1} = \alpha_{\mathsf{a}} + \frac{\alpha_{\mathsf{b}} \alpha_\omega}{\{p+1\}}, \tag{C.4}$$

*where $\alpha_p = q^p + q^{-p}$ and $\{m\} = q^m + q^{-m}$.*

The analysis of the decompositions of the fusion modules $W \times R$ and $W \times Q$ in Section 4 requires that we evaluate $X^{(p)}_m(\omega, x, y)$ at $y = \varepsilon q^m$. We formulate the following conjecture.

CONJECTURE C.1 *For $p \geqslant 1$, the function $X_m^{(p)}(\omega, x, y)$ specialised to $y = \varepsilon q^m$ evaluates to*

$$X_m^{(p)}(\omega, x, \varepsilon q^m) = (\omega x)^{p-m} \prod_{r=-\frac{m-p-1}{2}}^{\frac{m-p-1}{2}} (xq^{2r} + \varepsilon \omega)(\varepsilon q^{2r} + x\omega). \tag{C.5}$$

This linear system of Proposition C.1 has a simple solution for $X_m^{(p)}(\omega, x, y)$ in terms of the determinant of a tri-diagonal matrix. By implementing the determinant expression on a computer, we verified explicitly that Conjecture C.1 holds for $m = p, p+1, \ldots, p+7$. We now proceed to prove Proposition C.1.

PROOF OF PROPOSITION C.1. The main ingredient for the proof is the set of four recursion relations (27) satisfied by the Jones-Wenzl projectors, as well as the last relation in (26).

We first consider the case $m \geqslant p + 2$. Using the relation (27c) on $P_{2m}$, we find

$$\sigma_{0,m}^{(p)} = y_m^{(p)} + \frac{\alpha_{\mathsf{b}}[m]}{[2m]} z_m^{(p)}, \tag{C.6}$$

where

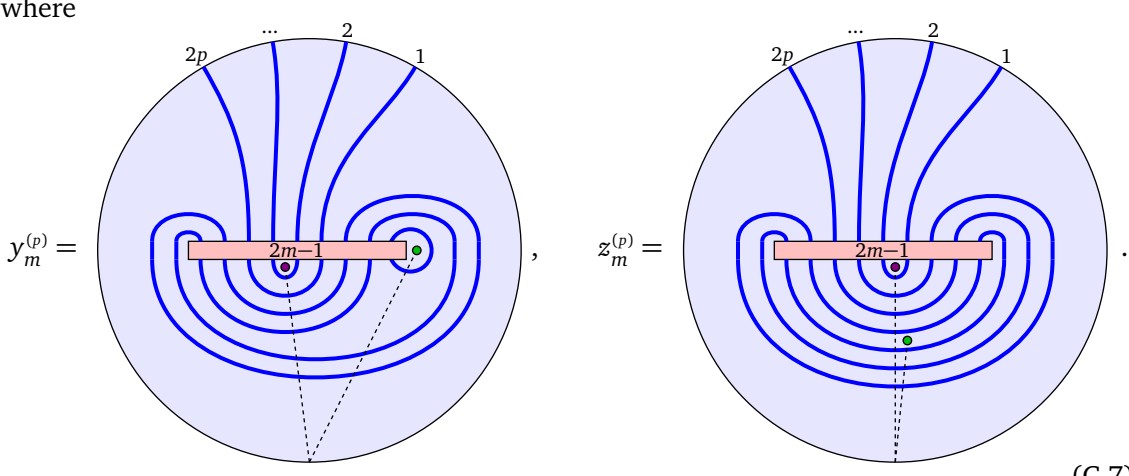

$$\tag{C.7}$$

Writing $z_m^{(p)} = P_{2p} \cdot z_m^{(p)}$, we use (27a) on $P_{2m-1}$ and find

$$z_m^{(p)} = P_{2p} \frac{[m+p]\Omega + [m-p]\Omega^{-1}}{[2m-1]} \cdot \sigma_{0,m-1}^{(p)}. \tag{C.8}$$

For $y_m^{(p)}$, applying (27d) to the projector $P_{2m-1}$ yields

$$y_m^{(p)} = \alpha_{\mathsf{a}} \, \sigma_{0,m-1}^{(p)} + \frac{[2m-2]}{[2m-1]} \widehat{y}_{m-1}^{(p)} + \frac{\alpha_{\mathsf{b}}[m-1]}{[2m-1]} \widetilde{y}_{m-1}^{(p)}, \tag{C.9}$$

where

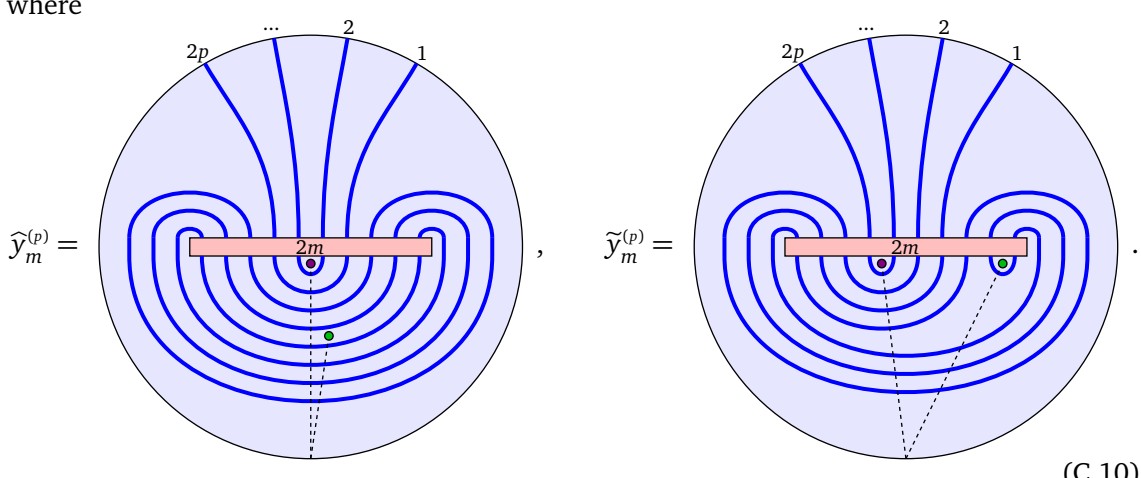

$$\tag{C.10}$$

Writing $\widehat{y}_m^{(p)} = P_{2p} \cdot \widehat{y}_m^{(p)}$ and using (27b) for $P_{2m}$, we obtain

$$\widehat{y}_m^{(p)} = -\frac{[2m]}{[2m-1]}\,\sigma_{0,m-1}^{(p)} + P_{2p}\,\frac{[m-p]\Omega+[m+p]\Omega^{-1}}{[2m]}\cdot z_m^{(p)}, \tag{C.11}$$

where the first term was simplified using (26). Similarly, we write $\widetilde{y}_m^{(p)} = P_{2p} \cdot \widetilde{y}_m^{(p)}$, use the relation (27b) for the projector $P_{2m}$, and find

$$\widetilde{y}_m^{(p)} = w_m^{(p)} + P_{2p}\,\frac{[m-p]\Omega+[m+p]\Omega^{-1}}{[2m]}\cdot y_m^{(p)}, \tag{C.12}$$

where

$$w_m^{(p)} = \quad\text{}\quad . \tag{C.13}$$

Using the relation (27b) on $P_{2m-1}$, we get

$$w_m^{(p)} = \frac{\alpha_{\mathsf{b}}[m]}{[2m-1]}\,\sigma_{0,m-1}^{(p)}. \tag{C.14}$$

Thus, for $m \geqslant p+2$, we obtain a closed system of recursion relations. Indeed, using (C.11), (C.12) and (C.14) to express $\widehat{y}_{m-1}^{(p)}$ and $\widetilde{y}_{m-1}^{(p)}$ in terms of the other variables, we find

$$\sigma_{0,m}^{(p)} = y_m^{(p)} + \frac{\alpha_{\mathsf{b}}[m]}{[2m]}\,z_m^{(p)}, \tag{C.15a}$$

$$\begin{aligned}y_m^{(p)} = {}&\alpha_{\mathsf{a}}\,\sigma_{0,m-1}^{(p)} + \frac{\alpha_{\mathsf{b}}^2[m-1]^2-[2m-2]^2}{[2m-1][2m-3]}\,\sigma_{0,m-2}^{(p)}\\&+ P_{2p}\,\frac{[m-1-p]\Omega+[m-1+p]\Omega^{-1}}{[2m-1]}\cdot\left(z_{m-1}^{(p)}+\frac{\alpha_{\mathsf{b}}[m-1]}{[2m-2]}\,y_{m-1}^{(p)}\right),\end{aligned} \tag{C.15b}$$

$$z_m^{(p)} = P_{2p}\,\frac{[m+p]\Omega+[m-p]\Omega^{-1}}{[2m-1]}\cdot\sigma_{0,m-1}^{(p)}. \tag{C.15c}$$

Eliminating $y_m^{(p)}$ and $z_m^{(p)}$, we obtain a recursion relation involving only the function $\sigma_{0,m}^{(p)}$ evaluated at different values of $m$:

$$\begin{aligned}\sigma_{0,m}^{(p)} = {}&\left(\alpha_{\mathsf{a}}1 + \frac{\alpha_{\mathsf{b}}[2p][m][m-1]P_{2p}(\Omega+\Omega^{-1})}{[p][2m][2m-2]}\right)\cdot\sigma_{0,m-1}^{(p)}\\&-\left(\frac{(\alpha_{\mathsf{b}}^2[m-1]^2-[2m-2]^2)(P_{2p}\Omega_{m-1,-p}P_{2p}\Omega_{m-1,p}-[2m-2]^21)}{[2m-1][2m-2]^2[2m-3]}\right)\cdot\sigma_{0,m-2}^{(p)},\end{aligned} \tag{C.16}$$

where we use the notation

$$\Omega_{m,p} = [m+p]\Omega+[m-p]\Omega^{-1}. \tag{C.17}$$

Applying the relation (27b) on $P_{2p}$, we get

$$P_{2p}\,\Omega \cdot \sigma^{(p)}_{0,m} = \Omega \cdot \sigma^{(p)}_{0,m} + \frac{[2p-1]}{[2p]}\,(P_{2p-1} \otimes 1_1)\,\Omega\,e_{2p} \cdot \sigma^{(p-1)}_{0,m}\,. \tag{C.18}$$

The second term is a combination of states with depths strictly smaller than $p$. A similar calculation can be made for $P_{2p}\,\Omega^{-1} \cdot \sigma^{(p)}_{0,m}$. As a result, we have

$$P_{2p}\,\Omega^{\pm 1} \cdot \sigma^{(p)}_{0,m} \equiv \Omega^{\pm 1} \cdot \sigma^{(p)}_{0,m}\,[[p-1]]\,. \tag{C.19}$$

Thus, for $m \geqslant p+2$, we obtain the recursion relation

$$\sigma^{(p)}_{0,m} \equiv \left(\alpha_{\mathsf{a}} 1 + \frac{\alpha_{\mathsf{b}}[2p][m][m-1](\Omega + \Omega^{-1})}{[p][2m][2m-2]}\right) \cdot \sigma^{(p)}_{0,m-1} \tag{C.20}$$

$$-\frac{[m-1-p][m-1+p][m-1]^2}{[2m-1][2m-2]^2[2m-3]}\left(\alpha_{\mathsf{b}}^2 - \frac{[2m-2]^2}{[m-1]^2}\right)\left((\Omega + \Omega^{-1})^2 - \frac{[2m-2]^2}{[m-1]^2}\right) \cdot \sigma^{(p)}_{0,m-2}\,[[p-1]]\,.$$

Let us now examine the initial conditions, namely the expressions for $\sigma^{(p)}_{0,p}$ and $\sigma^{(p)}_{0,p+1}$. For $m = p$, we have $\sigma^{(p)}_{0,p} = P_{2p} \cdot u_{0,0}(2p)$ directly from the definition of $\sigma^{(p)}_{0,p}$. For $m = p+1$, the above derivation can be easily adapted and, up to terms of depth smaller than $p$, we get

$$\sigma^{(p)}_{0,p+1} = y^{(p)}_{p+1} + \frac{\alpha_{\mathsf{b}}[p+1]}{[2p+2]}\,z^{(p)}_{p+1}\,, \tag{C.21a}$$

$$y^{(p)}_{p+1} \equiv \alpha_{\mathsf{a}}\sigma^{(p)}_{0,p} + \frac{\alpha_{\mathsf{b}}[p]}{[2p+1]}\,P_{2p}\Omega \cdot u_{0,0}(2p)\,[[p-1]]\,, \tag{C.21b}$$

$$z^{(p)}_{p+1} = P_{2p}\,\frac{[2p+1]\Omega + [1]\Omega^{-1}}{[2p+1]} \cdot \sigma^{(p)}_{0,p}\,. \tag{C.21c}$$

After some simple manipulations, this yields

$$\sigma^{(p)}_{0,p+1} \equiv \left(\alpha_{\mathsf{a}} 1 + \frac{\alpha_{\mathsf{b}}[p+1](\Omega + \Omega^{-1})}{[2p+2]}\right) \cdot u_{0,0}(2p)\,[[p-1]]\,. \tag{C.22}$$

Using $u_{0,0}(2p) \equiv \sigma^{(p)}_{0,p}\,[[p-1]]$, we readily see that (C.22) coincides with (C.20) specified to $m = p+1$ and with $\sigma^{(p)}_{0,p-1} = 0$. Hence, we set the initial conditions of the recursive system to be

$$\sigma^{(p)}_{0,p-1} = 0\,, \qquad \sigma^{(p)}_{0,p} = P_{2p} \cdot u_{0,0}(2p)\,. \tag{C.23}$$

Using (C.20), we prove inductively on increasing values of $m$ that, modulo states of depth less than $p$, the state $\sigma^{(p)}_{0,m}$ is proportional to $u_{0,0}(2p)$, as in (C.2). Finally, by applying a shift $m \mapsto m+1$ in (C.20) and expressing it as a recursive relation satisfied by the functions $X^{(p)}_m(\omega, x, y)$, we obtain (C.3), thus ending the proof of Proposition C.1. ∎

In the case $p = 0$, the diagram in (C.1) has no loop segments connected to the perimeter of the disc, and each of the corresponding seed state $\sigma^{(0)}_{0,m}$ is a constant. Arguments similar to those presented above lead to the following results.

PROPOSITION C.2 *The seed states $\sigma^{(0)}_{0,m}$ with $m \geqslant 0$ are polynomials in $\alpha_{\mathsf{a}} = x + x^{-1}$, $\alpha_{\mathsf{b}} = y + y^{-1}$ and $\alpha_{\mathsf{ab}} = w + w^{-1}$ that satisfy the linear system for $m \geqslant 1$*

$$\sigma^{(0)}_{0,m+1} = \left(\alpha_{\mathsf{a}} + \frac{2\alpha_{\mathsf{b}}\alpha_{\mathsf{ab}}}{\{m\}\{m+1\}}\right)\sigma^{(0)}_{0,m} - \frac{[m]^2(\alpha_{\mathsf{b}}^2 - \{m\}^2)(\alpha_{\mathsf{ab}}^2 - \{m\}^2)}{[2m+1][2m-1]\{m\}^2}\,\sigma^{(0)}_{0,m-1}\,, \tag{C.24}$$

*with the initial conditions*

$$\sigma^{(0)}_{0,0} = 1\,, \qquad \sigma^{(0)}_{0,1} = \alpha_{\mathsf{a}} + \frac{\alpha_{\mathsf{b}}\alpha_{\mathsf{ab}}}{\{1\}}\,, \tag{C.25}$$

*where $\{m\} = q^m + q^{-m}$.*

CONJECTURE C.2 *The intermediate seed states $\sigma_{0,m}^{(0)}(w,x,y)$ specialised to $y = \varepsilon q^m$ evaluate to*

$$\sigma_{0,m}^{(0)}(w,x,\varepsilon q^m) = (wx)^{-m} \prod_{r=-\frac{m-1}{2}}^{\frac{m-1}{2}} (xq^{2r} + \varepsilon w)(\varepsilon q^{2r} + xw). \qquad \text{(C.26)}$$

We observe that this system of equations is obtained from the similar system of Proposition C.1 by taking the limit $p \to 0$ and setting $\omega \to w$. The solution of Proposition C.2 for $\sigma_{0,m}^{(0)}(w,x,y)$ can again be expressed as the determinant of a tri-diagonal matrix. Implementing this on a computer, we checked that Conjecture C.2 holds for $m = 0,1,\dots,7$.

# D   The structure of $Y_{0,0,x,y,[z,w]}(N)$ for non-generic $w$

In this appendix, we prove the decomposition (185) of $Y_{0,0,x,y,[z,w]}(N)$ for $N = \frac{m}{2}$. This result can then be extended straightforwardly to $N > \frac{m}{2}$ by using the insertion algorithm. For this derivation, we use the projector

$$\widehat{Q} = \prod_{\substack{r=1 \\ (r,n)\neq(m,n_\varepsilon)}}^{m} \prod_{n=0}^{2r-1} \frac{F - f_{r,n}\,1}{f_0 - f_{r,n}}. \qquad \text{(D.1)}$$

In particular, we have $Z_{0,0,x,y,[z,w=\varepsilon q^m]}(2m) = \widehat{Q}\cdot Y_{0,0,x,y,[z,w=\varepsilon q^m]}(2m)$. Let us consider the pair of elements of $Z_{0,0,x,y,[z,w=\varepsilon q^m]}$ defined as

$$\psi = \widehat{Q}\cdot v_{0,0}(2m), \qquad \lambda = (F - f_0 1)\cdot\psi, \qquad \text{(D.2)}$$

where $v_{0,0}(2m) = P_{2m}\cdot u_{0,0}(2m)$. The action of $F$ on the pair $(\psi,\lambda)$ reads

$$F\cdot\psi = f_0\,\psi + \lambda, \qquad F\cdot\lambda = f_0\,\lambda. \qquad \text{(D.3)}$$

Moreover, because

$$F\cdot v \equiv \left(q^m\Omega + q^{-m}\Omega^{-1}\right)\cdot v\,[[m-1]] \qquad \forall v \in Y_{0,0,x,y,[z,w]}(2m), \qquad \text{(D.4)}$$

we write for $v \in \{0,\dots,2m-1\}$

$$\Pi_{m,v}\widehat{Q}\cdot v \equiv \prod_{\substack{r=1 \\ (r,n)\neq(m,n_\varepsilon)}}^{m} \prod_{n=0}^{2r-1} \frac{q^m\omega_{m,v} + q^{-m}\omega_{m,v}^{-1} - f_{r,n}}{f_0 - f_{r,n}} \Pi_{m,v}\cdot v$$

$$\equiv \prod_{\substack{r=1 \\ (r,n)\neq(m,n_\varepsilon)}}^{m} \prod_{n=0}^{2r-1} \frac{f_{m,v} - f_{r,n}}{f_0 - f_{r,n}} \Pi_{m,v}\cdot v \equiv \delta_{v,n_\varepsilon} \Pi_{m,v}\cdot v\,[[m-1]], \qquad \text{(D.5)}$$

and therefore

$$\psi = \sum_{v=0}^{2m-1} \Pi_{m,v}\widehat{Q}\cdot v_{0,0}(2m) \equiv \Pi_{m,n_\varepsilon}\cdot v_{0,0}(2m) \equiv \Pi_{m,n_\varepsilon}\cdot u_{0,0}(2m)\,[[m-1]]. \qquad \text{(D.6)}$$

We conclude that $\psi$ is nonzero.

Let us first prove that $\lambda \in Y^{(0)}_{0,0,x,y,[z,w=\varepsilon q^m]}(2m)$. For any $v \in Y_{0,0,x,y,[z,w]}(2m)$, we have

$$
\begin{aligned}
\prod_{n=0}^{2m-1}(F - f_{m,n}1) \cdot v &= \sum_{\nu=0}^{2m-1} \Pi_{m,\nu} \prod_{n=0}^{2m-1}(F - f_{m,n}1) \cdot v \\
&\equiv \sum_{\nu=0}^{2m-1} \prod_{n=0}^{2m-1}(q^m \omega_{m,\nu} + q^{-m}\omega_{m,\nu}^{-1} - f_{m,n}) \Pi_{m,\nu} \cdot v \; [[m-1]] \\
&\equiv \sum_{\nu=0}^{2m-1} \prod_{n=0}^{2m-1}(f_{m,\nu} - f_{m,n}) \Pi_{m,\nu} \cdot v \equiv 0 \; [[m-1]].
\end{aligned}
\tag{D.7}
$$

This implies that

$$
\prod_{n=0}^{2m-1}(F - f_{m,n}1) \cdot v \in Y^{(m-1)}_{0,0,x,y,[z,w]}(2m).
\tag{D.8}
$$

Let $m' \in \{1, \ldots, m-1\}$. Any link state $v' \in Y^{(m')}_{0,0,x,y,[z,w]}(2m)$ is of the form $v' = c^{\dagger}_{j_1} \ldots c^{\dagger}_{j_{m-m'}} \cdot v''$, where $j_1, \ldots, j_{m-m'} \in \{0, \ldots, 2m-1\}$ and $v'' \in Y_{0,0,x,y,[z,w]}(2m')$. The push-through property (29) implies that $F c^{\dagger}_j = c^{\dagger}_j F$, where the central elements $F$ on the left and right sides belong to $\mathcal{E}PTL_N(\beta)$ and $\mathcal{E}PTL_{N-2}(\beta)$, respectively. Using this, we find

$$
\prod_{n=0}^{2m'-1}(F - f_{m',n}1) \cdot v' = c^{\dagger}_{j_1} \ldots c^{\dagger}_{j_{m-m'}} \prod_{n=0}^{2m'-1}(F - f_{m',n}1) \cdot v'' \equiv 0 \; [[m'-1]],
\tag{D.9}
$$

where the elements $F$ and $1$ are elements of $\mathcal{E}PTL_{2m}(\beta)$ and $\mathcal{E}PTL_{2m'}(\beta)$ before and after the first equality, respectively. We conclude that

$$
\prod_{n=0}^{2m'-1}(F - f_{m',n}1) \cdot v' \in Y^{(m'-1)}_{0,0,x,y,[z,w]}(2m).
\tag{D.10}
$$

Thus, each subsequent application of $\prod_{n=0}^{2r-1}(F - f_{r,n}1)$ on $v_{0,0}(2m)$ decreases the value of $p$ by one unit in the filtration of modules $Y^{(p)}_{0,0,x,y,[z,w]}$. We conclude that $\lambda$ belongs to the submodule of depth $p = 0$.

Let us now derive a crucial property of $\lambda$, namely its factorisation as

$$
\lambda = \left[ (xy)^{-m} \prod_{s=-\frac{m-1}{2}}^{\frac{m-1}{2}} (y + \varepsilon x q^{2s})(\varepsilon xy + q^{2s}) \right] \mu,
\tag{D.11}
$$

where $\mu \in Y^{(0)}_{0,0,x,y,[z,w=\varepsilon q^m]}$ is a state independent of $x$ and $y$. We illustrate in detail this derivation in the cases $m = 1$ and $m = 2$ before describing the general case. Here, in analogy with (45), we introduce the states

$$
\widehat{v}_0(2m) = (c^{\dagger}_0)^m \cdot u_{0,0}(0), \qquad \widehat{w}_0(2m) = P_{2m} \cdot \widehat{v}_0(2m).
\tag{D.12}
$$

- For $m = 1$, we have

$$
\lambda = \frac{1}{2\varepsilon(q + q^{-1})}(F - f_{1,0}1)(F - f_{1,1}1) \cdot v_{0,0}(2).
\tag{D.13}
$$

The action of $F$ yields

$$
F \cdot v_{0,0}(2) = (q \Omega + q^{-1}\Omega^{-1} + c^{\dagger}_1 c_0) \cdot v_{0,0}(2), \qquad F c_0 \cdot v_{0,0}(2) = f_0 c_0 \cdot v_{0,0}(2).
\tag{D.14}
$$

Hence, writing $\lambda = (\Pi_{1,0} + \Pi_{1,1}) \cdot \lambda$, we find after some simple manipulations

$$\lambda = \Pi_{1,n_\varepsilon} c_1^\dagger c_0 \cdot v_{0,0}(2) = \left(c_0 \cdot v_{0,0}(2)\right)\left(\Pi_{1,n_\varepsilon} c_1^\dagger \cdot u_{0,0}(0)\right). \tag{D.15}$$

The diagram $c_0 \cdot v_{0,0}(2)$ is easily computed and reads

$$c_0 \cdot v_{0,0}(2) = \alpha_{\mathsf a} - \frac{\alpha_{\mathsf b}\alpha_{\mathsf{ab}}}{\beta} = (x + x^{-1}) + \varepsilon(y + y^{-1}) = (xy)^{-1}(y + \varepsilon x)(\varepsilon xy + 1). \tag{D.16}$$

Furthermore, the state $\mu = \Pi_{1,n_\varepsilon} c_1^\dagger \cdot u_{0,0}(0)$ is clearly nonzero.

- For $m = 2$, we have

$$\lambda = \prod_{\substack{r=1 \\ (r,n)\neq(m,n_\varepsilon)}}^{2} \prod_{n=0}^{2r-1} (f_0 - f_{r,n})^{-1} \prod_{r=1}^{2} \prod_{n=0}^{2r-1} (F - f_{r,n}1) \cdot v_{0,0}(4). \tag{D.17}$$

The action of $F$ yields

$$F \cdot v_{0,0}(4) = (q^2 \Omega + q^{-2}\Omega^{-1} + C_2^\dagger c_0) \cdot v_{0,0}(4), \tag{D.18a}$$

$$F c_0 \cdot v_{0,0}(4) = (q \Omega + q^{-1}\Omega^{-1} + c_1^\dagger c_0) c_0 \cdot v_{0,0}(4), \tag{D.18b}$$

$$F c_0^2 \cdot v_{0,0}(4) = f_0 c_0^2 \cdot v_{0,0}(4), \tag{D.18c}$$

where $C_2^\dagger = q^{-1}c_1^\dagger + c_2^\dagger + qc_3^\dagger$. Inserting the projectors onto $\Omega$ eigenspaces, we obtain after some manipulations

$$\lambda = \left(c_0^2 \cdot v_{0,0}(4)\right)\mu, \qquad \text{where} \qquad \mu = \sum_{\nu=0,1} (f_0 - f_{1,\nu})^{-1} \Pi_{2,n_\varepsilon} C_2^\dagger \Pi_{1,\nu} c_1^\dagger \cdot u_{0,0}(0). \tag{D.19}$$

We identify the diagram $c_0^2 \cdot v_{0,0}(4)$ as the intermediate seed state $\sigma_{0,2}^{(0)}(w, x, y)$ specialised to $w = \varepsilon q^m$. Moreover, it is clear from (C.24) and (C.25) that $\sigma_{0,m}^{(0)}(w, x, y)$ is invariant under the exchange $y \leftrightarrow w$. This allows us to evaluate $c_0^2 \cdot v_{0,0}(4)$ directly from (143b):

$$c_0^2 \cdot v_{0,0}(4) = (yx)^{-2}(y + \varepsilon xq)(y + \varepsilon xq^{-1})(\varepsilon xy + q)(\varepsilon xy + q^{-1}). \tag{D.20}$$

Applying $P_4$ on (D.19), we find after some simplifications

$$P_4 \cdot \mu = \frac{\varepsilon/2}{(q - q^{-1})^2} \widehat{w}_0(4), \tag{D.21}$$

and hence $\mu$ is nonzero.

- For $m \geqslant 2$, the derivation follows the same ideas. The action of $F$ yields

$$F c_0^{m-r} \cdot v_{0,0}(2m) = \begin{cases} (q^r \Omega + q^{-r}\Omega^{-1} + C_r^\dagger c_0) c_0^{m-r} \cdot v_{0,0}(2m), & r \in \{1, \ldots, m\}, \\ f_0 c_0^m \cdot v_{0,0}(2m), & r = 0, \end{cases} \tag{D.22}$$

where

$$C_r^\dagger = \sum_{j=1}^{2r-1} q^{j-r} c_j^\dagger. \tag{D.23}$$

After a short calculation, we find

$$\lambda = \left(c_0^m \cdot v_{0,0}(2m)\right)\mu, \qquad \text{where} \qquad \mu = \Pi_{m,n_\varepsilon} C_m^\dagger \widehat{C}_{m-1}^\dagger \widehat{C}_{m-2}^\dagger \ldots \widehat{C}_1^\dagger \cdot u_{0,0}(0), \tag{D.24}$$

and

$$\widehat{C}_r^\dagger = \sum_{v=0}^{2r-1} (f_0 - f_{r,v})^{-1} \Pi_{r,v} C_r^\dagger. \tag{D.25}$$

Using the reflection symmetry of $P_{2m}$, one finds that $c_0^m \cdot v_{0,0}(2m) = \sigma_{0,m}^{(0)}(w,x,y)$ with $w = \varepsilon q^m$. Using $\sigma_{0,m}^{(0)}(w,x,y) = \sigma_{0,m}^{(0)}(y,x,w)$ and (176c), we directly obtain (D.11). Finally, we compute $P_{2m} \cdot \mu$ using repeatedly the identity

$$P_{2m} (c_0^\dagger)^{m-r} \Pi_{r,v} C_r^\dagger = \frac{1}{2r} \sum_{k=1}^{2r-1} \omega_{r,v}^{-k} q^{k-r} P_{2m} (c_0^\dagger)^{m-r+1} \Omega^{k-1}, \qquad r = 1, 2, \ldots, m. \tag{D.26}$$

This yields

$$P_{2m} \cdot \mu = \gamma_m \widehat{w}_0(2m), \quad \gamma_m = \frac{\varepsilon^m}{2^m m!} \sum_{\substack{k_1, \ldots, k_m \\ k_r \in \{-r+1, -r+2, \ldots, r-1\}}} (\varepsilon q)^{k_m} \prod_{r=1}^{m-1} \left( q^{k_r} \sum_{v=0}^{2r-1} \frac{\omega_{r,v}^{k_{r+1}-k_r}}{f_{m,n_\varepsilon} - f_{r,v}} \right). \tag{D.27}$$

The following proposition states that this formula for $\gamma_m$ simplifies to a much simpler expression.

PROPOSITION D.1 *The constant $\gamma_m$ is given by*

$$\gamma_m = \frac{\varepsilon/2}{\prod_{j=1}^{m-1} (q^j - q^{-j})^2}. \tag{D.28}$$

The proof is given at the end of the section.

We use (D.3) and (D.11) to determine the structure of $Z_{0,0,x,y,[z,w=\varepsilon q^m]}(2m)$ as a function of $x$ and $y$. We recall that the submodule $Y_{0,0,x,y,[z,w=\varepsilon q^m]}^{(0)}$ is always isomorphic to $W_{0,\varepsilon q^m}$, which has the structure

$$W_{0,\varepsilon q^m} \simeq \begin{bmatrix} Q_{0,\varepsilon q^m} \\ \searrow \\ R_{0,\varepsilon q^m} \end{bmatrix}. \tag{D.29}$$

The analysis distinguishes between two cases.

- Case (i): for $y = -\varepsilon x^{\pm 1} q^{2s}$ with $s \in \{-\frac{m-1}{2}, -\frac{m-3}{2}, \ldots, \frac{m-1}{2}\}$. We now proceed to prove that the state $\Pi_{m,n_\varepsilon} \cdot \psi$ generates a one-dimensional submodule isomorphic to $W_{m,\varepsilon}(2m)$. First, recall that $\psi \equiv \Pi_{m,n_\varepsilon} \cdot u_{0,0}(2m) [[m-1]]$, and hence $\Pi_{m,n_\varepsilon} \cdot \psi$ is nonzero. One readily computes the action of $e_j$ on $\psi$ for $j = 1, 2, \ldots, 2m-1$ as

$$e_j \cdot \psi = e_j \widehat{Q} P_{2m} \cdot u_{0,0}(2m) = \widehat{Q} e_j P_{2m} \cdot u_{0,0}(2m) = 0. \tag{D.30}$$

Moreover, using (D.22), we obtain

$$e_0 \cdot \psi = c_0^\dagger \widehat{Q} c_0 \cdot v_{0,0}(2m) = \left( c_0^m \cdot v_{0,0}(2m) \right) c_0^\dagger \widehat{C}_{m-1}^\dagger \ldots \widehat{C}_1^\dagger \cdot u_{0,0}(0). \tag{D.31}$$

As pointed out above in the calculation of $\lambda$, the prefactor $c_0^m \cdot v_{0,0}(2m)$ is in fact the intermediate seed state $\sigma_{0,m}^{(0)}(w = \varepsilon q^m, x, y)$ — it is the factor in the bracket in (D.11) and thus vanishes for $y = -\varepsilon x^{\pm 1} q^{2s}$ with $s \in \{-\frac{m-1}{2}, -\frac{m-3}{2}, \ldots, \frac{m-1}{2}\}$. We conclude that $e_0 \cdot \psi = 0$. Hence, the generators of $\mathcal{E}PTL_{2m}(\beta)$ act on $\Pi_{m,n_\varepsilon} \cdot \psi$ as

$$\Omega \Pi_{m,n_\varepsilon} \cdot \psi = \varepsilon \Pi_{m,n_\varepsilon} \cdot \psi, \qquad e_j \Pi_{m,n_\varepsilon} \cdot \psi = 0 \quad \text{for} \quad j = 0, \ldots, 2m-1, \tag{D.32}$$

which is indeed the action in $W_{m,n_\varepsilon}(2m)$. Putting the above results together, we find

$$Z_{0,0,x,y,[z,w=\varepsilon q^m]} \simeq \begin{bmatrix} \begin{array}{c} Q_{0,\varepsilon q^m} \\ \searrow \\ \phantom{xxx} R_{0,\varepsilon q^m} \end{array} \end{bmatrix} \oplus W_{m,\varepsilon} . \tag{D.33}$$

- Case (ii): for all the other values of $y$. The structure in this case was previously discussed in [13]. Let us consider the state $e_0 \cdot \psi$. It is clear from (D.31) that this state is a linear combination of link states of zero depth. It is therefore an element of the submodule $Y^{(0)}_{0,0,x,y,[z,w=\varepsilon q^m]}$, isomorphic to $W_{0,\varepsilon q^m}$. This module has a nontrivial submodule $R_{0,\varepsilon q^m}$, which is the kernel of the Gram bilinear form on $W_{0,\varepsilon q^m}$. We now show that $e_0 \cdot \psi$ is not in the kernel submodule $R_{0,\varepsilon q^m}$ of $Y^{(0)}_{0,0,x,y,[z,w=\varepsilon q^m]}$. Let us denote by $\Phi$ the obvious homomorphism $W_{0,\varepsilon q^m} \to Y_{0,0,x,y,[z,w=\varepsilon q^m]}$ that maps any link state $v$ of $W_{0,\varepsilon q^m}$ to the link state of zero depth in $Y^{(0)}_{0,0,x,y,[z,w=\varepsilon q^m]}$ with identical loop segments, and the points a and b inserted together at the location of the marked point of $v$. The state $e_0 \cdot \psi$ has a unique preimage in $W_{0,\varepsilon q^m}$. Let us also recall that a bilinear form on $W_{0,\varepsilon q^m} \otimes Y_{0,0,x,y,[z,w=\varepsilon q^m]}$ was defined in [13]. This bilinear form satisfies the identity $\langle u, v \rangle = \langle u, \Phi(v) \rangle$, for all $u, v \in W_{0,\varepsilon q^m}$. In particular, we can write

$$\langle v_0(2m), \Phi^{-1}(e_0 \cdot \psi) \rangle = \langle v_0(2m), e_0 \cdot \psi \rangle = \beta \langle v_0(2m), P_{2m} \cdot u_{0,0}(2m) \rangle = \beta \, \sigma^{(0)}_{0,m}(\varepsilon q^m, x, y), \quad \text{(D.34)}$$

where we recall that $v_k(N)$ is defined in (45). This intermediate state is nonzero for the values of $y$ considered here. This proves that the state $e_0 \cdot \psi$ is *not* an element of the radical submodule $R_{0,\varepsilon q^m}$ in (D.29). As a result, we have the structure

$$Z_{0,0,x,y,[z,w=\varepsilon q^m]} \simeq \begin{bmatrix} \begin{array}{c} \phantom{xxx} W_{m,\varepsilon} \\ \swarrow \\ Q_{0,\varepsilon q^m} \\ \searrow \\ \phantom{xxx} R_{0,\varepsilon q^m} \end{array} \end{bmatrix} . \tag{D.35}$$

Moreover, from Proposition D.1, the state $\mu$ in (D.11) is nonzero, and as a consequence the pair $(\psi, \lambda)$ forms a Jordan cell of rank 2 that ties the two isomorphic factors $W_{m,\varepsilon}$ and $R_{0,\varepsilon q^m}$.

We end this section by proving the formula for $\gamma_m$.

PROOF OF PROPOSITION D.1. We define

$$S_{r,k} = \frac{1}{2r} \sum_{v=0}^{2r-1} \frac{\omega_{r,v}^k}{f_{m,n_\varepsilon} - f_{r,v}} , \tag{D.36}$$

so that

$$\gamma_m = \frac{\varepsilon^m}{2m} \sum_{k_1,\dots,k_m} (\varepsilon q)^{k_m} \prod_{r=1}^{m-1} q^{k_r} S_{r,k_{r+1}-k_r} , \tag{D.37}$$

where $k_r$ runs over the values $-r+1, -r+2, \dots, r-1$. Following the arguments presented in section 5.3 of [21], we rewrite $S_{r,k}$ as

$$S_{r,k} = -\frac{q^{-r}}{2r} \sum_{v=0}^{2r-1} \frac{\omega^{k+1}}{(\omega - \varepsilon q^{-r+m})(\omega - \varepsilon q^{-r-m})} \bigg|_{\omega \to \omega_{r,v}} = \oint_{\mathcal{C}_r} d\omega \, g_{r,k}(\omega), \tag{D.38}$$

where

$$g_{r,k}(\omega) = -\frac{q^{-r}}{2\pi i} \frac{\omega^k}{(\omega - \varepsilon q^{-r+m})(\omega - \varepsilon q^{-r-m})} \frac{1}{\omega^{2r} - 1}, \tag{D.39}$$

and the closed contour $\mathcal{C}_r$ encircles all the poles at $\omega = \omega_{r,\nu}$ in the counter-clockwise direction, but not the other poles of $g_{r,k}(\omega)$.

Let $k \in \{0, 1, \ldots, 2r\}$. In this case, $g_{r,k}(\omega)$ has no pole at $\omega = 0$. Moreover, changing variables to $y = \omega^{-1}$, we obtain an integral over $y$ whose integrand has no pole at $y = 0$. As a result, we may deform the contour $\mathcal{C}_r$ so that it instead encircles the two other poles of $g_{r,k}(\omega)$. Evaluating the corresponding residues, we find

$$\begin{aligned} S_{r,k} &= \sum_{\sigma \in \{+1,-1\}} \text{Res}\left[\frac{q^{-r}\omega^k}{(\omega - \varepsilon q^{-r+m})(\omega - \varepsilon q^{-r-m})} \frac{1}{\omega^{2r} - 1}, \omega \to \varepsilon q^{-r+\sigma m}\right] \\ &= \frac{\varepsilon^{k-1}}{q^m - q^{-m}}\left(\frac{q^{(-r+m)k}}{q^{(-r+m)2r} - 1} - \frac{q^{(-r-m)k}}{q^{(-r-m)2r} - 1}\right). \end{aligned} \tag{D.40}$$

For $k \notin \{0, 1, \ldots, 2r\}$, $S_{r,k}$ is obtained directly using the symmetry $S_{r,k+2r} = S_{r,k}$. We therefore write

$$\gamma_m = \frac{\varepsilon}{2m} \frac{1}{(q^m - q^{-m})^{m-1}} \sum_{k_1,\ldots,k_m} q^{k_m} \prod_{r=1}^{m-1} q^{k_r} \widehat{S}_{r,k_{r+1}-k_r}, \tag{D.41}$$

where

$$\widehat{S}_{r,k} = \begin{cases} \dfrac{q^{(-r+m)k}}{q^{(-r+m)2r} - 1} - \dfrac{q^{(-r-m)k}}{q^{(-r-m)2r} - 1}, & k \in \{0, 1, \ldots, 2r\}, \\[4mm] \dfrac{q^{(-r+m)(k+2r)}}{q^{(-r+m)2r} - 1} - \dfrac{q^{(-r-m)(k+2r)}}{q^{(-r-m)2r} - 1}, & k \in \{-2r, -2r+1, \ldots, -1\}. \end{cases} \tag{D.42}$$

We now evaluate the sums in (D.41), first for $k_m$, second for $k_{m-1}$, third for $k_{m-2}$, and so on. We find that

$$\gamma_m = \frac{\varepsilon}{2m} \frac{1}{(q^m - q^{-m})^{m-1}} \sum_{k_1,\ldots,k_t} q^{k_t}\left[\prod_{r=1}^{t-1} q^{k_r}\widehat{S}_{r,k_{r+1}-k_r}\right] K_t(k_t), \qquad t = 1, 2, \ldots, m, \tag{D.43}$$

for some functions $K_t(k)$ that satisfy

$$K_{t-1}(k) = \sum_{\ell=1-t}^{t-1} q^\ell \widehat{S}_{t-1,\ell-k} K_t(\ell), \tag{D.44}$$

and the initial condition $K_m(k) = 1$. This recusive system is solved inductively over decreasing values of $t$. The solution reads

$$K_t(k) = \frac{1}{(q-q^{-1})^{m-t}} \frac{[m]^{m-t}}{([m-1]!)^2} \sum_{r=-m+t,-m+t+2,\ldots,m-t} q^{kr} \frac{[\frac{m+r+t-2}{2}]! \, [\frac{m-r+t-2}{2}]!}{[\frac{m+r-t}{2}]! \, [\frac{m-r-t}{2}]!}, \tag{D.45}$$

where $[k]! = \prod_{j=1}^k [j]$. The proof of the induction step is tedious but straightforward. Indeed, starting from the right side of (D.44), one starts by splitting the sum in two, the first for $\ell < k$ and the second for $\ell \geqslant k$. These two sums are then evaluated individually using the geometric series. Finally, the resulting expression is simplified using some simple manipulations. Setting $t = 1$, we find

$$K_1(k) = \frac{1}{(q-q^{-1})^{m-1}} \frac{[m]^{m-1}}{([m-1]!)^2} \sum_{r=-m+1,-m+3,\ldots,m-1} q^{kr}, \tag{D.46}$$

and

$$\gamma_m = \frac{\varepsilon}{2m} \frac{K_1(0)}{(q^m - q^{-m})^{m-1}} = \frac{\varepsilon/2}{\prod_{j=1}^{m-1}(q^j - q^{-j})^2} \,, \tag{D.47}$$

ending the proof. ∎

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
