# Peer review of "Fusion of irreducible modules in the periodic Temperley--Lieb algebra"

_SciPost Physics, doi:SciPost Phys. 17, 132 (2024)_

## Round 1 · Referee Report · Anonymous (Referee 1) · 2024-3-15

Strengths

1-Original approach to bulk lattice fusion
2-Clear and pedagogical presentation

Report

This paper introduces a new class of representations of the affine Temperley-Lieb (aTL) algebra. These representations are close in spirit, but not the same, as a set of aTL representations defined by the authors in a previous paper. The introduction of these sets of representations are motivated by the search for a lattice analog of the bulk fusion rules in the logartihmic conformal field theory (CFT) describing loop models. The authors clearly indicate why the representations introduced in this paper are better candidates than the previous ones. It is an important open problem to describe fusion rules in logarithmic CFT. Although important results have been found in the case of boundary CFT through an algebraic lattice fusion procedure, a bulk lattice counterpart is yet to be found. I believe this work is an important step in this direction. The pedestrian approach it proposes is something to rely on to find new ideas for a putative systematic algebraic approach.

The paper is well written and structured. It contains in general a good amount of details and pedagogical explanations. I have only a few suggestions that could potentially improve the readability.

Requested changes

1-In page 22, it could be good to give details on how to derive the dimensions of modules. The reader is referred to the previous paper where the analogous proof is not very long. Maybe a similar, possibly shorter, proof could be included.

2-In page 23, in the description of non generic parameters, composition factors of the modules are given but never discussed before. It would be good to say more on them beforehand.

3-For the proof of the decomposition in the generic case in page 24, the reader is referred to the techniques of the previous paper. In the previous paper, explicit homomorphisms from standard modules were given. It would be nice to give explicit homomorphisms in this new case as well, if not too long. Or it could possibly be put in an appendix.

4- From page 52, the notation of tensor product with a square is defined as the fusion product of Vir. But it is used to denote the fusion product of both Vir and Vir x Vir. It would be good to emphasize this.

---

## Round 1 · Referee Report · Anonymous (Referee 2) · 2024-4-12

Report

This paper is the latest in a series of works that study constructions in appropriate statistical lattice models that mimic fusion products in their, presumably conformally invariant, scaling limits. Here, the authors are concerned with bulk fusion, rather than the chiral fusion typically studied by conformal field theorists, and their lattice models are built from the periodic Temperley--Lieb algebras.

In a previous paper, the authors proposed a definition for the bulk lattice fusion product of irreducible modules over these algebras. This paper notes that this definition does not correspond to an associative product and proposes an improved definition that does. The improvement is noted by the authors to be a little peculiar as it requires the introduction of one or two auxiliary parameters, depending on the specifics of the lattice modules. They also state that the improvement is an even better candidate for bulk lattice fusion, leaving the door open to further tweaks.

Irrespective of whether further tweaks are required, this paper reports an impressive amount of work, even if this work ultimately relies upon some very reasonable conjectures (Claims 3.1, 4.1 and 4.2; Conjectures 4.1 and 4.2). I expect that these claims are extremely difficult to prove, but it is nevertheless laudable that the authors indicate them so prominently, instead of burying them in dense prose.

Aside from the associativity result mentioned, they give the complete decomposition of the improved bulk lattice fusion product of two standard modules when the parameters are suitably generic. They also describe some results involving the radical submodules of the standard modules and the corresponding quotients. Finally, they also explore a few examples with non-generic parameters, illustrating that in this case the product involves reducible but indecomposable direct summands (as expected for a logarithmic scaling limit).

This is a remarkable, if extremely technical, improvement on the state of the art and so I warmly recommend publication. I have only a few small suggestions that the authors may wish to implement in a revision.

First, I would like to ask the authors to correct their terminology concerning singular and null vectors of the Virasoro algebra, eg. in between (1.6) and (1.7). A singular vector is a (non-generating) highest-weight state and a null vector is a state that is orthogonal to the entire module. So a radical submodule corresponds to null vectors, at least one of which is singular.

Second, I suggest that the authors think of adding some remarks about the "exactness" of their fusion definition. Assuming that bulk fusion in the scaling limit is modelled as a tensor product, one expects that fusing with a given fixed module is right-exact but perhaps not left-exact. Given the authors results about radicals and quotients, it should be possible to say if their results are consistent with right-exactness. It would be very interesting to know if they are consistent with left-exactness or not, because examples where left-exactness fails are still considered quite exotic and hard to understand in the CFT community.

[I actually wonder if this has something to do with the example (4.19) in which fusion is found to be incompatible with isomorphisms...]

I'd also like to see a little discussion around the fact that some fusion products are found to be 0, eg. (4.73) and (4.77), also QxQ in Section 5. What does this mean? Does it have any bearing of the existence of conjugate fields in the scaling limit?

Is there a typo in (3.8)? The w seems out of place.

Finally, I'd like to suggest replacing the term "unwinded" throughout by "unwound", just for grammar pedants... I don't like worrying about whether defects are running out of breath or not... :)

Recommendation

Ask for minor revision

---

## Round 2 · Referee Report · Anonymous (Referee 2) · 2024-9-6

Report

I thank the authors for the updated version which has addressed all my concerns. I warmly recommend publication.

Recommendation

Publish (easily meets expectations and criteria for this Journal; among top 50%)

---

## Round 2 · Referee Report · Anonymous (Referee 1) · 2024-10-3

Report

I thank the authors for taking into account my comments and I recommend the new version for publication.

Recommendation

Publish (easily meets expectations and criteria for this Journal; among top 50%)

---

## Round 2 · Author Response

Dear editors,

We thank the referees for their careful reading and comments on our manuscript. The version we are now resubmitting contains changes addressing the issues that they raised. You will find below our answers to their requests.

Best regards, Yacine Ikhlef and Alexi Morin-Duchesne

Answer to Referee 1:

  1. We wrote a new Appendix A describing the arguments leading to the dimension counting and to the decomposition of the modules Y. In working out these proofs, we found that the characterisation of the modules Y given in Section 3.2 wasn't quite clear or correct. We therefore reworked a large part of this section so that it is correct and clearer, and adapted the rest of the text accordingly.

  2. We removed the mention of composition factors in Section 3.3, as this indeed only starts to make sense once the decomposition of the modules Y is given in Section 3.4.

  3. In Appendix A, we added an explanation of the homomorphisms from standard modules to quotients of the modules Y.

  4. At the beginning of Section 6.3, we added a remark about using the same notation for fusion products over Vir and over Vir x Vir.

Answer to Referee 2:

  1. We modified the terminology of singular and null states according to the referee's comment.

  2. We agree with the referee that obtaining a functorial description of our construction and understanding its exactness are interesting questions -- we are currently working on this problem. This is beyond the scope of the current paper, as it would require that we understand how to define more generally the fusion of two arbitrary modules. We added a comment at the end of the conclusion stating that these questions are interesting open problems.

  3. We stress that none of the fusion products that we compute actually vanish. Some of them however vanish in certain channels [z,w]. In general, we express the "full" fusion product of two fields in terms of the fusion rules. If some fusion channels vanish, then the corresponding fusion rules contains finitely many channels, which is usual in rational CFT. We added a paragraph at the end of Section 4.2 to clarify this point.

  4. We fixed the typo in (3.8).

  5. We replaced "unwinded" by "unwound".

---

## Round 2 · List of Changes

• Added Appendix A, with details on the dimension counting and the decomposition of the modules Y.
  • Modified the description of link states in Section 3.2.
  • Added a comment at the end of the Conclusion, about the functorial description of our fusion scheme.
  • Added a paragraph at the end of Section 4.2 to clarify a point on vanishing fusion channels.
  • Implemented minor changes requested by the Referees (see resubmission letter).

---

## Editorial Decision

published